# *MedAgentBoard*: Benchmarking Multi-Agent Collaboration with Conventional Methods for Diverse Medical Tasks

**Yinghao Zhu**[1,2,*]**, Ziyi He**[2,*]**, Haoran Hu**[1,*]**, Xiaochen Zheng**[4,*]**,**
**Xichen Zhang**[2]**, Zixiang Wang**[1]**, Junyi Gao**[3,5]**, Liantao Ma**[1,6,†]**, Lequan Yu**[2,†]
[1]National Engineering Research Center for Software Engineering, Peking University
[2]School of Computing and Data Science, The University of Hong Kong
[3]Centre for Medical Informatics, The University of Edinburgh
[4]ETH Zurich    [5]Health Data Research UK
[6]Key Laboratory of High Confidence Software Technologies, Ministry of Education
yhzhu99@gmail.com, malt@pku.edu.cn, lqyu@hku.hk

## Abstract

The rapid advancement of Large Language Models (LLMs) has stimulated interest in multi-agent collaboration for addressing complex medical tasks. However, the practical advantages of multi-agent collaboration approaches remain insufficiently understood. Existing evaluations often lack generalizability, failing to cover diverse tasks reflective of real-world clinical practice, and frequently omit rigorous comparisons against both single-LLM-based and established conventional methods. To address this critical gap, we introduce *MedAgentBoard*, a comprehensive benchmark for the systematic evaluation of multi-agent collaboration, single-LLM, and conventional approaches. MedAgentBoard encompasses four diverse medical task categories: (1) medical (visual) question answering, (2) lay summary generation, (3) structured Electronic Health Record (EHR) predictive modeling, and (4) clinical workflow automation, across text, medical images, and structured EHR data. Our extensive experiments reveal a nuanced landscape: while multi-agent collaboration demonstrates benefits in specific scenarios, such as enhancing task completeness in clinical workflow automation, it does not consistently outperform advanced single LLMs (e.g., in textual medical QA) or, critically, specialized conventional methods that generally maintain better performance in tasks like medical VQA and EHR-based prediction. MedAgentBoard offers a vital resource and actionable insights, emphasizing the necessity of a task-specific, evidence-based approach to selecting and developing AI solutions in medicine. It underscores that the inherent complexity and overhead of multi-agent collaboration must be carefully weighed against tangible performance gains. All code, datasets, detailed prompts, and experimental results are open-sourced at: https://medagentboard.netlify.app/.

## 1 Introduction

Large Language Models (LLMs) have demonstrated remarkable capabilities across numerous domains, signaling a transformative era for artificial intelligence in medicine [1, 2]. Advanced models such as GPT-4 [3] and DeepSeek [4, 5] have achieved performance comparable to, or even exceeding, human physicians in medical licensing examinations [6, 7], demonstrating proficiency in comprehending complex medical language and addressing clinical inquiries [8, 9]. To augment these capabilities and address more intricate problems, LLM-driven multi-agent collaboration has emerged as a promising paradigm. This approach involves multiple, often specialized, LLM-based agents interacting and collaborating—frequently in role-playing scenarios—to solve complex tasks [10],

---

[*] Equal contribution, [†] Corresponding authors.

potentially mitigating some of the inherent reasoning limitations found in monolithic LLMs [11]. In the medical domain, initial explorations of multi-agent collaboration have reported encouraging results [12, 13, 14], particularly in tasks such as medical question answering (QA), where they occasionally outperform single-LLM approaches.

Despite this initial promise, a critical re-evaluation of the broader applicability and comparative advantages of multi-agent collaboration in healthcare is warranted, primarily due to two key limitations in existing research. First, current evaluations often lack **generalizability**, typically confining assessments to specific task types such as multiple-choice medical QA. This narrow focus overlooks the diversity of real-world clinical applications and fails to encompass tasks that accurately mirror actual clinical workflows, where clinicians may require free-form diagnostic support or complex data interpretation rather than merely selecting from predefined answers. Second, studies often present **incomplete baselines**, primarily benchmarking multi-agent collaboration approaches against single LLMs while neglecting rigorous comparisons with established conventional machine learning methods, which are generally fine-tuned on task-specific datasets. These conventional approaches may remain highly competitive, or even superior, in terms of accuracy, efficiency, or reliability for specific medical tasks. These prevailing gaps underscore an urgent research question: *To what extent do multi-agent collaboration approaches genuinely enhance capabilities across a diverse and realistic range of clinical contexts when benchmarked against both single LLMs and well-established conventional techniques?*

To address this question, we introduce *MedAgentBoard*, a benchmark meticulously designed to ensure its authority and relevance in healthcare. This is achieved by selecting tasks that reflect the diverse needs of key medical AI stakeholders—patients, clinicians, and researchers [15]—and encompass the varied data modalities and technical intricacies characteristic of real-world medical applications [16]. MedAgentBoard thus comprises four task categories: (1) medical (visual) question answering and (2) lay summary generation, primarily serving patients by making complex textual and visual medical information more accessible; and (3) structured Electronic Health Record (EHR) data predictive modeling and (4) clinical workflow automation, targeting clinicians and researchers by leveraging structured data for decision support and operational efficiency. MedAgentBoard's curated selection provides a robust platform for comparing AI approaches across a representative spectrum of medical challenges, thereby fostering fair evaluations to guide AI development and deployment in healthcare. A core tenet of MedAgentBoard is its commitment to a comprehensive comparison of multi-agent collaboration approaches and single LLMs against strong conventional baselines, thereby offering a more complete understanding of their relative merits. Our contributions are threefold:

- MedAgentBoard provides a comprehensive benchmark for the rigorous evaluation and extensive comparative analysis of multi-agent collaboration, single LLMs, and conventional methods across diverse medical tasks and data modalities. By synthesizing prior research with LLM-era evaluations, it directly addresses critical gaps in generalizability and the completeness of existing baselines.

- MedAgentBoard distinguishes itself from prior work (detailed in Table 1) by offering a unified platform for adjudicating the often conflicting claims regarding the efficacy of multi-agent collaboration. It provides clarity in a rapidly evolving landscape where the true advantages of such collaborative approaches are still under intense debate, underscoring the necessity for standardized and comprehensive evaluation.

- MedAgentBoard distills findings into actionable insights to assist researchers and practitioners in making informed decisions regarding the selection, development, and deployment of AI solutions in diverse medical settings.

Table 1: *Comparison of existing benchmarks for LLMs and multi-agent collaboration frameworks.*

| Benchmarks | Benchmarked Tasks | Benchmarked Methods | Brief Conclusion |
|---|---|---|---|
| MedHELM [17] | Text Generation, EHR Predictive Modeling, Medical QA, Programming | Single LLM | MedHELM provides fair comparison and assessment of LLM capabilities in healthcare settings. |
| MedAgentsBench [18] | Medical QA | Single LLM, Multi-agent | The latest thinking LLMs exhibit exceptional performance in complex medical reasoning tasks. |
| Strategic Reasoning [19] | 5-round Ultimatum Game, Personality pairings | Single LLM, Multi-agent | Multi-agent shows great potential for simulating strategic behavior consistent with human gameplay. |
| MAST [20] | Programming, Cross-app Web Tasks, Mathematical Reasoning, Knowledge QA | Single LLM, Multi-agent | Multi-agent does not consistently outperform a well-prompted single LLM baseline. |
| Multi-agent Debate [21] | Programming, Mathematical Reasoning, Knowledge QA | Single LLM, Multi-agent | Multi-agent seldom outperforms simple single LLM reasoning (CoT or, Self-Consistency). |
| MDAgents [12] | Medical QA, Medical VQA, Clinical Reasoning | Single LLM, Multi-agent | Multi-agent outperforms seven out of ten benchmarked tasks. |
| *MedAgentBoard* (Ours) | Medical (V)QA, EHR Predictive Modeling, Lay Summary Generation, Clinical Workflow Automation | Conventional Methods, Single LLM, Multi-agent | Multi-agent does not universally outperform advanced single LLMs or specialized conventional methods. |

## 2 Related Work

Table 2: *An illustrative overview of different multi-agent collaboration paradigms.*

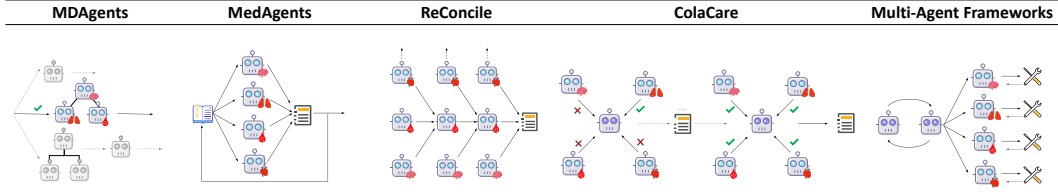

**Multi-agent collaboration.** An LLM agent is an LLM-driven system capable of autonomous, goal-directed behavior, encompassing aspects like reasoning, planning, and memory [22]. Multi-agent collaboration leverages multiple such agents to tackle complex problems that may be beyond the capabilities of a single agent. General-purpose collaborative agent frameworks like AutoGPT [23], XAgent [10], and MetaGPT [24] have demonstrated the potential of decomposing tasks and assigning specialized roles to different agents. The core of multi-agent collaboration frameworks lies in their collaborative mechanism design [11]. As illustrated in Table 2, in medical contexts, these mechanisms include prompting agents with distinct roles to reason (MDAgents [12]), discuss (MedAgents [14], ReConcile [13]), vote [25], debate [26], or simulate multi-disciplinary team discussions (ColaCare [27]). These interactions aim to converge on a shared, more robust response [26], demonstrating improvements in factuality, mathematical abilities, and overall reasoning capabilities of multi-agent solutions compared to individual agents [26, 28].

**Evaluating LLMs in healthcare applications.** Existing benchmarks that evaluate LLMs' capabilities in medicine [29, 30] typically employ medical QA datasets such as MedQA [31], PubMedQA [32], PathVQA [33], and VQA-RAD [34], focusing predominantly on closed-form medical QA tasks. This narrow focus limits the assessment of broader healthcare applications. Recent critiques have highlighted that medical exam benchmarks provide limited signals for assessing true clinical utility [35]. In response, newer benchmarks have begun to extend evaluation to open-ended free-form QA settings [7] through human evaluation [36] or by using LLM-as-a-judge approaches [37]. However, these benchmarks still exhibit limitations in their coverage of data modalities and tasks beyond question answering, underscoring the need for evaluations that more accurately measure performance on real-world medical tasks [17]. Moreover, most of these benchmarks compare only LLM-based methods, neglecting conventional non-LLM approaches that might still remain competitive [38]. MedAgentBoard addresses these gaps by providing a comprehensive evaluation framework that encompasses diverse methods, modalities, and a wider range of tasks that better reflect clinical utility in real-world healthcare settings.

## 3 *MedAgentBoard*: Tasks, Datasets, Evaluations, and Methods

As illustrated in Figure 1, MedAgentBoard is structured around four distinct medical task categories, chosen to represent a diverse range of clinical needs, data modalities, and reasoning complexities. For each task, we aim to compare multi-agent collaboration, single LLM approaches, and strong conventional baselines, providing a holistic view of their relative capabilities.

### 3.1 Task 1: Medical (Visual) Question Answering

**Tasks and datasets.** This task evaluates the ability of AI systems to answer questions based on medical textual knowledge (QA) or a combination of visual and textual inputs (VQA). It encompasses two primary sub-types: multiple-choice QA, testing specific knowledge recall and discriminative reasoning, and open-ended free-form QA, assessing generative capabilities and nuanced understanding. To support this, we employ established datasets: MedQA [31], featuring USMLE-style questions, and PubMedQA [32], comprising questions based on biomedical abstracts, for textual QA. For medical VQA, which integrates visual information from pathology slides or radiological images, we use PathVQA [33] and VQA-RAD [34]. These datasets are selected for their widespread use in benchmarking medical AI and their representation of diverse question styles and modalities.

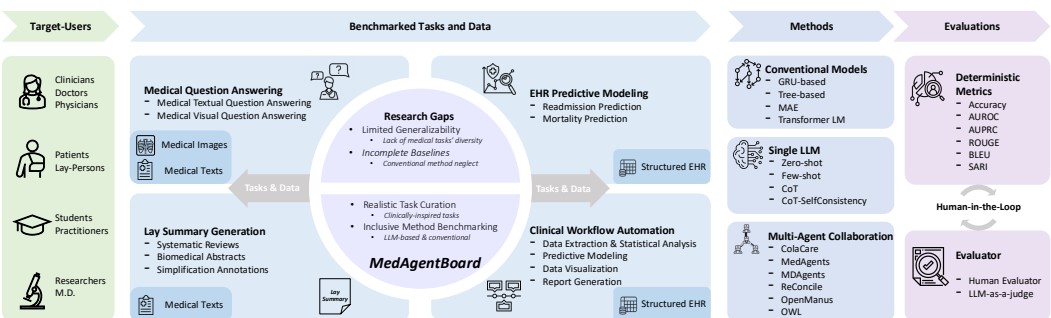

Figure 1: *The illustrative overview of MedAgentBoard.*

**Evaluations and methods.** Performance on multiple-choice QA is measured by accuracy. For free-form QA, we use LLM-as-a-judge [37] scoring for semantic correctness, clinical relevance, and factual consistency. Conventional methods for QA/VQA typically involve fine-tuning pre-trained models (e.g., BioLinkBERT [39], GatorTron [40] for textual QA; M³AE [41], BiomedGPT [42], MUMC [43], LLaVA-Med [44], Med-Flamingo [45] for VQA) as classifiers. This inherently limits their applicability to free-form QA beyond constrained answer vocabularies. Single LLMs (for text) and Vision-Language Models (VLMs) (for VQA) are evaluated using a range of prompting strategies including zero-shot, few-shot in-context learning (ICL) [46], Chain-of-Thought (CoT) [47], and self-consistency [25], chosen to span various levels of prompting complexity. Multi-agent collaboration is represented by frameworks such as MedAgents [14], ReConcile [13], MDAgents [12], and ColaCare [27]. These are adapted to facilitate discussion among LLM-based agents, often simulating different clinical roles, to derive a collaborative answer. This collaboration relies solely on textual exchange to ensure a fair comparison across methodologies.

## 3.2 Task 2: Lay Summary Generation

**Tasks and datasets.** Lay summary generation focuses on transforming complex medical texts, such as research articles, into versions that are accurate, concise, and readily comprehensible to a non-expert audience [48]. This task rigorously tests not only the comprehension of specialized medical language but also the nuanced skill of rephrasing information for laypersons without sacrificing critical meaning or introducing inaccuracies [49]. For this task, we leverage a diverse set of datasets: Cochrane [50], providing plain language summaries of systematic reviews; eLife [51] and PLOS [51], containing author-written summaries of research articles. Additionally, specialized corpora such as Med-EASi [52], which focuses on fine-grained simplification annotations, and PLABA [53], a dataset of plain language adaptations of biomedical abstracts, are included. This selection ensures a comprehensive assessment across varied styles and complexities of medical text simplification.

**Evaluations and methods.** Evaluation relies on ROUGE-L [54] to measure content overlap with reference summaries and SARI [55] to specifically assess simplification effectiveness (appropriateness of word additions, deletions, and retentions). Conventional approaches involve fine-tuning pre-trained sequence-to-sequence models like T5 [56], PEGASUS [57], and BART [58] on specific lay summary datasets. Single LLM approaches are evaluated using zero-shot prompting, optimized prompts with detailed guidelines, and optimized prompts with few-shot ICL strategies. For multi-agent collaboration, we adapt principles from AgentSimp [59], a general text simplification framework. Our implementation orchestrates nine specialized agents in a pipeline: a "project director" establishes guidelines, a "structure analyst" extracts key information, and a "content simplifier" performs initial transformation. This summary then undergoes refinement by specialized agents ("supervisor", "metaphor analyst", "terminology interpreter") before final review by the "proofreader". This structured, multi-agent pipeline is chosen to mimic a rigorous, collaborative editorial process.

## 3.3 Task 3: EHR Predictive Modeling

**Tasks and datasets.** This task centers on predicting patient-specific clinical outcomes using structured Electronic Health Record (EHR) data. We focus on two clinically significant prediction targets: in-hospital patient mortality and 30-day hospital readmission [60, 61, 62]. These tasks require models to discern complex predictive patterns from heterogeneous, often high-dimensional structured

data (e.g., patient demographics, laboratory tests), presenting challenges distinct from natural language or image processing [63]. For robust evaluation, we employ data from established large-scale databases: MIMIC-IV [64, 65], a comprehensive de-identified critical care database, for predicting both in-hospital mortality and 30-day readmission; and the publicly accessible Tongji Hospital (TJH) dataset [66], containing COVID-19 patient outcomes, for mortality prediction. Preprocessing steps transform raw EHR data into feature vectors suitable for each modeling paradigm: data from the latest patient visit (for tabular models) or longitudinal data from all patient visits (for deep learning-based sequential models) are used for conventional machine learning [67, 68, 69, 70, 71, 72]. For LLM-based prompting, we orchestrate a prompt template comprising EHR information with task instructions, reference ranges, and units, following best practices for EHR predictive modeling with LLMs [38].

**Evaluations and methods.** Standard classification metrics, Area Under the Receiver Operating Characteristic curve (AUROC) and Area Under the Precision-Recall Curve (AUPRC), are employed. Both metrics are valuable for imbalanced healthcare datasets. Conventional methods include traditional machine learning models (Decision Tree, XGBoost [73]), deep learning models (GRU, LSTM), and EHR-specific deep learning models (AdaCare [74], ConCare [75], GRASP [76]) designed for longitudinal EHR data. Single LLM approaches involve prompting the LLM with structured patient data, formatted as text, for zero-shot clinical outcome prediction [77]. Multi-agent collaboration approaches (e.g., MedAgents [14], ReConcile [13], ColaCare [27]) adapt QA-like frameworks where agents debate risk factors from textualized data. MDAgents [12] is excluded from this task as its emphasis on complex interaction modes and checks is less relevant for structured data prediction, potentially reducing its utility to that of simpler fixed-interaction agents in this context.

### 3.4 Task 4: Clinical Workflow Automation

**Tasks and datasets.** Clinical workflow automation evaluates AI systems' capabilities in handling routine to complex clinical data analysis tasks traditionally requiring significant clinical expertise. We focus on four distinct task types representing common health data science scenarios: (1) data extraction and statistical analysis (identification, cleaning, and transformation of relevant variables from structured EHR datasets); (2) predictive modeling (model selection, training procedures, evaluation); (3) data visualization (creation of appropriate visual representations); and (4) report generation (synthesis of analytical findings into integrated documentation highlighting key insights). These tasks vary in complexity, providing a comprehensive evaluation landscape. We use the longitudinal structured MIMIC-IV and TJH datasets, consistent with Task 3. To synthesize tasks in these four categories, we initially generate a larger pool of analytical questions using Gemini 2.5 Pro [78] (Gemini-2.5-Pro-Exp-03-25), prompted with schema information and data samples from these datasets. After careful manual review and selection, we curate a benchmark suite of 100 analytical questions (50 for MIMIC-IV, 50 for TJH) designed to simulate real-world clinical data analysis scenarios. To ensure task diversity, we categorize the questions across four components of the analytical workflow. For data extraction and statistical analysis, we define four sub-tasks: data wrangling, data querying, data statistics, and data preprocessing. For data visualization tasks, we include two types: one focusing on extracting data from datasets, performing statistical analysis, and visualizing data distributions; the other involving first defining a modeling task and then requesting visualization of model parameters or performance metrics. Additional task categories include modeling and reporting. The generation process employs distinct prompt templates tailored to each analytical component, ensuring a comprehensive coverage of tasks with varying complexity levels representative of typical analytical workflows in healthcare research.

**Evaluations and methods.** Evaluation is conducted by an expert panel that assesses each generated solution by comparing it against a manually curated "reference answer". For data extraction/statistics, we assess correctness of data selection, transformation, and missing value handling. For predictive modeling, we evaluate appropriateness of model selection, training implementation, inclusion of necessary metrics, and adherence to validation practices. For data visualization, assessment covers correctness of techniques, alignment with objectives, and readability. For report generation, we examine completeness, accuracy, and coherence. Multiple independent evaluators examine each solution, categorize errors, and synthesize results. We compare single LLM approaches (model receives task content and dataset schema to generate Python code) with multi-agent collaboration frameworks. For the latter, we evaluate three established frameworks known for orchestrating

specialized agents for analytical tasks: SmolAgents [79], OpenManus [80], and Owl [81]. All methods are evaluated on their ability to produce accurate, executable, and clinically relevant analytical solutions.

# 4 Experimental Results

This section presents a synthesis of experimental findings across all task categories in MedAgentBoard. Our goal is to provide a nuanced understanding of when each modeling paradigm (conventional, single LLM, multi-agent collaboration) is most suitable for specific medical applications.

## 4.1 Benchmarking Results on Medical QA and VQA

Our experiments on medical QA/VQA (Table 3) reveal distinct performance patterns. In textual medical QA, LLM-based approaches demonstrate a clear advantage over conventional methods. Advanced prompting techniques, such as CoT-SC, achieve top scores on MedQA multiple-choice (89.90%). Notably, highly capable single LLMs, even with simpler zero-shot prompting (e.g., DeepSeek-V3 on PubMedQA free-form, achieving 91.23% LLM-as-a-judge score), can deliver promising results. While multi-agent collaboration frameworks like MedAgents show competitiveness (e.g., 83.85% on PubMedQA multiple-choice), they do not consistently surpass the best single LLM configurations.

Conversely, in medical VQA, specialized conventional VLMs such as M³AE, MUMC, and BiomedGPT maintain a dominant position. Their superiority likely arises from direct fine-tuning on task-specific image-text pairs or extensive pre-training on relevant medical VQA datasets. This creates a substantial performance gap for current general-purpose VLMs, including both single VLMs and those based on multi-agent collaboration approaches. The added complexity of multi-agent collaboration, therefore, requires careful justification against tangible benefits, especially when simpler, well-prompted single LLMs or specialized conventional models offer strong performance.

Table 3: *Benchmarking results for medical QA and VQA tasks.*

| Category | Methods | Medical QA | | | Medical VQA | | |
|---|---|---|---|---|---|---|---|
| | | MedQA (T, MC) | PubMedQA (T, MC) | PubMedQA (T, FF) | PathVQA (T I, MC) | VQA-RAD (T I, MC) | VQA-RAD (T I, FF) |
| Conventional | BioLinkBERT | 32.45±2.90 | 70.40±3.17 | - | - | - | - |
| | Gatortron | 36.60±3.11 | 59.30±3.68 | - | - | - | - |
| | M³AE | - | - | - | 90.65±1.61 | **89.05**±2.04 | **71.96**±2.25 |
| | BiomedGPT | - | - | - | 86.95±1.57 | 83.50±1.96 | 71.17±2.14 |
| | MUMC | - | - | - | **91.40**±1.74 | 84.85±1.21 | 68.44±1.42 |
| | LLaVA-Med | - | - | - | 59.25±2.12 | 48.70±3.04 | 19.94±2.22 |
| | Med-Flamingo | - | - | - | 66.15±1.95 | 45.10±2.01 | 18.38±3.05 |
| Single LLM | Zero-shot | 77.50±2.57 | 80.60±3.01 | **91.23**±0.78 | 66.90±3.80 | 67.45±2.41 | 46.42±2.08 |
| | Few-shot | 76.85±2.69 | 77.45±2.39 | 89.35±0.87 | 65.35±4.07 | 65.85±2.88 | 43.69±3.84 |
| | SC | 77.70±2.62 | 81.15±3.11 | 90.86±0.86 | 66.40±3.49 | 67.45±2.41 | 46.20±2.22 |
| | CoT | 87.30±2.79 | 83.30±2.90 | 83.59±0.79 | 73.40±2.75 | 68.95±2.48 | 38.88±2.35 |
| | CoT-SC | **89.90**±2.43 | 83.35±2.67 | 84.25±1.14 | 74.50±3.20 | 69.55±2.35 | 39.61±2.82 |
| Multi-agent | MedAgents | 85.25±2.67 | **83.85**±2.52 | 81.63±1.18 | 75.90±3.77 | 77.10±2.42 | 43.02±2.38 |
| | ReConcile | 78.00±3.39 | 77.85±3.71 | 78.21±1.00 | 49.00±3.94 | 70.45±4.66 | 43.02±2.82 |
| | MDAgents | 78.80±2.53 | 77.20±2.69 | 62.54±1.12 | 72.25±4.06 | 67.85±3.39 | 45.70±3.03 |
| | ColaCare | 84.65±2.70 | 83.50±2.32 | 81.72±0.79 | 74.45±3.74 | 74.05±2.25 | 44.47±2.60 |

***Note:*** T : Text modality; I : Image modality. MC: Multiple-choice question answering; FF: Free-form (open-ended) question answering. Few-shot employs two example QA pairs extracted from training set; SC: Self-consistency; CoT: Chain-of-thought. Accuracy (%) is assessed for MC, while LLM-as-a-judge score is assessed for FF settings. All metrics are the higher, the better. **Bold** indicates the best performance, and Underlined indicates the second-best performance per column (dataset and task). All scores are reported as mean±standard deviation by applying bootstrapping on all test set samples 10 times. Test sets are sampled from official splits to ensure representative evaluation. Training and validation sets for conventional methods use the datasets' original splits. Specifically, we sample 200 questions from the original test set for each dataset's provided settings (except VQA-RAD FF setting, which contains only 179 open-ended questions). For PathVQA and VQA-RAD's closed-form multiple-choice, we extract questions with yes/no answers to provide nuanced choices to the LLM. For LLM-based approaches: DeepSeek-V3-0324 [4] is adopted to act as each agent, with Qwen-VL-Max for visual content reasoning. As ReConcile methods encourage diversity for agent assignment, ReConcile uses Deepseek-V3-0324, Qwen-Max-Latest, Qwen-VL-Max for QA, and Qwen-VL-Max, Qwen2.5-VL-32B, Qwen2.5-VL-72B [82] for VQA.

*[Task 1's Key Findings and Implications]* ① Advanced general-purpose single LLM (DeepSeek) excel in textual MedQA, often matching or exceeding multi-agent collaboration; ② Specialized conventional VLMs remain superior for MedVQA; general-purpose VLMs (single/multi-agent) significantly lag; ③ Multi-agent benefits are inconsistent in QA/VQA; complexity must be weighed against performance gains.

## 4.2 Benchmarking Results on Lay Summary Generation

For lay summary generation (Table 4), conventional fine-tuned sequence-to-sequence models like BART-CNN (e.g., 42.24% ROUGE-L on Cochrane) and PEGASUS (e.g., 59.11% ROUGE-L on PLABA) consistently achieve high scores on automated metrics such as ROUGE-L and SARI across diverse datasets. This highlights their proficiency in learning the specific stylistic transformations and content mappings required for this task from large parallel corpora.

In contrast, while single LLMs can produce fluent and readable summaries, neither they nor the multi-agent collaboration approach (AgentSimp) consistently surpass these fine-tuned conventional models based on the automated metrics used. For instance, on Cochrane, BART-CNN achieves 42.24% ROUGE-L, while the best LLM-based method (Opt.+ICL) reaches 35.58%, and AgentSimp scores 34.25%. This observation might be partly attributed to automated metrics often favoring outputs that are stylistically similar to the reference training data, which conventional models are explicitly optimized to generate. The evaluated multi-agent collaboration does not demonstrate a clear advantage over well-prompted single LLMs or the leading conventional methods in terms of these scores. Interestingly, highly capable LLMs such as DeepSeek-V3 can achieve commendable performance with basic prompting, suggesting that with latest advanced LLMs, extensive prompt engineering may be less critical for this task, a finding also reflected in the DeepSeek-R1's technical report [5].

Table 4: *Benchmarking results for the lay summary generation task.*

| Category | Methods | Cochrane | | eLife | | PLOS | | Med-EASi | | PLABA | |
|---|---|---|---|---|---|---|---|---|---|---|---|
| | | RL(↑) | SARI(↑) | RL(↑) | SARI(↑) | RL(↑) | SARI(↑) | RL(↑) | SARI(↑) | RL(↑) | SARI(↑) |
| Conventional | BART | 37.82±0.66 | 37.42±0.22 | 46.02±0.32 | 45.62±0.40 | 41.30±0.48 | 37.37±0.25 | 44.78±1.89 | **45.31**±1.43 | 57.70±0.97 | 42.02±0.45 |
| | T5 | 22.88±0.67 | 34.95±0.34 | 44.26±0.43 | 45.22±0.22 | 41.09±0.40 | 37.29±0.24 | **46.20**±2.32 | 44.86±1.50 | 57.19±1.11 | 40.04±0.30 |
| | BART-CNN | **42.24**±0.72 | **39.75**±0.36 | **47.08**±0.32 | 46.18±0.52 | **44.24**±0.53 | 37.43±0.32 | 44.74±2.08 | 45.15±1.13 | 58.86±0.97 | **42.91**±0.38 |
| | PEGASUS | 41.64±0.65 | 39.41±0.45 | 46.08±0.51 | **46.30**±0.32 | 42.59±0.46 | 37.41±0.17 | 44.16±1.98 | 43.47±1.49 | **59.11**±1.02 | 41.82±0.68 |
| Single LLM | Basic | 33.65±0.51 | 38.29±0.36 | 29.43±0.44 | 42.88±0.47 | 32.84±0.48 | 37.61±0.41 | 24.80±0.95 | 36.12±1.42 | 37.56±0.44 | 32.16±0.67 |
| | Optimized | 33.85±0.60 | 38.25±0.41 | 31.47±0.24 | 43.34±0.65 | 31.30±0.42 | 36.85±0.46 | 19.00±0.63 | 36.92±1.24 | 38.16±0.39 | 32.06±0.70 |
| | Opt.+ICL | 35.58±0.63 | 38.56±0.28 | 33.12±0.38 | 43.85±0.57 | 33.00±0.45 | **37.84**±0.42 | 22.58±0.63 | 37.63±1.30 | 41.37±0.42 | 33.90±0.66 |
| Multi-agent | AgentSimp | 34.25±0.55 | 38.50±0.29 | 30.21±0.32 | 42.78±0.61 | 31.94±0.35 | 37.17±0.22 | 22.77±1.36 | 36.61±1.41 | 37.28±0.50 | 32.43±0.75 |

*Note:* Metrics reported are ROUGE-L (RL) and SARI. ↑ denotes higher is better. **Bold** indicates the best performance, and Underlined indicates the second-best performance per column (Dataset and Metric). We use the Genetics subset for the PLOS dataset. We sample 100 source text - target simplified text pairs from the original test set for each dataset. For training and validation of conventional models, we merge the original training and validation sets from each dataset, then randomly divide this combined data. All scores are reported as mean±standard deviation by applying bootstrapping on all test set samples 10 times. For LLM-based approaches: `DeepSeek-V3-0324` is adopted to act as each agent or for single LLM prompting. **Opt.+ICL** builds upon the optimized prompting setting by additionally providing two in-context learning examples. The BART model uses huggingface `facebook/bart-large`, BART-CNN refers to `facebook/bart-large-cnn`, T5: `google-t5/t5-base`, and PEGASUS: `google/pegasus-large`.

> *[Task 2's Key Findings and Implications]* ① Fine-tuned conventional models (e.g., BART, PEGASUS) lead in lay summary generation based on ROUGE/SARI; ② Single LLMs and current multi-agent collaboration approaches do not consistently outperform these specialized models on automated metrics; ③ Advanced LLMs can perform well with simple prompts, questioning the necessity of complex multi-agent setups for this task.

## 4.3 Benchmarking Results on EHR Predictive Modeling

In EHR predictive modeling (Table 5), conventional methods demonstrate clear superiority. Specialized models, including sequence-based deep learning approaches like GRU, LSTM, and AdaCare for longitudinal MIMIC-IV data (e.g., AdaCare achieving an AUROC of 94.28% for MIMIC-IV mortality), and ensemble methods such as XGBoost for TJH mortality (AUROC of 98.05%), significantly outperform LLM-based strategies. These conventional models are inherently better suited for capturing complex numerical patterns and temporal dependencies within structured EHR data.

State-of-the-art single LLMs, including GPT-4o and DeepSeek-R1, exhibit notable zero-shot predictive capabilities (e.g., GPT-4o AUROC of 85.99% for MIMIC-IV mortality; 95.72% for TJH mortality). While impressive for models not explicitly trained on these specific tasks, their performance does not match that of fully trained conventional models. Their current utility in this domain might be confined to scenarios with extremely limited data or for rapid prototyping.

Multi-agent collaboration methods, such as MedAgents, ReConcile, and ColaCare, built upon capable base LLMs like DeepSeek-V3, generally show performance improvements over their single base LLM (e.g., for MIMIC-IV mortality, ColaCare's 82.91% AUROC versus DeepSeek-V3's 76.86%). However, these multi-agent approaches do not consistently outperform the best-performing single LLMs like GPT-4o or DeepSeek-R1 and remain substantially outperformed by conventional methods.

This suggests that current QA-style collaborative frameworks may not be optimally suited for structured data prediction, and the observed modest improvements are largely driven by the power of the underlying base LLM rather than a transformative advantage from the multi-agent strategy itself.

Table 5: *Benchmarking results for the EHR predictive modeling task.*

| Category | Methods | MIMIC-IV Mortality | | MIMIC-IV Readmission | | TJH Mortality | |
|---|---|---|---|---|---|---|---|
| | | AUROC($\uparrow$) | AUPRC($\uparrow$) | AUROC($\uparrow$) | AUPRC($\uparrow$) | AUROC($\uparrow$) | AUPRC($\uparrow$) |
| Conventional | Decision Tree | 51.81$\pm$3.69 | 10.48$\pm$2.64 | 51.55$\pm$2.72 | 23.65$\pm$3.72 | 92.20$\pm$1.83 | 87.79$\pm$3.04 |
| | XGBoost | 64.62$\pm$4.97 | 17.66$\pm$5.12 | 64.23$\pm$4.34 | 34.31$\pm$6.65 | 98.05$\pm$0.94 | 95.58$\pm$2.18 |
| | GRU | 92.49$\pm$3.03 | 72.05$\pm$7.58 | 81.30$\pm$3.81 | 63.70$\pm$6.56 | 93.57$\pm$1.71 | 90.40$\pm$3.19 |
| | LSTM | 93.12$\pm$3.24 | 76.18$\pm$7.90 | **82.52**$\pm$3.78 | 66.32$\pm$6.58 | 92.98$\pm$1.91 | 86.97$\pm$4.12 |
| | AdaCare | **94.28**$\pm$3.52 | **81.93**$\pm$6.97 | 82.26$\pm$3.80 | **68.82**$\pm$6.76 | **99.02**$\pm$0.46 | **98.86**$\pm$0.53 |
| | ConCare | 94.08$\pm$3.70 | 80.65$\pm$6.98 | 79.17$\pm$4.42 | 64.27$\pm$6.97 | 91.00$\pm$2.14 | 91.72$\pm$2.30 |
| | GRASP | 93.14$\pm$3.03 | 72.55$\pm$8.36 | 77.76$\pm$4.17 | 62.42$\pm$7.08 | 94.25$\pm$1.58 | 92.03$\pm$2.54 |
| Single LLM | OpenBioLLM-8B | 58.69$\pm$6.06 | 12.85$\pm$3.77 | 50.21$\pm$4.97 | 24.23$\pm$4.02 | 56.75$\pm$3.92 | 49.76$\pm$4.67 |
| | Qwen2.5-7B | 61.57$\pm$7.12 | 13.58$\pm$3.17 | 55.86$\pm$3.98 | 25.32$\pm$4.02 | 79.83$\pm$2.68 | 70.87$\pm$4.61 |
| | Gemma-3-4B | 57.78$\pm$7.40 | 15.16$\pm$5.11 | 60.02$\pm$4.23 | 29.05$\pm$5.37 | 76.01$\pm$3.46 | 71.62$\pm$4.97 |
| | DeepSeek-V3 | 76.86$\pm$4.71 | 33.47$\pm$9.58 | 62.68$\pm$4.49 | 30.91$\pm$5.30 | 89.67$\pm$1.90 | 82.93$\pm$3.58 |
| | GPT-4o | 85.99$\pm$3.85 | 42.20$\pm$9.92 | 62.72$\pm$4.87 | 34.43$\pm$5.73 | 95.72$\pm$1.21 | 93.04$\pm$2.08 |
| | HuatuoGPT-o1-7B | 70.39$\pm$7.60 | 20.33$\pm$5.51 | 50.54$\pm$4.88 | 24.30$\pm$4.22 | 85.34$\pm$2.61 | 77.31$\pm$4.26 |
| | DeepSeek-R1-7B | 40.94$\pm$3.97 | 9.43$\pm$2.27 | 53.19$\pm$4.13 | 24.53$\pm$3.66 | 52.70$\pm$1.95 | 47.89$\pm$4.09 |
| | DeepSeek-R1 | 83.95$\pm$4.60 | 42.10$\pm$9.95 | 73.92$\pm$3.78 | 43.59$\pm$6.42 | 85.59$\pm$1.97 | 76.87$\pm$3.56 |
| | o3-mini-high | 71.23$\pm$7.19 | 28.99$\pm$7.88 | 63.30$\pm$4.85 | 36.13$\pm$6.18 | 84.42$\pm$2.52 | 75.65$\pm$4.48 |
| Multi-agent | MedAgents | 81.53$\pm$4.93 | 35.26$\pm$8.09 | 67.55$\pm$4.24 | 41.13$\pm$4.72 | 88.16$\pm$2.00 | 81.03$\pm$3.16 |
| | ReConcile | 77.81$\pm$5.19 | 33.57$\pm$9.15 | 68.86$\pm$4.35 | 49.19$\pm$6.20 | 93.58$\pm$1.71 | 88.38$\pm$3.65 |
| | ColaCare | 82.91$\pm$4.49 | 34.72$\pm$8.95 | 68.85$\pm$4.18 | 45.25$\pm$5.81 | 89.34$\pm$1.78 | 81.63$\pm$3.27 |

*Note:* $\uparrow$ denotes higher is better. **Bold** indicates the best performance, and Underlined indicates the second-best performance per column (dataset and task). All scores are reported as mean$\pm$standard deviation by applying bootstrapping on all test set samples 100 times. We sample 200 patients from the original test set for each dataset's prediction task. For LLM-based approaches (single and multi-agent), predictions are made in a zero-shot manner, with patient EHR data formatted as text prompts. For multi-agent models, the base LLM for each agent is `DeepSeek-V3-0324`. ReConcile uses `DeepSeek-V3-0324`, `Qwen-Max-Latest`, `Qwen-VL-Max` (consistent with the non-visual agents in Task 1). Specific LLMs used for single LLM rows are listed in the table.

> **[Task 3's Key Findings and Implications]** ① Conventional ML/DL models (e.g., AdaCare, XGBoost) significantly outperform all LLM-based approaches in EHR prediction; ② Advanced single LLMs (e.g., GPT-4o) show zero-shot potential but lag conventional methods and are not consistently surpassed by current multi-agent collaboration; ③ The complexity of multi-agent frameworks is not justified by performance gains in structured EHR prediction.

## 4.4 Benchmarking Results for Clinical Workflow Automation

For clinical workflow automation (Table 6), our results indicate that multi-agent collaboration can offer advantages in task completeness over single LLM approaches, particularly as such collaborative approaches are often designed with tool-use capabilities (e.g., Python code execution) crucial for such tasks. Frameworks such as SmolAgent and OpenManus generally achieve higher rates of successfully generating outputs for components like modeling code, visualizations, and reports, thereby reducing instances of "No Result". For example, in the TJH modeling task, OpenManus achieves a 64.0% "Correct" rate with only 4.17% "No Result", compared to the single LLM which has 50.00% "No Result". The reliability of these human-assessed findings is supported by moderate to substantial inter-rater agreement (Fleiss' Kappa of 0.61 for Data, 0.56 for Modeling, 0.54 for Visualization, and 0.40 for Reporting).

Despite these improvements in completeness, the overall rate of "Correct" end-to-end solutions remains modest across all methods and datasets. This underscores the substantial challenge of fully automating complex clinical data analysis workflows, with correct modeling, visualization, and reporting rarely exceeding 40-50%, and often being much lower, particularly on the more complex MIMIC-IV dataset (e.g., SmolAgent 29.25% "Correct" for MIMIC-IV Visualization). Data extraction and basic data manipulation tasks (selection, filtering, simple statistics) represent the most successfully automated component, with SmolAgent achieving 90.25% "Correct" on MIMIC-IV for the Data task. Performance tends to degrade significantly for subsequent, more intricate workflow stages.

We also observe significant performance variability among different multi-agent frameworks. Open-Manus and SmolAgent generally outperform Owl. Single LLM approaches often struggle with maintaining context and correctly sequencing complex analytical steps, as evidenced by high rates of "Model Not Saved" or "No Result" in modeling. Overall, the application of general-purpose multi-agent frameworks to healthcare domains still demonstrates substantial room for improvement.

Table 6: *Benchmarking results for the clinical workflow automation task on MIMIC-IV and TJH.*

| Task Type | Evaluation Category | MIMIC-IV | | | | TJH | | | |
|---|---|---|---|---|---|---|---|---|---|
| | | Single LLM | SmolAgents | OpenManus | Owl | Single LLM | SmolAgents | OpenManus | Owl |
| **Data** | Correct | 80.58% | **90.25%** | 65.33% | 50.00% | 67.92% | 70.54% | **78.23%** | 37.23% |
| | No Result | 16.67% | 4.17% | 20.84% | 34.66% | 1.23% | 3.85% | 0.00% | 32.08% |
| | Incorrect Answer | 2.75% | 4.17% | 13.84% | 6.91% | 15.46% | 20.38% | 11.38% | 15.31% |
| | Incomplete/Partial | 0.00% | 0.00% | 0.00% | 8.42% | 15.38% | 5.23% | 7.77% | 11.54% |
| | Correct w/ Presentation Issues | 0.00% | 1.42% | 0.00% | 0.00% | 0.00% | 0.00% | 2.61% | 3.85% |
| **Modeling** | Correct | 9.08% | **47.62%** | 39.84% | 0.00% | 8.42% | 48.91% | **64.0%** | 15.34% |
| | No Result | 14.15% | 12.77% | 15.38% | 76.92% | 50.00% | 0.00% | 4.17% | 41.67% |
| | Preprocessing Only | 0.00% | 0.00% | 11.61% | 18.08% | 0.00% | 0.00% | 4.17% | 30.83% |
| | Missing Metrics | 0.00% | 12.85% | 11.54% | 1.31% | 1.42% | 20.91% | 13.92% | 4.17% |
| | Model Not Saved | 57.46% | 6.23% | 6.23% | 3.69% | 31.83% | 13.50% | 5.42% | 8.00% |
| | Anomalous Numerical Results | 15.46% | 9.00% | 11.54% | 0.00% | 0.00% | 4.17% | 4.17% | 0.00% |
| | Fails Requirements | 3.85% | 11.54% | 3.85% | 0.00% | 8.34% | 12.50% | 4.17% | 0.00% |
| **Visualization** | Correct | 18.09% | **29.25%** | 22.34% | 12.5% | 48.69% | 44.92% | 46.23% | 32.08% |
| | No Visualization | 41.67% | 8.33% | 8.33% | 58.33% | 25.85% | 23.00% | 23.08% | 43.54% |
| | Anomalous Numerical Results | 23.58% | 30.5% | 33.42% | 20.83% | 5.08% | 3.85% | 7.69% | 10.3% |
| | Poor Readability | 9.75% | 15.33% | 11.17% | 8.33% | 0.00% | 2.61% | 5.08% | 1.31% |
| | Info Not Extractable | 4.17% | 9.67% | 6.83% | 0.00% | 0.00% | 0.00% | 2.61% | 0.00% |
| | Viz. Only (No Model) | 1.33% | 1.33% | 8.16% | 0.00% | 11.38% | 8.92% | 15.31% | 6.31% |
| | Viz. Meaningless (Model Fail) | 1.42% | 5.58% | 9.75% | 0.00% | 9.00% | 16.69% | 0.00% | 6.46% |
| **Reporting** | Clear Presentation | 5.15% | 17.92% | **34.46%** | 20.54% | 9.83% | 37.59% | **39.0%** | 20.92% |
| | No Report | 55.23% | 17.92% | 15.38% | 66.69% | 65.25% | 8.33% | 37.5% | 48.67% |
| | Lacks Conclusion/Summary | 32.00% | 51.31% | 28.31% | 8.92% | 14.00% | 30.67% | 7.00% | 2.75% |
| | Anomalous Numerical Results | 0.00% | 7.69% | 1.31% | 3.85% | 0.00% | 0.00% | 0.00% | 0.00% |
| | Too Simple (w/ Evidence) | 5.08% | 3.92% | 11.61% | 0.00% | 5.33% | 17.92% | 12.33% | 18.0% |
| | Poor Readability | 2.54% | 1.23% | 8.92% | 0.00% | 5.58% | 5.50% | 4.17% | 9.67% |

*Note:* Percentages reflect the distribution of outcomes for each method across task components. Evaluations are conducted independently by an expert panel of six PhD/MD students with diverse expertise (3 CS PhD students in AI for healthcare, 1 MD, 1 biomedical engineering, 1 biostatistics) to ensure clinical validity; their assessments are subsequently validated and consolidated. The base LLM for all LLM-based approaches (single and multi-agent) is `DeepSeek-V3-0324`.

> *[Task 4's Key Findings and Implications]* ① Multi-agent collaboration approaches (e.g., SmolAgent, OpenManus), often leveraging Python code execution, improve task completeness in complex workflows over single LLMs; ② Overall correctness for full automation remains low, indicating need for better agent capabilities; ③ Multi-agent framework choice is critical; single LLMs struggle with multi-step reasoning, state persistence, and effective tool use in these tasks.

# 5 Discussion

**When is multi-agent collaboration truly beneficial?** Our results suggest that the benefits of multi-agent collaboration are most apparent in tasks requiring decomposition of complex problems into manageable sub-tasks, explicit role assignment, iterative refinement, and the integration of external tools. This is evident in clinical workflow automation, where collaboration improves task completeness. This success is rooted in high task decomposability, where a complex analysis can be broken into logical sub-tasks (e.g., load data, run model, plot results) that align with agent specialization. Conversely, in tasks like medical VQA, which are often perception-bound, collaboration can fail due to an information fidelity bottleneck; critical visual details lost in the initial perception cannot be recovered through textual discussion. For tasks where a strong monolithic model can achieve high performance (e.g., textual QA with advanced LLMs) or where specialized architectures excel (e.g., EHR prediction with conventional models), the added complexity and computational cost of current multi-agent frameworks may not be justified. The "wisdom of the crowd" effect in multi-agent collaboration needs to overcome the inherent capabilities of the best single agents and also adapt to the specific nature of the data and task.

**Limitations.** Our study has several limitations. First, the performance of all LLM-driven methods, including multi-agent collaboration, is inherently tied to the capabilities of the underlying foundation models. Consequently, the comparative rankings presented here may shift as new and more powerful LLMs are developed. Second, while MedAgentBoard introduces four diverse task categories, its scope is not exhaustive of the full spectrum of real-world clinical challenges. For example, our clinical workflow automation evaluation is based on a curated set of 100 tasks; while substantial, this set does not fully capture the complexity and dynamism of all potential clinical data analysis scenarios. Finally, our evaluation metrics, while comprehensive, could be further enriched by incorporating deeper qualitative human assessments to more thoroughly gauge aspects such as clinical utility, trustworthiness, and the subtle dynamics of agent collaboration.

**Future work.** Building on our findings, future research should focus on several key areas. A critical direction is the development of multi-agent strategies specifically designed for diverse

medical data modalities, moving beyond text-centric collaboration. Exploring hybrid architectures that synergize the feature extraction strengths of conventional models with the complex reasoning capabilities of LLM-based agents could unlock significant performance gains, particularly in tasks like EHR prediction. Furthermore, as these systems approach clinical viability, investigating their robustness, interpretability, and ethical dimensions—including fairness, bias, and privacy—will be paramount. Finally, to address the limitations in scope of the current benchmark, advancing the field will necessitate the design of more challenging, dynamic, and diverse tasks that better reflect the complexities of real-world clinical environments and truly require sophisticated collaborative problem-solving.

**Broader impact.** The insights from MedAgentBoard carry significant broader implications for the development and deployment of AI in medicine. By providing a nuanced, evidence-based comparison, our work encourages a shift away from technological hype toward a more pragmatic, task-oriented approach. This can guide researchers and healthcare organizations to invest resources more effectively, choosing specialized conventional models for their proven reliability in certain tasks while selectively applying multi-agent systems where their collaborative strengths offer a distinct advantage, such as in complex workflow automation. However, this research also highlights potential risks. An over-reliance on our findings could be misinterpreted as a general indictment of multi-agent systems, potentially stifling innovation. It is crucial to view these results as a snapshot in a rapidly evolving field. Moreover, the increasing sophistication of any AI system intended for clinical use underscores the urgent need to address critical ethical considerations. Issues of accountability when an autonomous system errs, the potential for algorithmic bias to perpetuate health disparities, and ensuring patient data privacy remain central challenges that must be proactively managed as these technologies mature.

# 6 Conclusion

This paper introduces MedAgentBoard, a comprehensive benchmark for evaluating multi-agent collaboration, single LLMs, and conventional methods across a diverse set of medical tasks and data modalities. Our findings underscore that while multi-agent collaboration shows promise in specific complex scenarios like workflow automation, it does not universally outperform advanced single LLMs or, critically, specialized conventional methods which remain superior in tasks like medical VQA and EHR prediction. MedAgentBoard provides a valuable resource for the community and offers actionable insights to guide the selection and development of AI methods, emphasizing that the path to practical medical AI involves a nuanced understanding of the strengths and weaknesses of each approach relative to the specific clinical challenge at hand.

# Acknowledgement

This work was supported by National Natural Science Foundation of China (62402017), Research Grants Council of Hong Kong (27206123, 17200125, C5055-24G, and T45-401/22-N), and the Hong Kong Innovation and Technology Fund (GHP/318/22GD), Beijing Traditional Chinese Medicine Science and Technology Development Fund (BJZYZD-2025-13), Peking University Clinical Medicine Plus X (Young Scholars Project-the Fundamental Research Funds for the Central Universities PKU2025PKULCXQ024; Pilot Program-Key Technologies Project 2024YXXLHGG007), and Peking University "TengYun" Clinical Research Program (TY2025015). Liantao Ma was supported by Beijing Natural Science Foundation (L244063, L244025), Beijing Municipal Health Commission Research Ward Excellence Clinical Research Program (BRWEP2024W032150205), and Xuzhou Scientific Technological Projects (KC23143). Junyi Gao acknowledged the receipt of studentship awards from the Health Data Research UK-The Alan Turing Institute Wellcome PhD Programme in Health Data Science (grant 218529/Z/19/Z) and Baidu Scholarship.

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

# Appendix

## A   Data Privacy and Code Availability Statement

To ensure the fairness and reproducibility of this research, no new patient data was collected. All datasets employed in this paper are publicly available or accessible upon request and were used under their respective licenses.

The TJH EHR dataset [66] utilized in this study is publicly available on GitHub (https://github.com/HAIRLAB/Pre_Surv_COVID_19). The MIMIC-IV dataset (structured EHR data, version 3.1) [83] is open to researchers and can be accessed on request via PhysioNet (https://physionet.org/content/mimiciv/3.1/).

For Task 1 (Medical (Visual) Question Answering, Section 3.1), we used:

- MedQA [31], featuring USMLE-style questions.
- PubMedQA [32], comprising questions based on biomedical abstracts.
- PathVQA [33], focusing on pathology images.
- VQA-RAD [34], derived from clinical radiology images.

These are established public benchmarks and were used as described in Appendix B.1.

For Task 2 (Lay Summary Generation, Section 3.2), we leveraged:

- Cochrane [50], providing plain language summaries of systematic reviews.
- eLife [51] and PLOS [51], containing author-written summaries of research articles.
- Med-EASi [52], which focuses on fine-grained simplification annotations.
- PLABA [53], a dataset of plain language adaptations of biomedical abstracts.

These datasets are publicly available and were used as detailed in Appendix B.2.

The TJH and MIMIC-IV datasets were also used for Task 3 (EHR Predictive Modeling, Section 3.3, Appendix B.3) and Task 4 (Clinical Workflow Automation, Section 3.4, Appendix B.4).

Throughout the experiments, we strictly adhered to all applicable data use agreements and ethical guidelines, reaffirming our commitment to responsible data handling and usage.

The performance of certain LLMs such as GPT-4o and GPT o3-mini-high was evaluated using the secure Azure OpenAI API. The performance of DeepSeek models (e.g., DeepSeek-V3, DeepSeek-R1) was obtained via DeepSeek's official APIs. Usage of these APIs complied with their respective terms of service, and human review of the data processed by these APIs was handled according to provider policies. All other models, including conventional machine learning (ML) models, deep learning (DL) models, and other LLMs (as detailed in Appendix B.3), were deployed and evaluated locally.

All code developed for the MedAgentBoard benchmark, the curated benchmark tasks (including 100 analytical questions for Task 4), detailed prompts used for LLM-based methods, and experimental results are open-sourced. They can be accessed online at the project website: https://medagentboard.netlify.app/. This website also serves as a platform for up-to-date benchmark results and further resources.

## B   Implementation Details

### B.1   Implementation Details in Task 1

**Datasets.** Four datasets are utilized for this task, two for medical question answering (MedQA, PubMedQA) and two for medical visual question answering (PathVQA, VQA-RAD). Further details are provided below:

(1) **MedQA** MedQA consists of questions from professional medical board examinations in the US, Mainland China, and Taiwan. Our study employs the English-language, five-option multiple-choice questions derived from the United States Medical Licensing Examination (USMLE).

(2) **PubMedQA** PubMedQA is a biomedical research question answering dataset. Each instance includes a question, a PubMed abstract (excluding its conclusion), and an answer. Answers are provided in two formats: a closed-ended label ("Yes/No/Maybe") and a free-form "long answer", which is the original abstract's conclusion. We utilize PubMedQA for both closed-ended QA ("Yes/No/Maybe" labels) and open-ended free-form QA (long answers).

(3) **PathVQA** PathVQA is a visual question answering (VQA) dataset centered on pathology images, featuring both open-ended and binary (yes/no) questions. Our study exclusively uses the binary "yes/no" questions.

(4) **VQA-RAD** VQA-RAD is a VQA dataset derived from clinical radiology images, containing open-ended and closed-ended questions. We utilize its binary "yes/no" questions (as multiple-choice) and its open-ended free-form questions.

Details on the splits of these datasets can be found in Table 7.

Table 7: *Dataset splits for medical QA and VQA tasks.* MC denotes multiple-choice; FF denotes free-form settings.

| Dataset | Train | Valid | Test | Sampled Test |
|---------|-------|-------|------|--------------|
| MedQA | 10,178 | 1,272 | 1,273 | 200 (MC) |
| PubMedQA | 500 | 500 | 500 | 200 (MC/FF) |
| PathVQA | 19,755 | 6,279 | 6,761 | 200 (MC) |
| VQA-RAD | 1,793 | 451 | 451 | 200 (MC), 179 (FF) |

To balance the computational costs (time and financial resources) associated with LLM inference and to ensure robust conclusions, MedAgentBoard selects approximately 200 samples per dataset for testing. This sample size, larger than those used in some prior works (e.g., MDAgents [12], which utilized 50 samples per dataset), aims for enhanced statistical reliability of our findings.

**Model training, methods details, and hyperparameters.** Training for conventional models adheres to the original implementation code provided in their respective GitHub repositories. A notable exception is Gatortron, for which we utilize pre-trained weights from HuggingFace and subsequently fine-tune it with an MLP classification head. The repository links are:

(1) **BioLinkBERT:** https://github.com/michiyasunaga/LinkBERT

(2) **Gatortron:** https://huggingface.co/UFNLP/gatortron-base

(3) **M³AE:** https://github.com/zhjohnchan/M3AE

(4) **BiomedGPT:** https://github.com/taokz/BiomedGPT

(5) **MUMC:** https://github.com/pengfeiliHEU/MUMC

(6) **LLaVA-Med:** https://github.com/microsoft/LLaVA-Med

(7) **Med-Flamingo:** https://github.com/snap-stanford/med-flamingo

Details on the fine-tuning configurations for each conventional model are presented in Table 8. LLaVA-Med and Med-Flamingo are evaluated directly on the test sets in a zero-shot manner. As these models have already been extensively trained on large-scale medical VQA datasets, this approach allows us to assess their generalized, out-of-the-box performance on our benchmarks.

Table 8: *Fine-tuning configuration for conventional models in medical QA and VQA.*

| Model | Training Epochs | Batch Size | Learning Rate |
|-------|-----------------|------------|---------------|
| BioLinkBERT | 20 | 32 | 3e-5 |
| Gatortron | 10 | 16 | 3e-6 |
| M³AE | 50 | 64 | 5e-6 |
| BiomedGPT | 20 | 16 | 5e-5 |
| MUMC | 10 | 2 | 5e-5 |

We evaluate single LLMs using a spectrum of prompting techniques, implemented as detailed in our publicly available codebase (within the project's code repository for the specific implementation). These strategies include:

- **Zero-shot Prompting:** The LLM receives the question (and options, for multiple-choice tasks) without any examples. For multiple-choice questions, it is instructed to return only the option letter (e.g., A, B, C); for free-form questions, it is asked for a concise answer.
- **Few-shot Prompting (In-Context Learning - ICL):** The prompt includes a few examples of question-answer pairs relevant to the specific dataset and task type (multiple-choice or free-form) before presenting the actual question to the LLM.
- **Chain-of-Thought (CoT) Prompting:** The LLM is instructed to "work through this step-by-step" and provide its reasoning process before the final answer. Responses are requested in a JSON format, encapsulating both the "Thought" (detailing the reasoning steps) and the "Answer" (the final derived answer).
- **Self-Consistency (SC):** Multiple responses (typically 5) are generated using zero-shot prompting. The final answer is determined by a majority vote among these independent responses.
- **CoT with Self-Consistency (CoT-SC):** This method combines CoT prompting with self-consistency. Multiple responses are generated using CoT prompting (each expected to include "Thought" and "Answer"). The final answer is then determined by a majority vote on the "Answer" fields extracted from these structured responses.

For visual question answering (VQA) tasks, the image is encoded (e.g., base64) and included in the prompt alongside the textual question, adhering to standard multimodal input formats. The system message primes the LLM for either general medical QA (e.g., "You are a medical expert answering medical questions with precise and accurate information.") or medical VQA (e.g., "You are a medical vision expert analyzing medical images and answering questions about them."), depending on the task's nature.

For evaluating multi-agent collaboration, we adapt the official GitHub implementations of the selected frameworks. These are integrated into our unified MedAgentBoard evaluation pipeline to ensure consistent experimental conditions and fair comparisons across all approaches. The specific frameworks and their source repositories are:

(1) **MedAgents:** https://github.com/gersteinlab/MedAgents
(2) **ReConcile:** https://github.com/dinobby/ReConcile
(3) **MDAgents:** https://github.com/mitmedialab/MDAgents
(4) **ColaCare:** https://github.com/PKU-AICare/ColaCare

Our implemented code for multi-agent collaboration specific to this task can be found at: https://github.com/yhzhu99/MedAgentBoard/tree/main/medagentboard/medqa

**Hardware and software configuration.** All training of conventional models and relevant experiments are conducted on a system equipped with four NVIDIA RTX 3090 GPUs, each possessing 24GB of VRAM. CUDA driver version 12.4 is consistently utilized. Versions of Python, PyTorch, and other auxiliary packages are maintained as per the specific requirements detailed in the original implementations of each conventional model.

All LLM-based evaluations utilize the Qwen series models (via Alibaba Cloud Model Studio) and DeepSeek-V3 (version DeepSeek-V3-0324, via the official DeepSeek API). These LLM-related experiments are conducted between April 1, 2025, and April 15, 2025.

### B.2 Implementation Details in Task 2

**Datasets.** Five datasets are used for this task:

(1) **Cochrane:** The Cochrane Database of Systematic Reviews (CDSR) is a resource aggregating systematic reviews in healthcare. It comprises pairs of technical abstracts and corresponding plain-language summaries, covering various healthcare domains. These summaries are written by the review authors themselves.
(2) **eLife:** As part of the larger CELLS dataset, the eLife dataset focuses on lay language summarization within the biomedical and life sciences domain. It consists of pairs of full scientific articles sourced from the eLife journal and expert-written lay summaries (called "digests"). Compared to other datasets like PLOS, eLife lay summaries are approximately twice as long and are written by expert editors, resulting in greater readability and abstractiveness.

(3) **PLOS:** We employ the Genetics subset of the PLOS dataset, another component of the CELLS resource, which provides data for lay language summarization from the biomedical domain, covering journals like PLOS Genetics, PLOS Biology, etc. It contains full biomedical articles from the PLOS Genetics journal with their author-written lay summaries.

(4) **Med-EASi:** Med-EASi is a uniquely crowdsourced and finely annotated dataset for the controllable simplification of short medical texts. It is built upon existing parallel corpora like SIMPWIKI and MSD.

(5) **PLABA:** The Plain Language Adaptation of Biomedical Abstracts (PLABA) dataset is designed for the task of plain language adaptation of biomedical text. It features pairs of PubMed abstracts and manually created, sentence-aligned adaptations and is sourced from PubMed abstracts relevant to popular MedlinePlus user questions (75 topics, 10 abstracts each).

Details on the split of the datasets can be found in Table 9.

Table 9: *Details about the splits of the lay summary datasets.*

| Dataset | Train | Validation | Test | Sampled Test |
|---------|-------|-----------|------|--------------|
| Cochrane | 3,568 | 411 | 480 | 100 |
| eLife | 4,346 | 241 | 142 | 100 |
| PLOS | 3,600 | 400 | 300 | 100 |
| Med-EASi | 1,399 | 196 | 300 | 100 |
| PLABA | 745 | 83 | 155 | 100 |

**Model training, methods details, and hyperparameters.** For the model training of conventional models, we follow the baseline implementation from the codebase of PLABA (`https://github.com/attal-kush/PLABA/blob/main/BaselineModelReports.py`), where pre-trained model weights are obtained from corresponding HuggingFace model cards:

(1) **BART:** `https://huggingface.co/facebook/bart-base`

(2) **T5:** `https://huggingface.co/google-t5/t5-base`

(3) **BART-CNN:** `https://huggingface.co/facebook/bart-large-cnn`

(4) **PEGASUS:** `https://huggingface.co/google/pegasus-large`

Details on the fine-tuning configuration for each model can be found in Table 10.

Table 10: *Fine-tuning configuration for conventional models in the lay summary generation task.*

| Model | Training Epochs | Batch Size | Learning Rate |
|-------|-----------------|-----------|---------------|
| BART | 10 | 2 | 5e-5 |
| T5 | 10 | 2 | 5e-5 |
| BART-CNN | 10 | 2 | 5e-5 |
| PEGASUS | 10 | 2 | 5e-5 |

The system message primes the LLM as an expert medical writer specializing in creating accessible lay summaries.

We evaluate single LLMs using a spectrum of prompting techniques. These strategies include:

• **Basic Prompting:** The LLM receives the medical text with a direct instruction to generate a single-paragraph lay summary in simple terms for a general audience.

• **Optimized Prompting:** The LLM is provided with the medical text and a more detailed prompt. This prompt includes specific guidelines for the lay summary, such as using plain language, avoiding jargon, explaining complex concepts simply, aiming for an 8th-grade reading level, using active voice, presenting findings truthfully, providing context, maintaining accuracy, and structuring the output as a single coherent paragraph.

• **Optimized Prompting with In-Context Learning (Optimized+ICL):** This approach augments the optimized prompt with two examples of medical text and their corresponding lay summaries, specific to the dataset being processed. These examples demonstrate the desired style and format before the LLM is asked to summarize the target medical text.

For multi-agent collaboration, we adapt principles from AgentSimp [59], a general text simplification framework, for the lay summary generation task. Our adapted framework defines nine distinct agent roles, each with specialized LLM-driven capabilities: Project Director, Structure Analyst, Content Simplifier, Simplify Supervisor, Metaphor Analyst, Terminology Interpreter, Content Integrator, Article Architect, and Proofreader. For this task, we implement a specific sequential pipeline orchestrating seven of these agents to transform complex medical text into an accessible single-paragraph summary:

(1) **Project Director Agent:** Analyzes the input medical text and establishes overarching simplification guidelines (e.g., target audience, key concepts, desired reading level).

(2) **Structure Analyst Agent:** Extracts crucial information, main conclusions, and essential structural elements from the medical text that must be conveyed.

(3) **Content Simplifier Agent:** Generates an initial draft of the single-paragraph lay summary based on the original text, guided by the director's guidelines and analyst's key information.

(4) **Simplify Supervisor Agent:** Critically reviews the initial draft for accuracy and clarity against the guidelines, providing feedback and a revised version.

(5) **Metaphor Analyst Agent:** Enhances the summary's accessibility by identifying complex medical concepts in the supervised draft and integrating illustrative metaphors or analogies.

(6) **Terminology Interpreter Agent:** Focuses on medical jargon in the metaphor-enhanced summary, ensuring technical terms are either replaced with simpler alternatives or clearly explained in plain language.

(7) **Proofreader Agent:** Conducts a final quality assurance check on the refined summary, correcting any remaining errors and ensuring overall coherence and adherence to the single-paragraph constraint.

This multi-step, role-based process allows for a comprehensive approach to simplification, from high-level planning to detailed textual refinement. Each agent utilizes the DeepSeek-V3 model.

**Hardware and software configuration.**   All training and experiments are run on four NVIDIA RTX 3090 GPUs, each with 24GB of VRAM. The software environment comprises CUDA driver version 12.4, Python 3.13, PyTorch 2.6.0, PyTorch Lightning 2.5.1, and Transformers 4.51.3.

All LLM-based evaluations utilize the DeepSeek-V3 (version `DeepSeek-V3-0324`, via the official DeepSeek API). These LLM-related experiments are conducted between May 1, 2025, and May 5, 2025.

### B.3  Implementation Details in Task 3

**Datasets.**   This task employs two datasets:

(1) **TJH Dataset** [84]: Derived from Tongji Hospital of Tongji Medical College, the TJH dataset consists of 485 anonymized COVID-19 inpatients treated in Wuhan, China, from January 10 to February 24, 2020. It includes 73 lab test features and 2 demographic features. The dataset is publicly available on GitHub (https://github.com/HAIRLAB/Pre_Surv_COVID_19).

(2) **MIMIC-IV Dataset** [85]: Sourced from the EHRs of the Beth Israel Deaconess Medical Center, the MIMIC dataset is extensive and widely used in healthcare research, particularly for simulating ICU scenarios. Specifically, this study utilizes version 3.1 of its structured EHR data [83] (https://physionet.org/content/mimiciv/3.1/), from which 17 lab test features and 2 demographic features are extracted. To minimize missing data, data segments from the same ICU stay are first consolidated daily. For patients with hospital stays exceeding seven days, records from the final seven days are retained, while earlier records are aggregated.

Details on dataset splits are in Table 11.

**Model training, methods details, and hyperparameters.**   For conventional deep learning-based EHR prediction models (GRU, LSTM, AdaCare, ConCare, GRASP), the AdamW optimizer [86] is employed, and training proceeds for a maximum of 50 epochs on the designated training set. To mitigate overfitting, an early stopping strategy is implemented with a patience of 5 epochs, monitored by the AUROC metric. The learning rate is selected via grid search from the set $\{1 \times 10^{-2}, 1 \times$

Table 11: *Details about the splits of the TJH and MIMIC-IV datasets.* "Re." stands for Readmission, indicating patients who are readmitted to the ICU within 30 days of discharge, while "No Re." represents patients who are not readmitted.

| Dataset | TJH | | | MIMIC-IV | | | | |
|---|---|---|---|---|---|---|---|---|
| | Total | Alive | Dead | Total | Alive | Dead | Re. | No Re. |
| *Test Set Statistics* | | | | | | | | |
| # Patients | 200 | 109 | 91 | 200 | 183 | 17 | 53 | 147 |
| # Total visits | 967 | 601 | 366 | 801 | 717 | 84 | 274 | 527 |
| # Avg. visits | 4.8 | 5.5 | 4.0 | 4.0 | 3.9 | 4.9 | 5.2 | 3.6 |
| *Training Set Statistics* | | | | | | | | |
| # Patients | 140 | 75 | 65 | 8750 | 8028 | 722 | 2112 | 6638 |
| # Total visits | 641 | 395 | 246 | 33423 | 30117 | 3306 | 10448 | 22975 |
| # Avg. visits | 4.6 | 5.3 | 3.8 | 3.8 | 3.8 | 4.6 | 4.9 | 3.5 |
| *Validation Set Statistics* | | | | | | | | |
| # Patients | 21 | 11 | 10 | 1250 | 1147 | 103 | 305 | 945 |
| # Total visits | 96 | 54 | 42 | 4685 | 4176 | 509 | 1522 | 3163 |
| # Avg. visits | 4.6 | 4.9 | 4.2 | 3.7 | 3.6 | 4.9 | 5.0 | 3.3 |

$10^{-3}, 1 \times 10^{-4}\}$. These models utilize a hidden dimension of 128 and a batch size of 256. For machine learning models (Decision Tree and XGBoost), as they do not directly support longitudinal EHR data, the last visit of a patient's record is selected, as it best reflects the patient's current health status.

Single LLMs are evaluated using a well-designed prompting template to effectively deliver structured EHR data. The prompting strategy employs a feature-wise list-style format for inputting EHR data and provides LLMs with feature units and reference ranges. Unit and reference ranges for each clinical feature are manually curated from medical guidelines. An in-context learning strategy is also available. The prompt templates for the prediction tasks with EHR data are shown in Appendix F.

For multi-agent collaboration approaches (e.g., MedAgents, ReConcile, ColaCare), this task is defined as a free-form QA task, adapting QA-like frameworks as illustrated in Task 1 for EHR prediction – where agents debate patient health status and risk factors from textualized data. MDAgents [12] is excluded because its emphasis on complex interaction modes and checks is less relevant here, potentially reducing its utility to that of simpler fixed-interaction agents.

**Hardware and software configuration.** The training of machine learning/deep learning models and the single LLM generation experiments are performed on a server equipped with 128GB of RAM and a single NVIDIA RTX 3090 GPU (CUDA 12.5). The primary software stack comprises Python 3.12, PyTorch 2.6.0, PyTorch Lightning 2.5.1, and Transformers 4.50.0. OpenAI's APIs, including the GPT-4o (`chatgpt-4o-latest`) and GPT o3-mini-high (`o3-mini-high`) models, and DeepSeek's official APIs, including the DeepSeek-V3 (`deepseek-v3-250324`) and DeepSeek-R1 (`deepseek-r1-250120`), are utilized. All other LLMs evaluated in this task are fetched from HuggingFace and deployed locally by LMStudio on a Mac Studio M2 Ultra with 192GB of RAM:

(1) **OpenBioLLM**: https://huggingface.co/aaditya/OpenBioLLM-Llama3-8B-GGUF

(2) **Gemma-3**: https://huggingface.co/lmstudio-community/gemma-3-4b-it-GGUF

(3) **Qwen2.5**: https://huggingface.co/lmstudio-community/Qwen2.5-7B-Instruct-1M-GGUF

(4) **HuatuoGPT-o1**: https://huggingface.co/QuantFactory/HuatuoGPT-o1-7B-GGUF

(5) **DeepSeek-R1-Distill-Qwen**: https://huggingface.co/lmstudio-community/DeepSeek-R1-Distill-Qwen-7B-GGUF

All these single LLM-related experiments are conducted between April 24, 2025, and May 3, 2025. Experiments with multi-agent collaboration approaches are conducted between April 26, 2025, and May 5, 2025.

### B.4  Implementation Details in Task 4

**Datasets.**  The datasets for question generation are identical to those in Task 3: the TJH dataset and the MIMIC-IV dataset.

For Task 4, we generate 100 analytical questions (50 per dataset) covering various clinical data analysis categories: data extraction and statistical analysis, predictive modeling, data visualization, and report generation. These questions simulate real-world analytical scenarios common in clinical research and practice. Further details on task construction and the generated tasks are available in Appendix G.1.

**Model training, methods details, and hyperparameters.**  During task generation, we use the `Gemini-2.5-Pro-Exp-03-25` LLM with its default parameter configuration. For validation across all four frameworks (Single LLM, SmolAgents, OpenManus, and Owl), we consistently use the `DeepSeek-V3-250324` LLM. Its temperature is fixed at 0.0, a setting recommended for tasks like "coding" or "math" to ensure deterministic and factual outputs, and its maximum token length is unrestricted.

**Hardware and software configuration.**  All experiments for this task are performed on a standard consumer-grade laptop, as the task primarily involves API calls rather than model training. The multi-agent frameworks use the following versions:

(1) **SmolAgents**: Version 1.14.0 (https://github.com/huggingface/smolagents)

(2) **OpenManus**: Version 0.3.0 (https://github.com/FoundationAgents/OpenManus)

(3) **Owl**: No explicit version number available; the implementation is based on the most recent major update from March 27, 2025 (https://github.com/camel-ai/owl)

The software environment comprises Python 3.10 with requisite libraries for HTTP requests, JSON parsing, and output formatting.

Experiments are conducted between April 10, 2025, and April 24, 2025.

## C  LLM-as-a-judge Details in Task 1's Free-form Medical QA/VQA

This section documents the prompt templates used for single LLM evaluations across different prompting strategies. For multi-agent collaboration frameworks, system prompts and task-specific instructions for individual agents are provided, with particular attention to the clinical workflow automation task design. Examples of few-shot demonstrations provided to models are included for reproducibility.

The LLM used for evaluation is specified, including prompts and scoring rubrics provided to the judge for assessing open-ended responses. Measures taken to ensure consistency and mitigate bias in LLM-based evaluation procedures are discussed.

### C.1  Prompt to Judge LLM for VQA-RAD Free-form Questions

Given that answers to VQA-RAD open-ended free-form questions are typically single words or concise key phrases (rather than full sentences), our evaluation instructs the model to assess whether its predicted answer is essentially equivalent to the ground truth answer (Evaluation Dimension: Binary Correctness).

> *LLM judge prompt for the VQA-RAD free-form VQA task.*
>
> ```
> **You are a Medical Expert specialized in questions associated with radiological images. Your task
> is to act as an impartial judge and evaluate the correctness of an AI model's response to a
> medical visual question.**
>
> **Inputs You Will Receive:**
>
> 1.  **Question:** The question asked, likely referring to an (unseen) medical image.
> 2.  **Ground Truth Answer:** The accepted correct answer based on the image and question.
> 3.  **Model's Answer:** The answer generated by the AI model you need to evaluate.
> ```

```
**Evaluation Dimension: Binary Correctness**

Assess whether the **Model's Answer** is essentially correct when compared to the **Ground Truth
Answer**, considering the **Question**.

**Criteria:**

*   **1:** The **Model's Answer** is essentially correct. It accurately answers the **Question**
and aligns with the core meaning of the **Ground Truth Answer**. Minor phrasing differences are
acceptable if the core meaning is preserved.
*   **0:** The **Model's Answer** is incorrect. It fails to answer the **Question** accurately, or
significantly contradicts the **Ground Truth Answer**.

**Output Requirement:**

**Output ONLY the single digit '1' (if correct) or '0' (if incorrect).** Do NOT provide any
justification, explanation, or any other text. Your entire response must be just the single digit
'1' or '0'.

**Evaluation Task:**

**Question:** {{QUESTION}}
**Ground Truth Answer:** {{GROUND_TRUTH}}
**Model's Response:** {{MODEL_ANSWER}}
```

## C.2 Prompt to Judge LLM for PubMedQA Free-form Questions

For PubMedQA, where answers are more free-form, the LLM judge is instructed to conduct a nuanced assessment of the model's response. This involves determining its degree of alignment with the ground truth answer—based on factual accuracy, completeness, and semantic similarity-and assigning a score from 1 to 10 (Evaluation Dimension: Correctness and Alignment with Ground Truth). To further improve the quality, consistency, and interpretability of these judgments, we apply Chain-of-Thought (CoT) prompting to the LLM judge. We scale the LLM-as-a-judge score by multiplying by 100, as shown in the experimental results table.

*LLM judge prompt for the PubMedQA free-form QA task.*

```
**You are a highly knowledgeable and critical Medical Expert. Your task is to act as an impartial
judge and rigorously evaluate the quality and correctness of an AI model's response to a medical
question. You will assess this *solely* by comparing the model's response to the provided Ground
Truth Answer, considering the original Question.**

**Inputs You Will Receive:**

1.  **Question:** The original question asked.
2.  **Ground Truth Answer:** The reference answer, considered correct and complete for the given
question. This is your primary standard for evaluation.
3.  **Model's Response:** The answer generated by the AI model you must evaluate.

**Evaluation Dimension: Correctness and Alignment with Ground Truth**

Assess the **Model's Response** based *only* on its factual accuracy, completeness, relevance, and
overall alignment compared to the **Ground Truth Answer**, considering the scope of the
**Question**.

*   **Factual Accuracy & Alignment:** Does the information presented in the **Model's Response**
accurately reflect the information in the **Ground Truth Answer**? Are the key facts, conclusions,
and nuances the same? Identify any contradictions, inaccuracies, or misrepresentations compared to
the ground truth.
*   **Completeness:** Does the **Model's Response** cover the essential information present in the
**Ground Truth Answer** needed to fully address the **Question**? Note significant omissions of
key details found in the ground truth.
*   **Relevance & Conciseness:** Is all information in the **Model's Response** relevant to
answering the **Question**, as exemplified by the **Ground Truth Answer**? Penalize irrelevant
information, excessive verbosity, or details not present in the ground truth that don't enhance
the answer's quality. **Focus on the accuracy and completeness relative to the ground truth, not
length.**
*   **Overall Semantic Equivalence:** Does the **Model's Response** convey the same meaning and
conclusion as the **Ground Truth Answer**, even if phrased differently?

**Scoring Guide (1-10 Scale):**
```

# D  Cost Analysis of Multi-Agent Collaboration in Task 1

To provide a more comprehensive comparison that addresses the practical overhead of different approaches, we conduct a cost analysis of the methods evaluated in Task 1. The analysis focuses on multi-agent collaboration frameworks compared against single-LLM prompting strategies. Cost is assessed using two key metrics: the average number of discussion rounds required per question (Table 12) and the estimated API cost for processing the selected test sets (Table 13). All cost estimations are based on the DeepSeek API pricing strategy.

The analysis of discussion rounds reveals that ReConcile exhibits the highest number of rounds, particularly in free-form tasks. This outcome is likely attributable to its use of diverse base models for each agent, which often leads to divergent opinions that require more rounds to resolve. In contrast, MDAgents demonstrates a significantly lower number of rounds. This efficiency stems from its difficulty-gating mechanism, which reduces communication overhead by defaulting to a single

Table 12: *Average number of discussion rounds per question across different multi-agent frameworks and datasets.*

| Framework | MedQA | PubMedQA | | PathVQA | VQA-RAD | |
| | Multiple Choice | Multiple Choice | Free-Form | Multiple Choice | Multiple Choice | Free-Form |
|---|---|---|---|---|---|---|
| ColaCare | 1.20 | 1.06 | 1.03 | 1.06 | 1.16 | 1.23 |
| MDAgents | 0.81 | 0.88 | 0.83 | 0.88 | 0.035 | 0.39 |
| MedAgents | 1.23 | 1.23 | 1.07 | 1.23 | 1.42 | 1.89 |
| ReConcile | 1.14 | 1.20 | 2.30 | 1.20 | 1.15 | 1.98 |

agent for simpler questions (where the discussion round count is zero), thereby lowering the overall average.

Table 13: *Estimated cost (USD) on the selected test set for Task 1, based on the DeepSeek API pricing strategy.*

| Method | MedQA | PubMedQA | | PathVQA | VQA-RAD | |
| | Multiple Choice | Multiple Choice | Free-Form | Multiple Choice | Multiple Choice | Free-Form |
|---|---|---|---|---|---|---|
| ColaCare | 2.64 | 4.05 | 5.99 | 1.47 | 1.19 | 1.51 |
| MDAgents | 2.13 | 4.15 | 4.85 | 0.79 | 0.27 | 0.72 |
| MedAgents | 2.71 | 4.32 | 5.52 | 2.34 | 1.54 | 2.23 |
| ReConcile | 3.05 | 5.29 | 9.32 | 1.89 | 1.85 | 2.78 |
| SingleLLM (Zero-shot) | 0.39 | 0.91 | 2.15 | 0.14 | 0.14 | 0.36 |
| SingleLLM (SC) | 0.75 | 1.02 | 8.56 | 0.21 | 0.20 | 1.41 |
| SingleLLM (CoT) | 1.84 | 2.81 | 3.28 | 0.55 | 0.51 | 0.52 |
| SingleLLM (CoT-SC) | 7.99 | 10.52 | 13.50 | 2.32 | 2.27 | 2.15 |

As shown in Table 13, the estimated API costs align with the findings on discussion rounds. Multi-agent frameworks are generally more expensive than simpler single-LLM approaches like zero-shot or CoT, with costs varying based on framework design and task complexity. ReConcile, being the most communication-heavy, also incurs the highest costs among multi-agent systems in several tasks. Notably, for the textual QA tasks (MedQA and PubMedQA), the single-LLM CoT-SC prompting strategy is the most expensive method overall. This high cost is a result of generating multiple, token-intensive chain-of-thought responses for self-consistency, and it surpasses even the most communication-intensive multi-agent frameworks. This finding highlights that complex single-LLM reasoning strategies can also incur substantial computational overhead, which must be weighed against their performance benefits.

# E    Additional Experiments of Different Prompting Strategies for Task 3

We conduct additional experiments on different prompting strategies for understanding structured EHRs. The results in Table 14 demonstrate the impact of prompt design on model performance. The "Opt.+ICL" setting is used for the main results in Table 5.

Table 14: *Additional experiments on different prompting strategies for task 3.*

| LLM | Prompting Strategy | MIMIC-IV Mortality | | MIMIC-IV Readmission | | TJH Mortality | |
| | | AUROC($\uparrow$) | AUPRC($\uparrow$) | AUROC($\uparrow$) | AUPRC($\uparrow$) | AUROC($\uparrow$) | AUPRC($\uparrow$) |
|---|---|---|---|---|---|---|---|
| | Basic | $78.07_{\pm6.13}$ | $76.86_{\pm4.71}$ | $66.70_{\pm4.76}$ | $34.02_{\pm5.89}$ | $89.59_{\pm1.93}$ | $85.06_{\pm3.01}$ |
| DeepSeek-V3 | Optimized | $79.78_{\pm4.60}$ | $43.20_{\pm9.95}$ | $65.05_{\pm4.39}$ | $31.83_{\pm5.21}$ | $88.56_{\pm2.00}$ | $81.13_{\pm3.74}$ |
| | Opt.+ICL | $76.86_{\pm4.71}$ | $33.47_{\pm9.58}$ | $62.68_{\pm4.49}$ | $30.91_{\pm5.30}$ | $89.67_{\pm1.90}$ | $82.93_{\pm3.58}$ |
| | Basic | $73.68_{\pm7.52}$ | $33.27_{\pm9.51}$ | $65.31_{\pm4.98}$ | $38.68_{\pm6.88}$ | $90.63_{\pm1.99}$ | $83.59_{\pm3.85}$ |
| DeepSeek-R1 | Optimized | $73.36_{\pm8.12}$ | $43.49_{\pm11.08}$ | $71.76_{\pm4.74}$ | $45.30_{\pm7.76}$ | $91.06_{\pm1.88}$ | $86.06_{\pm3.37}$ |
| | Opt.+ICL | $83.95_{\pm4.60}$ | $42.10_{\pm9.95}$ | $73.92_{\pm3.78}$ | $43.59_{\pm6.42}$ | $85.59_{\pm1.97}$ | $76.87_{\pm3.56}$ |

*Note:* **Basic**: Directly feeding EHR data values. **Optimized**: Additionally incorporating the unit and reference range of each feature for better LLM understanding. **Opt.+ICL**: Upon the optimized setting, additionally adding one in-context learning example.

# F    Prompt Details in Task 3's LLM Settings

This section provides details on the prompt templates used for LLM-based predictions with EHR data in Task 3. The following prompts include task descriptions and EHR data formatted for the LLM from both the MIMIC-IV and TJH datasets.

---

*Detailed task descriptions for mortality and readmission tasks.*

# (1) In-hospital mortality prediction: Your primary task is to assess the provided medical data and analyze the health records from ICU visits to determine the likelihood of the patient not surviving their hospital stay.

# (2) 30-day readmission prediction: Your primary task is to analyze the medical data to predict the probability of readmission within 30 days post-discharge. Include cases where a patient passes away within 30 days from the discharge date as readmissions.

---

*Basic setting prompt template of optimized with in-context for the mortality prediction task on the MIMIC-IV.*

You are an experienced critical care physician working in an Intensive Care Unit (ICU), skilled in interpreting complex longitudinal patient data and predicting clinical outcomes.
System Prompt: You are an experienced critical care physician working in an Intensive Care Unit (ICU), skilled in interpreting complex longitudinal patient data and predicting clinical outcomes.

User Prompt: I will provide you with longitudinal medical information for a patient. The data covers 3 visits that occurred at 2113-01-31, 2113-02-01, 2113-02-02.
Each clinical feature is presented as a list of values, corresponding to these visits. Missing values are represented as `NaN` for numerical values and "unknown" for categorical values. Note that units and reference ranges are provided alongside relevant features.

Patient Background:
- Sex: male
- Age: 50 years

Your Task:
Your primary task is to assess the provided medical data and analyze the health records from ICU visits to determine the likelihood of the patient not surviving their hospital stay.

Instructions & Output Format:
Please first perform a step-by-step analysis of the patient data, considering trends, abnormal values relative to reference ranges, and their clinical significance for survival. Then, provide a final assessment of the likelihood of not surviving the hospital stay.

Your final output must be a JSON object containing two keys:
1.   `"think"`: A string containing your detailed step-by-step clinical reasoning (under 500 words).
2.   `"answer"`: A floating-point number between 0 and 1 representing the predicted probability of mortality (higher value means higher likelihood of death).

Example Format: ```json { "think": "The patient presents with worsening X, stable Y, and improved Z. Factor A is a major risk indicator... Overall assessment suggests a high risk.", "answer": 0.85 }```

Handling Uncertainty:
In situations where the provided data is clearly insufficient or too ambiguous to make a reasonable prediction, respond with the exact phrase: `I do not know`.

Now, please analyze and predict for the following patient:

Clinical Features Over Time:
- Capillary refill rate: [0.0, 0.0, 0.0]
- Glasgow coma scale eye opening: [Spontaneously, Spontaneously, To Speech]
- Glasgow coma scale motor response: [Obeys Commands, Obeys Commands, Obeys Commands]
- Glasgow coma scale total: [0.0, 0.0, 0.0]
- Glasgow coma scale verbal response: [Oriented, Oriented, No Response]
- Diastolic blood pressure: [55.0, 74.0, 73.0]
- Fraction inspired oxygen: [80.0, 50.0, 70.0]
- Glucose: [119.0, 118.0, 127.0]
- Heart Rate: [86.0, 110.0, 118.0]
- Height: [157.0, NaN, NaN]
- Mean blood pressure: [67.0, 85.0, 102.0]
- Oxygen saturation: [96.0, 100.0, 100.0]
- Respiratory rate: [17.0, 26.0, 15.0]
- Systolic blood pressure: [105.0, 124.0, 156.0]

```
- Temperature: [37.89, 37.61, 37.17]
- Weight: [80.92, 80.92, 80.92]
- pH: [7.48, 7.47, 7.51]
```

*Optimized setting prompt template of optimized with in-context for the mortality prediction task on the MIMIC-IV.*

```
You are an experienced critical care physician working in an Intensive Care Unit (ICU), skilled in
interpreting complex longitudinal patient data and predicting clinical outcomes.

I will provide you with longitudinal medical information for a patient. The data covers 3 visits
that occurred at 2113-01-31, 2113-02-01, 2113-02-02.
Each clinical feature is presented as a list of values, corresponding to these visits. Missing
values are represented as `NaN` for numerical values and "unknown" for categorical values. Note
that units and reference ranges are provided alongside relevant features.

Patient Background:
- Sex: male
- Age: 50 years

Your Task:
Your primary task is to assess the provided medical data and analyze the health records from ICU
visits to determine the likelihood of the patient not surviving their hospital stay.

Instructions & Output Format:
Please first perform a step-by-step analysis of the patient data, considering trends, abnormal
values relative to reference ranges, and their clinical significance for survival. Then, provide a
final assessment of the likelihood of not surviving the hospital stay.

Your final output must be a JSON object containing two keys:
1.  `"think"`: A string containing your detailed step-by-step clinical reasoning (under 500
words).
2.  `"answer"`: A floating-point number between 0 and 1 representing the predicted probability of
mortality (higher value means higher likelihood of death).

Example Format: ```json { "think": "The patient presents with worsening X, stable Y, and improved
Z. Factor A is a major risk indicator... Overall assessment suggests a high risk.", "answer": 0.85
}```

Handling Uncertainty:
In situations where the provided data is clearly insufficient or too ambiguous to make a
reasonable prediction, respond with the exact phrase: `I do not know`.

Now, please analyze and predict for the following patient:

Clinical Features Over Time:
- Capillary refill rate (Unit: /. Reference range: /.): [0.0, 0.0, 0.0]
- Glasgow coma scale eye opening (Unit: /. Reference range: /.): [Spontaneously, Spontaneously, To
Speech]
- Glasgow coma scale motor response (Unit: /. Reference range: /.): [Obeys Commands, Obeys
Commands, Obeys Commands]
- Glasgow coma scale total (Unit: /. Reference range: /.): [0.0, 0.0, 0.0]
- Glasgow coma scale verbal response (Unit: /. Reference range: /.): [Oriented, Oriented, No
Response]
- Diastolic blood pressure (Unit: mmHg. Reference range: less than 80.): [55.0, 74.0, 73.0]
- Fraction inspired oxygen (Unit: /. Reference range: more than 21.): [80.0, 50.0, 70.0]
- Glucose (Unit: mg/dL. Reference range: 70 - 100.): [119.0, 118.0, 127.0]
- Heart Rate (Unit: bpm. Reference range: 60 - 100.): [86.0, 110.0, 118.0]
- Height (Unit: cm. Reference range: /.): [157.0, NaN, NaN]
- Mean blood pressure (Unit: mmHg. Reference range: less than 100.): [67.0, 85.0, 102.0]
- Oxygen saturation (Unit: %. Reference range: 95 - 100.): [96.0, 100.0, 100.0]
- Respiratory rate (Unit: breaths per minute. Reference range: 15 - 18.): [17.0, 26.0, 15.0]
- Systolic blood pressure (Unit: mmHg. Reference range: less than 120.): [105.0, 124.0, 156.0]
- Temperature (Unit: degrees Celsius. Reference range: 36.1 - 37.2.): [37.89, 37.61, 37.17]
- Weight (Unit: kg. Reference range: /.): [80.92, 80.92, 80.92]
- pH (Unit: /. Reference range: 7.35 - 7.45.): [7.48, 7.47, 7.51]
```

*Optimized setting with in-context learning prompt template of optimized with in-context for the mortality prediction task on the MIMIC-IV.*

```
You are an experienced critical care physician working in an Intensive Care Unit (ICU), skilled in
interpreting complex longitudinal patient data and predicting clinical outcomes.
```

I will provide you with longitudinal medical information for a patient. The data covers 3 visits that occurred at 2113-01-31, 2113-02-01, 2113-02-02.
Each clinical feature is presented as a list of values, corresponding to these visits. Missing values are represented as `NaN` for numerical values and "unknown" for categorical values. Note that units and reference ranges are provided alongside relevant features.

Patient Background:
- Sex: male
- Age: 50 years

Your Task:
Your primary task is to assess the provided medical data and analyze the health records from ICU visits to determine the likelihood of the patient not surviving their hospital stay.

Instructions & Output Format:
Please first perform a step-by-step analysis of the patient data, considering trends, abnormal values relative to reference ranges, and their clinical significance for survival. Then, provide a final assessment of the likelihood of not surviving the hospital stay.

Your final output must be a JSON object containing two keys:
1.  `"think"`: A string containing your detailed step-by-step clinical reasoning (under 500 words).
2.  `"answer"`: A floating-point number between 0 and 1 representing the predicted probability of mortality (higher value means higher likelihood of death).

Example Format: ```json { "think": "The patient presents with worsening X, stable Y, and improved Z. Factor A is a major risk indicator... Overall assessment suggests a high risk.", "answer": 0.85 }```

Handling Uncertainty:
In situations where the provided data is clearly insufficient or too ambiguous to make a reasonable prediction, respond with the exact phrase: `I do not know`.

Example:
Input information of a patient:
The patient is a female, aged 52 years.
The patient had 4 visits that occurred at 0, 1, 2, 3.
Details of the features for each visit are as follows:
- Capillary refill rate (Unit: /. Reference range: /.): ["unknown", "unknown", "unknown", "unknown"]
- Glascow coma scale eye opening (Unit: /. Reference range: /.): ["Spontaneously", "Spontaneously", "Spontaneously", "Spontaneously"]
- Glascow coma scale motor response (Unit: /. Reference range: /.): ["Obeys Commands", "Obeys Commands", "Obeys Commands", "Obeys Commands"]
...... (other features omitted for brevity)

Response:
```json { "think": "Patient is 52 years old. GCS components indicate full alertness and responsiveness (spontaneous eye opening, obeys commands) consistently across the recorded time points. While capillary refill is unknown, the neurological status appears stable and good. Assuming other vital signs and labs (not shown) are not critically deranged, the current data suggests a lower risk of mortality.", "answer": 0.3 } ```

Now, please analyze and predict for the following patient:

Clinical Features Over Time:
- Capillary refill rate (Unit: /. Reference range: /.): [0.0, 0.0, 0.0]
- Glascow coma scale eye opening (Unit: /. Reference range: /.): [Spontaneously, Spontaneously, To Speech]
- Glascow coma scale motor response (Unit: /. Reference range: /.): [Obeys Commands, Obeys Commands, Obeys Commands]
- Glascow coma scale total (Unit: /. Reference range: /.): [0.0, 0.0, 0.0]
- Glascow coma scale verbal response (Unit: /. Reference range: /.): [Oriented, Oriented, No Response]
- Diastolic blood pressure (Unit: mmHg. Reference range: less than 80.): [55.0, 74.0, 73.0]
- Fraction inspired oxygen (Unit: /. Reference range: more than 21.): [80.0, 50.0, 70.0]
- Glucose (Unit: mg/dL. Reference range: 70 - 100.): [119.0, 118.0, 127.0]
- Heart Rate (Unit: bpm. Reference range: 60 - 100.): [86.0, 110.0, 118.0]
- Height (Unit: cm. Reference range: /.): [157.0, NaN, NaN]
- Mean blood pressure (Unit: mmHg. Reference range: less than 100.): [67.0, 85.0, 102.0]
- Oxygen saturation (Unit: %. Reference range: 95 - 100.): [96.0, 100.0, 100.0]
- Respiratory rate (Unit: breaths per minute. Reference range: 15 - 18.): [17.0, 26.0, 15.0]
- Systolic blood pressure (Unit: mmHg. Reference range: less than 120.): [105.0, 124.0, 156.0]
- Temperature (Unit: degrees Celsius. Reference range: 36.1 - 37.2.): [37.89, 37.61, 37.17]
- Weight (Unit: kg. Reference range: /.): [80.92, 80.92, 80.92]
- pH (Unit: /. Reference range: 7.35 - 7.45.): [7.48, 7.47, 7.51]

You are an experienced doctor specializing in COVID-19 treatment, skilled in interpreting
longitudinal patient data and predicting clinical outcomes.

I will provide you with longitudinal medical information for a patient. The data covers 5
visits/time points that occurred at 2020-01-23, 2020-01-30, 2020-02-04, 2020-02-05, 2020-02-06.
Each clinical feature is presented as a list of values, corresponding to these time points.
Missing values are represented as `NaN` for numerical values and "unknown" for categorical values.
Note that units and reference ranges are provided alongside relevant features.

Patient Background:
- Sex: female
- Age: 70.0 years

Your Task:
Your primary task is to assess the provided medical data and analyze the health records from ICU
visits to determine the likelihood of the patient not surviving their hospital stay.

Instructions & Output Format:
Please first perform a step-by-step analysis of the patient data, considering trends, abnormal
values relative to reference ranges, and their clinical significance for survival. Then, provide a
final assessment of the likelihood of not surviving the hospital stay.

Your final output must be a JSON object containing two keys:
1.  `"think"`: A string containing your detailed step-by-step clinical reasoning (under 500
words).
2.  `"answer"`: A floating-point number between 0 and 1 representing the predicted probability of
mortality (higher value means higher likelihood of death).

Example Format: ```json { "think": "The patient presents with worsening X, stable Y, and improved
Z. Factor A is a major risk indicator... Overall assessment suggests a high risk.", "answer": 0.85
}```

Handling Uncertainty:
In situations where the provided data is clearly insufficient or too ambiguous to make a
reasonable prediction, respond with the exact phrase: `I do not know`

Now, please analyze and predict for the following patient:

Clinical Features Over Time:
- Hypersensitive cardiac troponinI: [NaN, NaN, NaN, NaN, NaN]
- hemoglobin: [109.0, 112.0, NaN, 126.0, NaN]
- Serum chloride: [99.1, 102.9, NaN, 102.2, NaN]
- Prothrombin time: [13.6, NaN, NaN, NaN, NaN]
- procalcitonin: [0.06, 0.06, NaN, NaN, NaN]
- eosinophils(%): [0.0, 0.2, NaN, 0.1, NaN]
- Interleukin 2 receptor: [591.0, NaN, NaN, NaN, NaN]
- Alkaline phosphatase: [47.0, 61.0, NaN, 69.0, NaN]
- albumin: [34.9, 34.0, NaN, 38.4, NaN]
- basophil(%): [0.0, 0.2, NaN, 0.1, NaN]
- Interleukin 10: [8.1, NaN, NaN, NaN, NaN]
- Total bilirubin: [7.5, 7.7, NaN, 12.6, NaN]
- Platelet count: [169.0, 275.0, NaN, 238.0, NaN]
- monocytes(%): [5.9, 6.4, NaN, 6.1, NaN]
- antithrombin: [84.0, NaN, NaN, NaN, NaN]
- Interleukin 8: [21.9, NaN, NaN, NaN, NaN]
- indirect bilirubin: [3.8, 3.7, NaN, 7.5, NaN]
- Red blood cell distribution width: [12.8, 12.6, NaN, NaN, NaN]
- neutrophils(%): [75.0, 60.1, NaN, 66.4, NaN]
- total protein: [68.3, 62.2, NaN, 68.2, NaN]
- Quantification of Treponema pallidum antibodies: [0.07, NaN, NaN, NaN, NaN]
- Prothrombin activity: [94.0, NaN, NaN, NaN, NaN]
- HBsAg: [0.01, NaN, NaN, NaN, NaN]
- mean corpuscular volume: [94.6, 93.6, NaN, 94.4, NaN]
- hematocrit: [33.2, 33.7, NaN, 35.7, NaN]
- White blood cell count: [3.72, 5.31, NaN, 7.68, NaN]
- Tumor necrosis factorα: [10.4, NaN, NaN, NaN, NaN]
- mean corpuscular hemoglobin concentration: [328.0, 332.0, NaN, 353.0, NaN]
- fibrinogen: [5.33, NaN, NaN, NaN, NaN]
- Interleukin 1β: [5.0, NaN, NaN, NaN, NaN]
- Urea: [2.9, 3.7, NaN, 4.22, NaN]
- lymphocyte count: [0.71, 1.76, NaN, 2.1, NaN]
- PH value: [NaN, NaN, NaN, NaN, NaN]
- Red blood cell count: [3.51, 3.6, NaN, 3.78, NaN]
- Eosinophil count: [0.0, 0.01, NaN, 0.01, NaN]
- Corrected calcium: [2.17, 2.25, NaN, 2.32, NaN]
- Serum potassium: [3.34, 3.34, NaN, 3.9, NaN]

```
- glucose: [9.02, 9.42, NaN, NaN, NaN]
- neutrophils count: [2.79, 3.19, NaN, 5.09, NaN]
- Direct bilirubin: [3.7, 4.0, NaN, 5.1, NaN]
- Mean platelet volume: [11.6, 10.7, NaN, 9.7, NaN]
- ferritin: [567.2, NaN, NaN, NaN, NaN]
- RBC distribution width SD: [44.4, 42.7, NaN, NaN, NaN]
- Thrombin time: [16.7, NaN, NaN, NaN, NaN]
- (%)lymphocyte: [19.1, 33.1, NaN, 27.3, NaN]
- HCV antibody quantification: [0.06, NaN, NaN, NaN, NaN]
- DD dimer: [0.98, NaN, NaN, NaN, NaN]
- Total cholesterol: [3.28, 3.49, NaN, 4.47, NaN]
- aspartate aminotransferase: [30.0, 30.0, NaN, 20.0, NaN]
- Uric acid: [151.0, 135.0, NaN, 221.1, NaN]
- HCO3: [22.3, 24.6, NaN, 25.3, NaN]
- calcium: [2.07, 2.13, NaN, 2.29, NaN]
- Aminoterminal brain natriuretic peptide precursor(NTproBNP): [NaN, NaN, NaN, NaN, NaN]
- Lactate dehydrogenase: [328.0, 269.0, NaN, 226.0, NaN]
- platelet large cell ratio: [37.8, 30.0, NaN, 21.4, NaN]
- Interleukin 6: [47.82, NaN, NaN, NaN, NaN]
- Fibrin degradation products: [4.0, NaN, NaN, NaN, NaN]
- monocytes count: [0.22, 0.34, NaN, 0.47, NaN]
- PLT distribution width: [13.9, 12.5, NaN, 10.1, NaN]
- globulin: [33.4, 28.2, NaN, 29.8, NaN]
- γglutamyl transpeptidase: [21.0, 54.0, NaN, 53.0, NaN]
- International standard ratio: [1.04, NaN, NaN, NaN, NaN]
- basophil count(#): [0.0, 0.01, NaN, 0.01, NaN]
- mean corpuscular hemoglobin: [31.1, 31.1, NaN, 33.3, NaN]
- Activation of partial thromboplastin time: [34.8, NaN, NaN, NaN, NaN]
- High sensitivity Creactive protein: [42.3, 3.6, NaN, NaN, NaN]
- HIV antibody quantification: [0.1, NaN, NaN, NaN, NaN]
- serum sodium: [135.7, 140.4, NaN, 143.2, NaN]
- thrombocytocrit: [0.2, 0.29, NaN, 0.23, NaN]
- ESR: [66.0, 29.0, NaN, NaN, NaN]
- glutamicpyruvic transaminase: [19.0, 84.0, NaN, 67.0, NaN]
- eGFR: [77.2, 90.0, NaN, 84.6, NaN]
- creatinine: [69.0, 58.0, NaN, 64.0, NaN]
```

## *Optimized setting prompt template for the mortality prediction task on the TJH.*

```
You are an experienced doctor specializing in COVID-19 treatment, skilled in interpreting
longitudinal patient data and predicting clinical outcomes.

I will provide you with longitudinal medical information for a patient. The data covers 5
visits/time points that occurred at 2020-01-23, 2020-01-30, 2020-02-04, 2020-02-05, 2020-02-06.
Each clinical feature is presented as a list of values, corresponding to these time points.
Missing values are represented as `NaN` for numerical values and "unknown" for categorical values.
Note that units and reference ranges are provided alongside relevant features.

Patient Background:
- Sex: female
- Age: 70.0 years

Your Task:
Your primary task is to assess the provided medical data and analyze the health records from ICU
visits to determine the likelihood of the patient not surviving their hospital stay.

Instructions & Output Format:
Please first perform a step-by-step analysis of the patient data, considering trends, abnormal
values relative to reference ranges, and their clinical significance for survival. Then, provide a
final assessment of the likelihood of not surviving the hospital stay.

Your final output must be a JSON object containing two keys:
1.  `"think"`: A string containing your detailed step-by-step clinical reasoning (under 500
words).
2.  `"answer"`: A floating-point number between 0 and 1 representing the predicted probability of
mortality (higher value means higher likelihood of death).

Example Format: ```json { "think": "The patient presents with worsening X, stable Y, and improved
Z. Factor A is a major risk indicator... Overall assessment suggests a high risk.", "answer": 0.85
}```

Handling Uncertainty:
In situations where the provided data is clearly insufficient or too ambiguous to make a
reasonable prediction, respond with the exact phrase: `I do not know`

Now, please analyze and predict for the following patient:
```

```
Clinical Features Over Time:
- Hypersensitive cardiac troponinI (Unit: ng/L. Reference range: less than 14.): [NaN, NaN, NaN,
NaN, NaN]
- hemoglobin (Unit: g/L. Reference range: 140 - 180 for men, 120 - 160 for women.): [109.0, 112.0,
NaN, 126.0, NaN]
- Serum chloride (Unit: mmol/L. Reference range: 96 - 106.): [99.1, 102.9, NaN, 102.2, NaN]
- Prothrombin time (Unit: seconds. Reference range: 13.1 - 14.125.): [13.6, NaN, NaN, NaN, NaN]
- procalcitonin (Unit: ng/mL. Reference range than 0.05.): [0.06, 0.06, NaN, NaN, NaN]
- eosinophils(%) (Unit: %. Reference range: 1 - 6.): [0.0, 0.2, NaN, 0.1, NaN]
- Interleukin 2 receptor (Unit: pg/mL. Reference range: less than 625.): [591.0, NaN, NaN, NaN,
NaN]
- Alkaline phosphatase (Unit: IU/L. Reference range: 44 - 147.): [47.0, 61.0, NaN, 69.0, NaN]
- albumin (Unit: g/dL. Reference range: 3.5 - 5.5.): [34.9, 34.0, NaN, 38.4, NaN]
- basophil(%) (Unit: %. Reference range: 0.5 - 1.): [0.0, 0.2, NaN, 0.1, NaN]
- Interleukin 10 (Unit: pg/mL. Reference range: less than 9.8.): [8.1, NaN, NaN, NaN, NaN]
- Total bilirubin (Unit: μmol/L. Reference range: 5.1 - 17.): [7.5, 7.7, NaN, 12.6, NaN]
- Platelet count (Unit: x 10^9/L. Reference range: 150 - 450.): [169.0, 275.0, NaN, 238.0, NaN]
- monocytes(%) (Unit: %. Reference range: 2 - 10.): [5.9, 6.4, NaN, 6.1, NaN]
- antithrombin (Unit: %. Reference range: 80 - 120.): [84.0, NaN, NaN, NaN, NaN]
- Interleukin 8 (Unit: pg/mL. Reference range: less than 62.): [21.9, NaN, NaN, NaN, NaN]
- indirect bilirubin (Unit: μmol/L. Reference range: 3.4 - 12.0.): [3.8, 3.7, NaN, 7.5, NaN]
- Red blood cell distribution width (Unit: %. Reference range: 11.5 - 14.5 for men, 12.2 - 16.1
for women.): [12.8, 12.6, NaN, NaN, NaN]
- neutrophils(%) (Unit: %. Reference range: 45 - 70.): [75.0, 60.1, NaN, 66.4, NaN]
- total protein (Unit: g/L. Reference range: 60 - 83.): [68.3, 62.2, NaN, 68.2, NaN]
- Quantification of Treponema pallidum antibodies (Unit: /. Reference range: less than 1.0.):
[0.07, NaN, NaN, NaN, NaN]
- Prothrombin activity (Unit: %. Reference range: 70 - 130.): [94.0, NaN, NaN, NaN, NaN]
- HBsAg (Unit: IU/mL. Reference range: 0.0 - 0.01.): [0.01, NaN, NaN, NaN, NaN]
- mean corpuscular volume (Unit: fL. Reference range: 80 - 100.): [94.6, 93.6, NaN, 94.4, NaN]
- hematocrit (Unit: %. Reference range: 40 - 54 for men, 36 - 48 for women.): [33.2, 33.7, NaN,
35.7, NaN]
- White blood cell count (Unit: x 10^9/L. Reference range: 4.5 - 11.0.): [3.72, 5.31, NaN, 7.68,
NaN]
- Tumor necrosis factorα (Unit: pg/mL. Reference range: less than 8.1.): [10.4, NaN, NaN, NaN,
NaN]
- mean corpuscular hemoglobin concentration (Unit: g/L. Reference range: 320 - 360.): [328.0,
332.0, NaN, 353.0, NaN]
- fibrinogen (Unit: g/L. Reference range: 2 - 4.): [5.33, NaN, NaN, NaN, NaN]
- Interleukin 1β (Unit: pg/mL. Reference range: less than 6.5.): [5.0, NaN, NaN, NaN, NaN]
- Urea (Unit: mmol/L. Reference range: 1.8 - 7.1.): [2.9, 3.7, NaN, 4.22, NaN]
- lymphocyte count (Unit: x 10^9/L. Reference range: 1.0 - 4.8.): [0.71, 1.76, NaN, 2.1, NaN]
- PH value (Unit: /. Reference range: 7.35 - 7.45.): [NaN, NaN, NaN, NaN, NaN]
- Red blood cell count (Unit: x 10^12/L. Reference range: 4.5 - 5.5 for men, 4.0 - 5.0 for
women.): [3.51, 3.6, NaN, 3.78, NaN]
- Eosinophil count (Unit: x 10^9/L. Reference range: 0.02 - 0.5.): [0.0, 0.01, NaN, 0.01, NaN]
- Corrected calcium (Unit: mmol/L. Reference range: 2.12 - 2.57.): [2.17, 2.25, NaN, 2.32, NaN]
- Serum potassium (Unit: mmol/L. Reference range: 3.5 - 5.0.): [3.34, 3.34, NaN, 3.9, NaN]
- glucose (Unit: mmol/L. Reference range: 3.9 - 5.6.): [9.02, 9.42, NaN, NaN, NaN]
- neutrophils count (Unit: x 10^9/L. Reference range: 2.0 - 8.0.): [2.79, 3.19, NaN, 5.09, NaN]
- Direct bilirubin (Unit: μmol/L. Reference range: 1.7 - 5.1.): [3.7, 4.0, NaN, 5.1, NaN]
- Mean platelet volume (Unit: fL. Reference range: 7.4 - 11.4.): [11.6, 10.7, NaN, 9.7, NaN]
- ferritin (Unit: ng/mL. Reference range: 24 - 336 for men, 11 - 307 for women.): [567.2, NaN,
NaN, NaN, NaN]
- RBC distribution width SD (Unit: fL. Reference range: 40.0 - 55.0.): [44.4, 42.7, NaN, NaN, NaN]
- Thrombin time (Unit: seconds. Reference range: 12 - 19.): [16.7, NaN, NaN, NaN, NaN]
- (%)lymphocyte (Unit: %. Reference range: 20 - 40.): [19.1, 33.1, NaN, 27.3, NaN]
- HCV antibody quantification (Unit: IU/mL. Reference range: 0.04 - 0.08.): [0.06, NaN, NaN, NaN,
NaN]
- DD dimer (Unit: mg/L. Reference range: 0 - 0.5.): [0.98, NaN, NaN, NaN, NaN]
- Total cholesterol (Unit: mmol/L. Reference range: less than 5.17.): [3.28, 3.49, NaN, 4.47, NaN]
- aspartate aminotransferase (Unit: U/L. Reference range: 8 - 33.): [30.0, 30.0, NaN, 20.0, NaN]
- Uric acid (Unit: μmol/L. Reference range: 240 - 510 for men, 160 - 430 for women.): [151.0,
135.0, NaN, 221.1, NaN]
- HCO3 (Unit: mmol/L. Reference range: 22 - 29.): [22.3, 24.6, NaN, 25.3, NaN]
- calcium (Unit: mmol/L. Reference range: 2.13 - 2.55.): [2.07, 2.13, NaN, 2.29, NaN]
- Aminoterminal brain natriuretic peptide precursor(NTproBNP) (Unit: pg/mL. Reference range: 0 -
125.): [NaN, NaN, NaN, NaN, NaN]
- Lactate dehydrogenase (Unit: U/L. Reference range: 140 - 280.): [328.0, 269.0, NaN, 226.0, NaN]
- platelet large cell ratio (Unit: %. Reference range: 15 - 35.): [37.8, 30.0, NaN, 21.4, NaN]
- Interleukin 6 (Unit: pg/mL. Reference range: 0 - 7.): [47.82, NaN, NaN, NaN, NaN]
- Fibrin degradation products (Unit: μg/mL. Reference range: 0 - 10.): [4.0, NaN, NaN, NaN, NaN]
- monocytes count (Unit: x 10^9/L. Reference range: 0.32 - 0.58.): [0.22, 0.34, NaN, 0.47, NaN]
- PLT distribution width (Unit: fL. Reference range: 9.2 - 16.7.): [13.9, 12.5, NaN, 10.1, NaN]
- globulin (Unit: g/L. Reference range: 23 - 35.): [33.4, 28.2, NaN, 29.8, NaN]
- γglutamyl transpeptidase (Unit: U/L. Reference range: 7 - 47 for men, 5 - 25 for women.): [21.0,
54.0, NaN, 53.0, NaN]
```

```
- International standard ratio (Unit: ratio. Reference range: 0.8 - 1.2.): [1.04, NaN, NaN, NaN,
NaN]
- basophil count(#) (Unit: x 10^9/L. Reference range: 0.01 - 0.02.): [0.0, 0.01, NaN, 0.01, NaN]
- mean corpuscular hemoglobin (Unit: pg. Reference range: 27 - 31.): [31.1, 31.1, NaN, 33.3, NaN]
- Activation of partial thromboplastin time (Unit: seconds. Reference range: 22 - 35.): [34.8,
NaN, NaN, NaN, NaN]
- High sensitivity Creactive protein (Unit: mg/L. Reference range: 3 - 10.): [42.3, 3.6, NaN, NaN,
NaN]
- HIV antibody quantification (Unit: IU/mL. Reference range: 0.08 - 0.11.): [0.1, NaN, NaN, NaN,
NaN]
- serum sodium (Unit: mmol/L. Reference range: 135 - 145.): [135.7, 140.4, NaN, 143.2, NaN]
- thrombocytocrit (Unit: %. Reference range: 0.22 - 0.24.): [0.2, 0.29, NaN, 0.23, NaN]
- ESR (Unit: mm/hr. Reference range: less than 15  for men, less than 20 for women.): [66.0, 29.0,
NaN, NaN, NaN]
- glutamicpyruvic transaminase (Unit: U/L. Reference range: 0 - 35.): [19.0, 84.0, NaN, 67.0, NaN]
- eGFR (Unit: mL/min/1.73m². Reference range: more than 90.): [77.2, 90.0, NaN, 84.6, NaN]
- creatinine (Unit: µmol/L. Reference range: 61.9 - 114.9 for men, 53 - 97.2 for women.): [69.0,
58.0, NaN, 64.0, NaN]
```

*Optimized-setting with in-context learning prompt template for the mortality prediction task on the TJH.*

```
You are an experienced doctor specializing in COVID-19 treatment, skilled in interpreting
longitudinal patient data and predicting clinical outcomes.

I will provide you with longitudinal medical information for a patient. The data covers 5
visits/time points that occurred at 2020-01-23, 2020-01-30, 2020-02-04, 2020-02-05, 2020-02-06.
Each clinical feature is presented as a list of values, corresponding to these time points.
Missing values are represented as `NaN` for numerical values and "unknown" for categorical values.
Note that units and reference ranges are provided alongside relevant features.

Patient Background:
- Sex: female
- Age: 70.0 years

Your Task:
Your primary task is to assess the provided medical data and analyze the health records from ICU
visits to determine the likelihood of the patient not surviving their hospital stay.

Instructions & Output Format:
Please first perform a step-by-step analysis of the patient data, considering trends, abnormal
values relative to reference ranges, and their clinical significance for survival. Then, provide a
final assessment of the likelihood of not surviving the hospital stay.

Your final output must be a JSON object containing two keys:
1.  `"think"`: A string containing your detailed step-by-step clinical reasoning (under 500
words).
2.  `"answer"`: A floating-point number between 0 and 1 representing the predicted probability of
mortality (higher value means higher likelihood of death).

Example Format: ```json { "think": "The patient presents with worsening X, stable Y, and improved
Z. Factor A is a major risk indicator... Overall assessment suggests a high risk.", "answer": 0.85
}```

Handling Uncertainty:
In situations where the provided data is clearly insufficient or too ambiguous to make a
reasonable prediction, respond with the exact phrase: `I do not know`

Example:
Input information of a patient:
The patient is a male, aged 52.0 years.
The patient had 5 visits that occurred at 2020-02-09, 2020-02-10, 2020-02-13, 2020-02-14,
2020-02-17.
Details of the features for each visit are as follows:
- Hypersensitive cardiac troponinI (Unit: ng/L. Reference range: less than 14.): [1.9, 1.9, 1.9,
1.9, 1.9]
- hemoglobin (Unit: g/L. Reference range: 140 - 180 for men, 120 - 160 for women.): [139.0, 139.0,
142.0, 142.0, 142.0]
- Serum chloride (Unit: mmol/L. Reference range: 96 - 106.): [103.7, 103.7, 104.2, 104.2, 104.2]
...... (other features omitted for brevity)

Response:
```

```json
{ "think": "The patient is a 52-year-old male. Key labs like Troponin I are consistently
normal and low. Hemoglobin is borderline low for a male but stable. Serum chloride is within
normal limits and stable. Assuming other unlisted vital signs and labs are also stable or within
normal limits, the overall picture suggests a relatively stable condition without indicators of
severe organ damage or rapid decline commonly associated with high mortality risk in this context.
The risk appears low.", "answer": 0.25 }
```

Now, please analyze and predict for the following patient:

Clinical Features Over Time:
- Hypersensitive cardiac troponinI (Unit: ng/L. Reference range: less than 14.): [NaN, NaN, NaN,
NaN, NaN]
- hemoglobin (Unit: g/L. Reference range: 140 - 180 for men, 120 - 160 for women.): [109.0, 112.0,
NaN, 126.0, NaN]
- Serum chloride (Unit: mmol/L. Reference range: 96 - 106.): [99.1, 102.9, NaN, 102.2, NaN]
- Prothrombin time (Unit: seconds. Reference range: 13.1 - 14.125.): [13.6, NaN, NaN, NaN, NaN]
- procalcitonin (Unit: ng/mL. Reference range: less than 0.05.): [0.06, 0.06, NaN, NaN, NaN]
- eosinophils(%) (Unit: %. Reference range: 1 - 6.): [0.0, 0.2, NaN, 0.1, NaN]
- Interleukin 2 receptor (Unit: pg/mL. Reference range: less than 625.): [591.0, NaN, NaN, NaN,
NaN]
- Alkaline phosphatase (Unit: IU/L. Reference range: 44 - 147.): [47.0, 61.0, NaN, 69.0, NaN]
- albumin (Unit: g/dL. Reference range: 3.5 - 5.5.): [34.9, 34.0, NaN, 38.4, NaN]
- basophil(%) (Unit: %. Reference range: 0.5 - 1.): [0.0, 0.2, NaN, 0.1, NaN]
- Interleukin 10 (Unit: pg/mL. Reference range: less than 9.8.): [8.1, NaN, NaN, NaN, NaN]
- Total bilirubin (Unit: µmol/L. Reference range: 5.1 - 17.): [7.5, 7.7, NaN, 12.6, NaN]
- Platelet count (Unit: x 10^9/L. Reference range: 150 - 450.): [169.0, 275.0, NaN, 238.0, NaN]
- monocytes(%) (Unit: %. Reference range: 2 - 10.): [5.9, 6.4, NaN, 6.1, NaN]
- antithrombin (Unit: %. Reference range: 80 - 120.): [84.0, NaN, NaN, NaN, NaN]
- Interleukin 8 (Unit: pg/mL. Reference range: less than 62.): [21.9, NaN, NaN, NaN, NaN]
- indirect bilirubin (Unit: µmol/L. Reference range: 3.4 - 12.0.): [3.8, 3.7, NaN, 7.5, NaN]
- Red blood cell distribution width (Unit: %. Reference range: 11.5 - 14.5 for men, 12.2 - 16.1
for women.): [12.8, 12.6, NaN, NaN, NaN]
- neutrophils(%) (Unit: %. Reference range: 45 - 70.): [75.0, 60.1, NaN, 66.4, NaN]
- total protein (Unit: g/L. Reference range: 60 - 83.): [68.3, 62.2, NaN, 68.2, NaN]
- Quantification of Treponema pallidum antibodies (Unit: /. Reference range: less than 1.0.):
[0.07, NaN, NaN, NaN, NaN]
- Prothrombin activity (Unit: %. Reference range: 70 - 130.): [94.0, NaN, NaN, NaN, NaN]
- HBsAg (Unit: IU/mL. Reference range: 0.0 - 0.01.): [0.01, NaN, NaN, NaN, NaN]
- mean corpuscular volume (Unit: fL. Reference range: 80 - 100.): [94.6, 93.6, NaN, 94.4, NaN]
- hematocrit (Unit: %. Reference range: 40 - 54 for men, 36 - 48 for women.): [33.2, 33.7, NaN,
35.7, NaN]
- White blood cell count (Unit: x 10^9/L. Reference range: 4.5 - 11.0.): [3.72, 5.31, NaN, 7.68,
NaN]
- Tumor necrosis factorα (Unit: pg/mL. Reference range: less than 8.1.): [10.4, NaN, NaN, NaN,
NaN]
- mean corpuscular hemoglobin concentration (Unit: g/L. Reference range: 320 - 360.): [328.0,
332.0, NaN, 353.0, NaN]
- fibrinogen (Unit: g/L. Reference range: 2 - 4.): [5.33, NaN, NaN, NaN, NaN]
- Interleukin 1β (Unit: pg/mL. Reference range: less than 6.5.): [5.0, NaN, NaN, NaN, NaN]
- Urea (Unit: mmol/L. Reference range: 1.8 - 7.1.): [2.9, 3.7, NaN, 4.22, NaN]
- lymphocyte count (Unit: x 10^9/L. Reference range: 1.0 - 4.8.): [0.71, 1.76, NaN, 2.1, NaN]
- PH value (Unit: /. Reference range: 7.35 - 7.45.): [NaN, NaN, NaN, NaN, NaN]
- Red blood cell count (Unit: x 10^12/L. Reference range: 4.5 - 5.5 for men, 4.0 - 5.0 for
women.): [3.51, 3.6, NaN, 3.78, NaN]
- Eosinophil count (Unit: x 10^9/L. Reference range: 0.02 - 0.5.): [0.0, 0.01, NaN, 0.01, NaN]
- Corrected calcium (Unit: mmol/L. Reference range: 2.12 - 2.57.): [2.17, 2.25, NaN, 2.32, NaN]
- Serum potassium (Unit: mmol/L. Reference range: 3.5 - 5.0.): [3.34, 3.34, NaN, 3.9, NaN]
- glucose (Unit: mmol/L. Reference range: 3.9 - 5.6.): [9.02, 9.42, NaN, NaN, NaN]
- neutrophils count (Unit: x 10^9/L. Reference range: 2.0 - 8.0.): [2.79, 3.19, NaN, 5.09, NaN]
- Direct bilirubin (Unit: µmol/L. Reference range: 1.7 - 5.1.): [3.7, 4.0, NaN, 5.1, NaN]
- Mean platelet volume (Unit: fL. Reference range: 7.4 - 11.4.): [11.6, 10.7, NaN, 9.7, NaN]
- ferritin (Unit: ng/mL. Reference range: 24 - 336 for men, 11 - 307 for women.): [567.2, NaN,
NaN, NaN, NaN]
- RBC distribution width SD (Unit: fL. Reference range: 40.0 - 55.0.): [44.4, 42.7, NaN, NaN, NaN]
- Thrombin time (Unit: seconds. Reference range: 12 - 19.): [16.7, NaN, NaN, NaN, NaN]
- (%)lymphocyte (Unit: %. Reference range: 20 - 40.): [19.1, 33.1, NaN, 27.3, NaN]
- HCV antibody quantification (Unit: IU/mL. Reference range: 0.04 - 0.08.): [0.06, NaN, NaN, NaN,
NaN]
- DD dimer (Unit: mg/L. Reference range: 0 - 0.5.): [0.98, NaN, NaN, NaN, NaN]
- Total cholesterol (Unit: mmol/L. Reference range: less than 5.17.): [3.28, 3.49, NaN, 4.47, NaN]
- aspartate aminotransferase (Unit: U/L. Reference range: 8 - 33.): [30.0, 30.0, NaN, 20.0, NaN]
- Uric acid (Unit: µmol/L. Reference range: 240 - 510 for men, 160 - 430 for women.): [151.0,
135.0, NaN, 221.1, NaN]
- HCO3 (Unit: mmol/L. Reference range: 22 - 29.): [22.3, 24.6, NaN, 25.3, NaN]
- calcium (Unit: mmol/L. Reference range: 2.13 - 2.55.): [2.07, 2.13, NaN, 2.29, NaN]
- Aminoterminal brain natriuretic peptide precursor(NTproBNP) (Unit: pg/mL. Reference range: 0 -
125.): [NaN, NaN, NaN, NaN, NaN]
- Lactate dehydrogenase (Unit: U/L. Reference range: 140 - 280.): [328.0, 269.0, NaN, 226.0, NaN]

```
- platelet large cell ratio (Unit: %. Reference range: 15 - 35.): [37.8, 30.0, NaN, 21.4, NaN]
- Interleukin 6 (Unit: pg/mL. Reference range: 0 - 7.): [47.82, NaN, NaN, NaN, NaN]
- Fibrin degradation products (Unit: µg/mL. Reference range: 0 - 10.): [4.0, NaN, NaN, NaN, NaN]
- monocytes count (Unit: x 10^9/L. Reference range: 0.32 - 0.58.): [0.22, 0.34, NaN, 0.47, NaN]
- PLT distribution width (Unit: fL. Reference range: 9.2 - 16.7.): [13.9, 12.5, NaN, 10.1, NaN]
- globulin (Unit: g/L. Reference range: 23 - 35.): [33.4, 28.2, NaN, 29.8, NaN]
- γglutamyl transpeptidase (Unit: U/L. Reference range: 7 - 47 for men, 5 - 25 for women.): [21.0,
54.0, NaN, 53.0, NaN]
- International standard ratio (Unit: ratio. Reference range: 0.8 - 1.2.): [1.04, NaN, NaN, NaN,
NaN]
- basophil count(#) (Unit: x 10^9/L. Reference range: 0.01 - 0.02.): [0.0, 0.01, NaN, 0.01, NaN]
- mean corpuscular hemoglobin (Unit: pg. Reference range: 27 - 31.): [31.1, 31.1, NaN, 33.3, NaN]
- Activation of partial thromboplastin time (Unit: seconds. Reference range: 22 - 35.): [34.8,
NaN, NaN, NaN, NaN]
- High sensitivity Creactive protein (Unit: mg/L. Reference range: 3 - 10.): [42.3, 3.6, NaN, NaN,
NaN]
- HIV antibody quantification (Unit: IU/mL. Reference range: 0.08 - 0.11.): [0.1, NaN, NaN, NaN,
NaN]
- serum sodium (Unit: mmol/L. Reference range: 135 - 145.): [135.7, 140.4, NaN, 143.2, NaN]
- thrombocytocrit (Unit: %. Reference range: 0.22 - 0.24.): [0.2, 0.29, NaN, 0.23, NaN]
- ESR (Unit: mm/hr. Reference range: less than 15  for men, less than 20 for women.): [66.0, 29.0,
NaN, NaN, NaN]
- glutamicpyruvic transaminase (Unit: U/L. Reference range: 0 - 35.): [19.0, 84.0, NaN, 67.0, NaN]
- eGFR (Unit: mL/min/1.73m². Reference range: more than 90.): [77.2, 90.0, NaN, 84.6, NaN]
- creatinine (Unit: µmol/L. Reference range: 61.9 - 114.9 for men, 53 - 97.2 for women.): [69.0,
58.0, NaN, 64.0, NaN]
```

# G   More Details in Task 4's Benchmark Design, Execution, and Evaluation

## G.1   Details in Constructing Task Instructions

Our task construction process for clinical workflow automation is designed to cover the full spectrum of analytical activities typically performed in clinical research and practice. We identify four primary analytical categories based on common clinical data analysis workflows: (1) data extraction and statistical analysis, (2) predictive modeling, (3) data visualization, and (4) report generation. To ensure comprehensive coverage, we further divide the data extraction and statistical analysis category into four sub-tasks: data wrangling, data querying, data statistics, and data preprocessing. Similarly, data visualization tasks are classified into two types: one focuses on direct visualization from processed data, and another involves visualization of modeling results or parameters.

We employ a structured prompt-engineering approach using a large language model (`Gemini-2.5-Pro-Exp-03-25`) to generate candidate task instructions. Below is the system prompt used for task generation:

> *System prompt for task construction.*
>
> ```
> You are a data scientist.
> ```

For each task category, we develop specialized prompts that incorporate dataset-specific information (schema details and sample data) from both MIMIC-IV and TJH datasets. This ensures that generated tasks are grounded in the actual data structures available for analysis. The specialized prompts for each analytical category are presented as follows:

> *Prompt for task construction in data wrangling.*
>
> ```
> You have been given a CSV format dataset, and you need to provide a task based on the the dataset.
>
> The dataset examples are as follows:
> {{DATA_EXAMPLES}}
>
> You need to provide the response strictly according to the following requirements:
>
> 1. The task requires testing the ability of data wrangling;
> 2. Please return your response in JSON format, similar to the following:
> ```json
> {
>     'task': '...',
> ```

```
}
```

The DATA_EXAMPLES variable is created by randomly sampling 3 PatientIDs from each dataset, then aggregating their complete EHR records (demographics and lab results) into the prompt text:

*An example of the DATA_EXAMPLES variable.*

```
[
    [PatientID:241.0, RecordTime:2020-02-11, AdmissionTime:2020-02-10, DischargeTime:2020-02-19,
    Outcome:1.0, LOS:8.0, Sex:0.0, Age:70.0, hemoglobin:137.0, eosinophils(\%):0.2,
    basophil(\%):0.1, Platelet count:186.0, monocytes(\%):3.8, Red blood cell distribution width
    :12.4, neutrophils(\%):89.2, mean corpuscular volume:94.5, hematocrit:41.2, White blood cell
    count:10.75, mean corpuscular hemoglobin concentration:333.0, lymphocyte count:0.72, PH
    value:7.49, Red blood cell count:4.36, Eosinophil count:0.02, neutrophils count:9.59, Mean
    platelet volume:10.4, RBC distribution width SD:42.0, (\%)lymphocyte:6.7, platelet large cell
    ratio :28.5, monocytes count:0.41, PLT distribution width:12.2, basophil count(#):0.01, mean
    corpuscular hemoglobin :31.4, thrombocytocrit:0.19, ],
    ...
]
```

After generating an initial pool of candidate tasks, we conduct a manual review process to select the final 100 tasks (50 per dataset) based on clinical relevance, technical feasibility, and diversity of analytical techniques required.

---

*Generated analytical tasks in task 4.*

```
# ID: 1
# Dataset: TJH
# Task type: Data
# Task: Calculate the average 'Hypersensitive c-reactive protein' value for all patients who have
'Outcome: 1.0', considering only their lab records taken within the first 3 days (inclusive) of
their 'AdmissionTime'. Please exclude records where 'Hypersensitive c-reactive protein' is
missing.
-------------------------------------------------------------------------
# ID: 2
# Dataset: TJH
# Task type: Data
# Task: Identify the unique PatientID for all patients who have at least one record where the
'Lactate dehydrogenase' value is greater than 1000.0, regardless of the 'RecordTime'.
-------------------------------------------------------------------------
# ID: 3
# Dataset: TJH
# Task type: Data
# Task: Count the number of unique patients who have at least one record where both
'Hypersensitive c-reactive protein' is greater than 100.0 and 'D-D dimer' is greater than 5.0.
Please exclude records where either 'Hypersensitive c-reactive protein' or 'D-D dimer' is missing.
-------------------------------------------------------------------------
# ID: 4
# Dataset: TJH
# Task type: Data
# Task: For each patient, identify the record with the earliest `RecordTime` (first record) and
the record with the latest `RecordTime` (last record). List the unique `PatientID`s for patients
where the 'White blood cell count' in their first record is strictly less than the 'White blood
cell count' in their last record. Please exclude patients who have only one record or where 'White
blood cell count' is missing in either their first or last record.
-------------------------------------------------------------------------
# ID: 5
# Dataset: TJH
# Task type: Visualization
# Task: Create a line plot showing the average temporal trend of 'Hypersensitive c-reactive
protein' (Hypersensitive c-reactive protein) over time relative to the AdmissionTime for patients
grouped by 'Outcome'. The x-axis should represent the number of days from AdmissionTime, and the
y-axis should represent the average Hypersensitive c-reactive protein value for each group.
Display separate lines for patients with 'Outcome: 0.0' and 'Outcome: 1.0'.
-------------------------------------------------------------------------
# ID: 6
# Dataset: TJH
# Task type: Visualization
# Task: Create box plots to visualize the distribution of 'LOS' (Length of Stay) for patients,
comparing the distributions for different 'Outcome' values (0.0 and 1.0). Label the x-axis as
'Outcome' and the y-axis as 'Length of Stay (days)'.
-------------------------------------------------------------------------
# ID: 7
# Dataset: TJH
# Task type: Visualization
# Task: Create violin plots to visualize the distribution of 'White blood cell count' for records
where 'RecordTime' is equal to 'AdmissionTime'. Compare the distributions for patients with
'Outcome: 0.0' and 'Outcome: 1.0'. Label the x-axis as 'Outcome' and the y-axis as 'White blood
cell count'.
-------------------------------------------------------------------------
# ID: 8
# Dataset: TJH
# Task type: Visualization
# Task: Create separate histograms to visualize the distribution of 'Age' for patients with
'Outcome: 0.0' and patients with 'Outcome: 1.0'. Plot these histograms side-by-side for easy
comparison. Label the x-axis as 'Age' and the y-axis as 'Frequency'.
-------------------------------------------------------------------------
# ID: 9
# Dataset: TJH
# Task type: Visualization
# Task: Create line plots showing the temporal trend of 'Hypersensitive c-reactive protein' over
time for each patient. Calculate the time relative to the 'AdmissionTime' (in days) for each
'RecordTime'. Plot 'Hypersensitive c-reactive protein' on the y-axis against the relative time
(days from AdmissionTime) on the x-axis. Each patient's data points should be connected by lines,
and the lines should be colored according to the patient's final 'Outcome' (0.0 or 1.0). Label the
x-axis as 'Days from Admission' and the y-axis as 'Hypersensitive c-reactive protein'.
-------------------------------------------------------------------------
```

```
# ID: 10
# Dataset: TJH
# Task type: Visualization
# Task: Create a scatter plot visualizing the relationship between 'Age' and 'LOS' (Length of
Stay) for each patient. Each point on the scatter plot should represent a unique patient. Color
the points based on the patient's 'Outcome' (0.0 or 1.0). Label the x-axis 'Age' and the y-axis
'Length of Stay (days)'. Include a legend to distinguish between the two outcome groups.
-----------------------------------------------------------------------------
# ID: 11
# Dataset: TJH
# Task type: Data
# Task: Calculate the average range of 'White blood cell count' across all patients. For each
patient, the range is defined as the difference between their maximum and minimum recorded 'White
blood cell count' values. Only consider patients who have at least two 'White blood cell count'
records to calculate a valid range.
-----------------------------------------------------------------------------
# ID: 12
# Dataset: TJH
# Task type: Data
# Task: For each unique patient ('PatientID') who has at least two recorded measurements of
'Hypersensitive c-reactive protein' with non-null values, calculate the difference between the
value recorded on their last 'RecordTime' and the value recorded on their first 'RecordTime'.
After calculating this change for all eligible patients, find the average of these changes.
-----------------------------------------------------------------------------
# ID: 13
# Dataset: TJH
# Task type: Data
# Task: For each unique patient, determine the number of distinct types of lab measurements
recorded for them across all their entries. Exclude the standard patient metadata fields
(PatientID, RecordTime, AdmissionTime, DischargeTime, Outcome, LOS, Sex, and Age) from this count.
Finally, calculate the average number of unique lab measurement types per patient across the
entire dataset.
-----------------------------------------------------------------------------
# ID: 14
# Dataset: TJH
# Task type: Data
# Task: Process the dataset to create a single summary row for each unique patient. For each
PatientID, identify all recorded lab test measurements across their multiple visits. Calculate the
mean value for each unique lab test parameter recorded for that patient. The final output should
be a dataset where each row corresponds to a unique PatientID and contains the patient's Sex, Age,
Outcome, Length of Stay (LOS), and the mean value of every lab test parameter that was recorded at
least once for that patient.
-----------------------------------------------------------------------------
# ID: 15
# Dataset: TJH
# Task type: Data
# Task: Process the dataset to extract the first and last recorded values for each lab test for
every patient. Group the records by `PatientID` and sort them chronologically by `RecordTime`. For
each unique `PatientID` and each unique lab test parameter, identify the value recorded at the
earliest `RecordTime` and the value recorded at the latest `RecordTime`. The final output should
be a structured dataset where each row represents a unique `PatientID`. Include the patient's
`Sex`, `Age`, `Outcome`, and `LOS`. For each unique lab test parameter found in the dataset,
create two new columns: one representing the value from the first record for that parameter for
that patient, and another representing the value from the last record. If a lab test is recorded
only once for a patient, its value should populate both the 'First' and 'Last' columns for that
parameter. If a lab test is never recorded for a patient, the corresponding 'First' and 'Last'
columns should reflect missing values.
-----------------------------------------------------------------------------
# ID: 16
# Dataset: TJH
# Task type: Data
# Task: Analyze temporal changes in lab test results for each patient. For each unique PatientID,
iterate through their records and for each specific lab test parameter recorded, identify its
value at the earliest RecordTime and its value at the latest RecordTime where that specific test
was recorded. Calculate the change in value for each lab test by subtracting the earliest recorded
value from the latest recorded value for that test (Latest Value - Earliest Value). This
calculation should only be performed for lab tests recorded at least twice for the patient. The
final output should be a dataset where each row corresponds to a unique PatientID. Include the
patient's Sex, Age, Outcome, and LOS. For every lab test parameter present in the dataset, create
a new column representing the calculated change. If a lab test was recorded less than twice for a
patient, the corresponding change column should contain a missing value.
-----------------------------------------------------------------------------
# ID: 17
# Dataset: TJH
# Task type: Visualization
# Task: Build a binary classification model to predict the 'Outcome' for each patient.
```

1. **Data Preparation**: For each unique patient ('PatientID'), select the record closest to
their 'AdmissionTime' to represent their initial state upon admission. Select relevant features
(initial lab values, demographics like 'Sex', 'Age') for modeling.
2. **Model Training**: Train a classification model using the prepared initial patient data to
predict the 'Outcome'. Split your data into training and testing sets to evaluate generalization
performance.
3. **Evaluation**: Evaluate the trained model's performance on the test set using appropriate
metrics for binary classification.
4. **Visualization**: Visualize the importance of the features used by your trained model in
predicting the outcome. For tree-based models, this could be a bar chart showing feature
importance scores. For linear models, you could visualize the coefficients.
--------------------------------------------------------------------------------
# ID: 18
# Dataset: TJH
# Task type: Visualization
# Task: Predict the patient 'Outcome' based on clinical data collected during the initial phase of
their hospitalization.

1. **Data Preparation**: For each unique patient, extract all records within the first 48 hours
of their 'AdmissionTime'. Aggregate the clinical measurements from these records to create
patient-level features. Include 'Sex' and 'Age' as features. The target variable is 'Outcome'.
2. **Model Training**: Train a binary classification model using the prepared patient-level
features to predict the 'Outcome'. Split your data into training and testing sets to evaluate
generalization performance.
3. **Evaluation**: Evaluate the trained model's performance on the test set using appropriate
metrics for binary classification.
4. **Visualization**:
 * Generate a visualization comparing the distribution of 'Length of Stay (LOS)' for patients
 belonging to each 'Outcome' class. This helps illustrate the difference in hospital duration
 associated with the outcomes.
 * Based on your model's results , select the top 5 most influential aggregated clinical features.
 For each selected feature, create a visualization showing the distribution of that feature's
 values separately for patients with Outcome=0 and Outcome=1.
--------------------------------------------------------------------------------
# ID: 19
# Dataset: TJH
# Task type: Visualization
# Task: Build a binary classification model to predict the patient 'Outcome' using the clinical
measurements recorded closest to their discharge time.

1. **Data Preparation**: For each unique patient ('PatientID'), select the record with the latest
'RecordTime' that is on or before their 'DischargeTime'. Select relevant features (demographics
like 'Sex', 'Age', and the lab values from the selected latest record) for modeling.
2. **Model Training**: Train a classification model using the prepared patient data (features
from the latest record) to predict the 'Outcome'. Split your data into training and testing sets.
3. **Evaluation**: Evaluate the trained model's performance on the test set using appropriate
metrics for binary classification.
4. **Visualization**: Analyze how the *change* in key clinical markers during hospitalization
relates to the final outcome.
 * Identify 2-3 clinically relevant or model-important numerical features.
 * For each identified feature, calculate the difference between its value in the *latest* record
 (used for prediction) and its value in the *earliest* record (at or closest to admission time)
 for each patient.
 * Create a visualization for each selected feature, showing the distribution of these *changes*
 separately for patients with Outcome=0 and Outcome=1.
--------------------------------------------------------------------------------
# ID: 20
# Dataset: TJH
# Task type: Visualization
# Task: Predict the patient 'Outcome' using clinical data collected during the first week of their
hospitalization and visualize the average trajectories of key features compared between outcomes.

1. **Data Preparation**: For each unique patient, filter records where `RecordTime` is within 7
days *inclusive* of their `AdmissionTime`. If a patient was discharged *before* the 7th day,
include all their records up to their `DischargeTime`. Create patient-level features by
aggregating the clinical measurements available within this window. For numerical features,
calculate relevant statistics. Include demographic features (`Sex`, `Age`).The target variable is
`Outcome`.

2. **Model Training**: Split the prepared patient dataset (with aggregated features) into
training and testing sets. Train a binary classification model on the training data to predict
the `Outcome`.

3. **Evaluation**: Evaluate the trained model's performance on the test set using appropriate
metrics for binary classification.

4. **Visualization**: Select 2-3 continuous clinical features from the dataset that are measured
frequently and are likely to change over time during a hospital stay. For each selected feature:

```
  * Calculate the day relative to admission for every record (`RecordTime` - `AdmissionTime`).
  * For records within the first 7 days (relative day 0 to 7), calculate the *average* value of
    the feature for each specific day relative to admission, separately for patients with `Outcome`
    = 0 and patients with `Outcome` = 1.
  * Create a line plot showing the average trajectory of the feature over the first 7 days
    (x-axis: Days relative to Admission, y-axis: Average Feature Value). Include two lines on the
    plot, one for each outcome group (`Outcome` 0 and `Outcome` 1).
-------------------------------------------------------------------------
# ID: 21
# Dataset: TJH
# Task type: Visualization
# Task: Build models to predict the final patient 'Outcome' using the sequence of clinical data
available up to increasing time points during their hospitalization, and visualize how the model's
predictive performance evolves as more historical data is included.

 1. **Data Preparation**: For each patient, organize their records chronologically by `RecordTime`.
 Calculate the time elapsed since `AdmissionTime` for each record. Select a consistent set of
 relevant numerical and categorical clinical features from the records, including 'Sex' and 'Age'
 associated with the patient. Ensure a patient-wise split into training and testing sets.
 2. **Sequential Modeling and Evaluation Over Time**: Define a series of prediction time points
 relative to admission. For each time point `T`:
  * Construct a dataset where each sample represents a patient and consists of the sequence of
    their records from `AdmissionTime` up to `AdmissionTime + T` days (inclusive). For patients
    discharged before `AdmissionTime + T`, include all records up to their `DischargeTime`. The
    target variable is the patient's final `Outcome`.
  * Using the pre-defined patient split, prepare the training and testing data for this specific
    time point `T`.
  * Train a binary classification model capable of handling sequential or variable-length input on
    the training data for time point `T` to predict the final `Outcome`.
  * Evaluate the trained model's performance on the corresponding test data using the Area Under
    the ROC Curve (AUC).
 3. **Performance Trajectory Visualization**: Create a line plot visualizing the test AUC score
 obtained at each prediction time point `T`. The x-axis should represent the time points, and the
 y-axis should represent the corresponding test AUC values.
-------------------------------------------------------------------------
# ID: 22
# Dataset: TJH
# Task type: Visualization
# Task: Predict the patient 'Outcome' based on aggregated clinical data from their *entire*
hospital stay and visualize the distribution of key aggregated features comparing patients with
different outcomes.

 1. **Data Preparation**: For each unique patient ('PatientID'), extract all records recorded
 between their 'AdmissionTime' and 'DischargeTime' (inclusive). For numerical clinical
 measurements, aggregate the data across all these records for each patient by calculating summary
 statistics. Include 'Sex' and 'Age' as static patient features. The target variable is 'Outcome'.
 2. **Model Training**: Split the dataset (containing patient-level aggregated features and
 demographics) into training and testing sets. Train a binary classification model on the training
 data to predict the 'Outcome'.
 3. **Evaluation**: Evaluate the trained model's performance on the test set using appropriate
 binary classification metrics.
 4. **Visualization**: Select 2-3 aggregated numerical features. For each selected aggregated
 feature, create a visualization showing the distribution of its values separately for patients
 with Outcome=0 and Outcome=1.
-------------------------------------------------------------------------
# ID: 23
# Dataset: TJH
# Task type: Visualization
# Task: Build a regression model to predict the change in a specific key clinical marker,
'Hypersensitive c-reactive protein' (Hs-CRP), over the patient's hospital stay, using data
available at the time of admission.

 1. **Data Preparation**: For each unique patient ('PatientID'), identify the record with the
 earliest 'RecordTime' on or after their 'AdmissionTime' (representing the initial state) and the
 record with the latest 'RecordTime' on or before their 'DischargeTime' (representing the state
 near discharge). Extract the value of 'Hypersensitive c-reactive protein' from both the initial
 and latest records for each patient. Calculate the target variable: `Hs-CRP_Change` = (Hs-CRP
 value in the latest record) - (Hs-CRP value in the initial record). *Include only patients for
 whom 'Hypersensitive c-reactive protein' values are available in both the initial and latest
 records.*
  Create features using the data from the *initial* record for each patient: 'Sex', 'Age', and the
  values of other numerical clinical measurements available in that initial record.

 2. **Model Training**: Split the prepared dataset (where each sample is a patient, features are
 derived from the initial record, and the target is the calculated `Hs-CRP_Change`) into training
 and testing sets. Train a regression model to predict the `Hs-CRP_Change`.
```

3. **Evaluation**: Evaluate the trained model's performance on the test set using appropriate
   regression metrics.

   4. **Visualization**: Create a scatter plot on the test set showing the relationship between the
   model's *predicted* `Hs-CRP_Change` values and the *actual* `Hs-CRP_Change` values. Color-code or
   visually distinguish the points on this scatter plot based on the patient's final 'Outcome' (0 or
   1).
--------------------------------------------------------------------------------
# ID: 24
# Dataset: TJH
# Task type: Reporting
# Task: Analyze the provided patient dataset to understand factors associated with patient outcome.
This involves: 1. Identifying key lab parameters that show significant differences at or near
admission between patients with outcome 0 and outcome 1. 2. Analyzing and describing the trends
over time for a few selected important lab parameters for patients in each outcome group, relative
to their admission time. 3. Generating a report summarizing the findings, including demographic
overview, parameters significantly different at admission, observed trends, and potential clinical
insights.
--------------------------------------------------------------------------------
# ID: 25
# Dataset: TJH
# Task type: Reporting
# Task: Analyze the provided longitudinal patient dataset to investigate how the variability and
trajectory of key laboratory parameters within the initial period after admission correlate with
patient outcomes (specifically 'Outcome' and 'LOS'). This task requires: 1. Selecting a subset of
clinically significant laboratory parameters from the dataset. 2. For each patient, calculating
descriptive statistics or trend indicators for the selected parameters using only the records
within the first 48 or 72 hours following their 'AdmissionTime'. 3. Analyzing the relationship
between these calculated early temporal metrics and the patient's 'Outcome' (binary
classification) and 'LOS' (continuous variable). 4. Generating a concise report detailing the
chosen parameters, the derived temporal metrics, the statistical methods used for analysis, and a
summary of the findings highlighting which early lab dynamics are most strongly associated with
patient 'Outcome' and 'LOS', along with potential clinical interpretations.
--------------------------------------------------------------------------------
# ID: 26
# Dataset: TJH
# Task type: Reporting
# Task: Analyze the provided longitudinal patient dataset to identify temporal patterns and
trajectories of key laboratory parameters that are predictive of patient outcome (Outcome) and
length of stay (LOS). The task requires: 1. Selecting a subset of clinically relevant laboratory
parameters with sufficient temporal variability across patient records. 2. For each patient,
characterize the time-series data for the selected parameters. This may involve aligning data by
admission time and engineering features from the trajectories over the entire duration of
available records or within a defined analytical window. 3. Utilizing these engineered trajectory
features as input, build models to predict the patient's Outcome (binary classification) and LOS
(regression or classification into duration bins). 4. Analyze the trained models to identify which
engineered temporal features derived from the lab parameter trajectories are most influential or
predictive for patient Outcome and LOS. 5. Generate a report summarizing the chosen parameters,
the approach for trajectory characterization and feature engineering, the methodology for
predictive modeling, the performance metrics of the models, and the key findings regarding the
relationship between specific lab parameter trajectories and patient outcomes/LOS.
--------------------------------------------------------------------------------
# ID: 27
# Dataset: TJH
# Task type: Reporting
# Task: Analyze the provided longitudinal patient dataset to investigate the relationship between
the *intra-patient variability* of key laboratory parameters measured during hospitalization and
patient outcomes (Outcome and Length of Stay - LOS). This task requires: 1. Selecting a subset of
laboratory parameters that are frequently measured across multiple days for individual patients
and show potential for significant fluctuation within a patient's stay. 2. For each selected
parameter, calculate a measure of intra-patient variability across all available records for each
patient *during their hospitalization period* (between AdmissionTime and DischargeTime). You
should address how to handle parameters with only one measurement per patient. 3. Analyze the
association between the calculated variability measures and the patient's Outcome and LOS. 4.
Generate a report summarizing the parameters analyzed, the chosen variability metrics, the
analytical methods used, and the findings on which parameters' variability is most strongly
associated with Outcome and LOS, including relevant descriptive statistics and visualizations.
--------------------------------------------------------------------------------
# ID: 28
# Dataset: TJH
# Task type: Reporting

```
# Task: Analyze the provided longitudinal patient dataset to investigate the relationship between
the *change* in key laboratory parameters from the first recorded measurement to the last recorded
measurement during hospitalization and the patient's outcome ('Outcome'). This task requires: 1.
For a selected subset of laboratory parameters that appear across multiple records for individual
patients, identify the value from the earliest available record and the value from the latest
available record for each patient. 2. Calculate the absolute or percentage change for each
selected parameter between the earliest and latest measured values for each patient. 3. Compare
the distribution and statistical properties of these calculated 'first-to-last' changes for
patients in the 'Outcome=0' group versus patients in the 'Outcome=1' group using appropriate
statistical tests and visualizations. 4. Generate a report summarizing the selected parameters,
the methodology for defining and calculating the 'first-to-last' change, the results of the
comparative analysis highlighting parameters with statistically significant differences in change
between the outcome groups, and potential clinical interpretations regarding how the trajectory
(specifically, the overall change) of certain lab markers during hospitalization is associated
with patient outcome.
--------------------------------------------------------------------------------
# ID: 29
# Dataset: TJH
# Task type: Reporting
# Task: Analyze the relationship between patient outcomes (Outcome and Length of Stay) and the
values of key laboratory parameters measured within the 48-hour window immediately preceding
hospital discharge. This task requires: 1. For each patient, identify their DischargeTime and
define a 48-hour pre-discharge window (DischargeTime - 48 hours to DischargeTime). 2. For a
selected subset of clinically relevant and frequently measured laboratory parameters, extract the
value from the *latest* available record for that specific parameter within this 48-hour
pre-discharge window for each patient. If no record exists for a parameter within this window for
a patient, handle this missing data appropriately. 3. Compare the distributions and statistical
properties of these extracted 'pre-discharge' parameter values between the two Outcome groups (0
vs 1) using appropriate statistical tests and visualizations. Identify parameters with
statistically significant differences. 4. Analyze the correlation between these extracted
'pre-discharge' parameter values and the patient's Length of Stay (LOS) using appropriate
correlation coefficients. 5. Generate a report summarizing the methodology, the list of parameters
analyzed, the findings regarding their 'pre-discharge' values and associations with both Outcome
and LOS, including relevant descriptive statistics, test results, and interpretations of the
findings in a clinical context.
--------------------------------------------------------------------------------
# ID: 30
# Dataset: TJH
# Task type: Reporting
# Task: Analyze the provided longitudinal patient dataset to identify distinct patient cohorts
based on their *average* laboratory parameter profiles during hospitalization and investigate the
association of these cohorts with patient outcomes (Outcome and Length of Stay). This task
requires: 1. Selecting a subset of clinically relevant laboratory parameters that are frequently
measured across patients. 2. For each patient, calculate the *mean* value for each selected
parameter using all available measurements recorded *between* their 'AdmissionTime' and
'DischargeTime'. You will need to consider and document how to handle parameters with insufficient
measurements for a given patient. 3. Apply a clustering algorithm to group patients based on their
calculated mean laboratory parameter values. You should explore different numbers of clusters and
justify your choice. 4. Characterize each identified cluster by examining the average values of
the selected laboratory parameters within the cluster and comparing them across clusters. 5.
Statistically analyze whether the distribution of 'Outcome' (binary) and the mean 'LOS'
(continuous) differ significantly between the identified patient clusters using appropriate
statistical tests. 6. Generate a report summarizing the chosen parameters, the approach for
calculating mean values and handling missing data, the clustering methodology and resulting
cluster characteristics, the statistical analysis of outcome/LOS differences between clusters, and
potential clinical interpretations of the identified cohorts in relation to their lab profiles and
outcomes.
--------------------------------------------------------------------------------
# ID: 31
# Dataset: TJH
# Task type: Reporting
# Task: Analyze the provided longitudinal patient dataset to investigate the relationship between
the frequency of laboratory parameter measurements during hospitalization and patient outcome
('Outcome'). This task requires: 1. Selecting a subset of clinically relevant laboratory
parameters that are expected to be measured multiple times during a hospital stay. 2. For each
patient, determine the total number of measurements recorded for each selected parameter between
their 'AdmissionTime' and 'DischargeTime'. 3. Calculate the total duration of hospitalization for
each patient ('LOS'). 4. Analyze the relationship between the total number of measurements (or
measurement density) for the selected parameters and the patient's 'Outcome' (0 for discharged, 1
for died). This analysis could involve comparing the distribution of measurement counts/density
between outcome groups using appropriate statistical tests or visualization. 5. Generate a report
summarizing the selected parameters, the methodology for calculating measurement frequency, the
results of the comparative analysis highlighting parameters whose measurement frequency is
significantly associated with outcome, and potential interpretations regarding how monitoring
intensity might differ between patients with different outcomes.
--------------------------------------------------------------------------------
# ID: 32
# Dataset: TJH
```

```
# Task type: Reporting
# Task: Analyze the relationship between the extreme values (peak and nadir) of key laboratory
parameters recorded during hospitalization and patient outcomes (Outcome and Length of Stay). This
task requires: 1. Selecting a subset of clinically relevant laboratory parameters that are
measured multiple times for at least some patients. 2. For each patient, identify the maximum
(peak) and minimum (nadir) value recorded for each selected parameter across all their records
between their 'AdmissionTime' and 'DischargeTime'. You should document how you handle parameters
with only one measurement per patient during their stay. 3. Investigate the association between
these calculated peak and nadir values and the patient's 'Outcome' (binary classification) and
'LOS' (continuous variable). This analysis could involve comparing peak/nadir distributions
between outcome groups using statistical tests and visualizations, and assessing the correlation
between peak/nadir values and LOS. 4. Generate a report summarizing the chosen parameters, the
methodology for extracting extreme values, the results of the association analysis highlighting
parameters whose peak or nadir values are significantly related to Outcome or LOS, and potential
clinical interpretations.
--------------------------------------------------------------------------------
# ID: 33
# Dataset: TJH
# Task type: Reporting
# Task: Analyze the provided longitudinal patient dataset to investigate the relationship between
the *proportion of records with values outside statistically defined 'typical' ranges* for key
laboratory parameters during hospitalization and patient outcomes (Outcome and Length of Stay).
This task requires: 1. Selecting a subset of clinically relevant laboratory parameters that are
measured multiple times for at least some patients. 2. For each selected parameter, define
'typical' value ranges statistically. Clearly state the statistical method used for defining these
ranges. 3. For each patient, using only records with a RecordTime between their AdmissionTime and
DischargeTime, count the number of records where the value for a selected parameter falls outside
the defined 'typical' range. 4. Calculate the total number of records for that parameter within
the hospitalization period for each patient. 5. For each patient and selected parameter, calculate
the *proportion* of records with values outside the 'typical' range (count of atypical records /
total count of records for that parameter during hospitalization). Address patients with
insufficient measurements for a given parameter. 6. Analyze the association between these
calculated 'proportion of atypical records' metrics and the patient's 'Outcome' (binary) and 'LOS'
(continuous). This involves comparing the distribution of the 'proportion atypical' metric between
Outcome groups using statistical tests and visualizations, and assessing correlation with LOS. 7.
Generate a report summarizing the chosen parameters, the methodology for defining 'typical' ranges
and calculating the 'proportion atypical', the results of the association analysis with Outcome
and LOS highlighting parameters with significant findings, including relevant descriptive
statistics, test results, and clinical interpretations.
--------------------------------------------------------------------------------
# ID: 34
# Dataset: TJH
# Task type: Reporting
# Task: Analyze the relationship between patient outcomes ('Outcome') and their laboratory
parameter values recorded around a specific time point during hospitalization. The task requires:
1. Select a target day relative to admission. 2. For each patient, identify the record whose
'RecordTime' is closest to this target day. Define a reasonable time window around the target day
within which to search for the closest record. 3. For a selected subset of clinically relevant
laboratory parameters, extract the values from the identified 'closest' record for each patient.
Address missing values for parameters that were not measured on the identified record. 4. Compare
the distributions and statistical properties of these extracted parameter values on the target day
between the two Outcome groups (0 vs 1) using appropriate statistical tests and visualizations.
Identify parameters with statistically significant differences at a chosen significance level. 5.
Generate a report summarizing the chosen target day and search window, the list of parameters
analyzed, the methodology for selecting records and handling missing data, the results of the
comparative analysis highlighting parameters whose values around the target day are significantly
associated with outcome, including relevant descriptive statistics, test results, and
interpretations of the findings in a clinical context.
--------------------------------------------------------------------------------
# ID: 35
# Dataset: TJH
# Task type: Reporting
# Task: Analyze the provided longitudinal patient dataset to identify frequently co-occurring
laboratory tests within individual patient records and investigate if the prevalence of these
co-occurrence patterns differs significantly between patients with different outcomes ('Outcome' 0
vs. 1). The task requires: 1. Grouping the data by 'PatientID' and 'RecordTime' to identify the
set of laboratory parameters measured at each specific point in time for each patient. 2. Identify
frequent sets of laboratory parameters that are measured together across all records in the
dataset, using techniques such as frequent itemset mining with a defined minimum support threshold.
3. For the most frequent co-occurring sets of tests, calculate the proportion of records where
this set appears, separately for records belonging to patients with 'Outcome' 0 and records
belonging to patients with 'Outcome' 1. 4. Perform a statistical test to determine if the
difference in proportions for these frequent co-occurring sets between the two outcome groups is
statistically significant. 5. Generate a report summarizing the identified frequent co-occurrence
patterns, the methods used for frequency analysis and statistical comparison, the results
highlighting patterns with statistically significant differences in prevalence between outcome
groups, and discuss potential clinical reasons or implications for these findings.
--------------------------------------------------------------------------------
```

```
# ID: 36
# Dataset: TJH
# Task type: Modeling
# Task: Build a predictive model to determine the final outcome ('Outcome') for each patient. Your
model should utilize the patient's static attributes (Sex, Age) and process the sequential
clinical measurements recorded at different time points ('RecordTime') during their
hospitalization ('AdmissionTime' to 'DischargeTime'). The task is a binary classification problem
to predict whether 'Outcome' is 0 or 1 based on the available data up to a certain point in time.
-------------------------------------------------------------------------------
# ID: 37
# Dataset: TJH
# Task type: Modeling
# Task: Predict the total Length of Stay ('LOS') for each patient upon their admission. Your model
should utilize the patient's static attributes (Sex, Age) and the clinical measurements recorded
on their admission day ('RecordTime' == 'AdmissionTime'). If multiple records exist on the
admission day, you can use the first record or an aggregate of records from that day. This task is
a regression problem to predict the continuous value of 'LOS'.
-------------------------------------------------------------------------------
# ID: 38
# Dataset: TJH
# Task type: Modeling
# Task: Build a time-series regression model to predict the value of 'Hypersensitive c-reactive
protein' for a patient at their next recorded time point ('RecordTime'). Your model should utilize
the patient's historical clinical measurements recorded at previous time points, along with their
static attributes ('Sex', 'Age'). The task is to predict the continuous value of 'Hypersensitive
c-reactive protein' based on the available sequence of data for that patient up to the prediction
point.
-------------------------------------------------------------------------------
# ID: 39
# Dataset: TJH
# Task type: Modeling
# Task: Predict the likelihood of a patient having at least one critical laboratory value outside
its normal clinical range at their next recorded time point. The model should utilize the
patient's static attributes (Sex, Age) and their sequence of clinical measurements recorded up to
the current time ('RecordTime'). This is a binary classification problem where the target is 1 if
any of a predefined set of critical lab values falls outside its established normal clinical range
at the patient's subsequent 'RecordTime', and 0 otherwise. The specific critical lab values and
their corresponding normal ranges should be defined based on clinical standards or dataset
characteristics.
-------------------------------------------------------------------------------
# ID: 40
# Dataset: TJH
# Task type: Modeling
# Task: Build a regression model to predict the remaining Length of Stay ('DischargeTime' -
'RecordTime' in days) for a patient at any given record time ('RecordTime'). Your model should
leverage the patient's static attributes (Sex, Age) and the sequence of clinical measurements
recorded up to that specific 'RecordTime'.
-------------------------------------------------------------------------------
# ID: 41
# Dataset: TJH
# Task type: Modeling
# Task: Build a regression model to predict the maximum value of 'Hypersensitive c-reactive
protein' recorded for a patient throughout their entire hospital stay. Your model should utilize
the patient's static attributes (Sex, Age) and all available clinical measurements recorded during
the initial 72 hours from their 'AdmissionTime'. Handle patients without 'Hypersensitive
c-reactive protein' measurements within the first 72 hours appropriately.
-------------------------------------------------------------------------------
# ID: 42
# Dataset: TJH
# Task type: Modeling
# Task: Build a binary classification model to predict, for patients whose hospital stay ('LOS')
is greater than 7 days, whether their 'White blood cell count' will decrease below 10.0 at any
time point after the first 7 days of hospitalization. Your model should utilize the patient's
static attributes (Sex, Age) and the time-series of all available clinical measurements recorded
within the first 7 days from their 'AdmissionTime'.
-------------------------------------------------------------------------------
# ID: 43
# Dataset: TJH
# Task type: Modeling
# Task: Build a multiclass classification model to predict the trajectory pattern of 'Platelet
count' over the remainder of a patient's hospital stay. The model should use the patient's static
attributes (Sex, Age) and the sequence of clinical measurements recorded up to the current
'RecordTime'. The trajectory patterns ('Increasing', 'Decreasing', 'Stable') should be defined
based on the relative change in 'Platelet count' from the measurement at the current 'RecordTime'
to the final measurement recorded before 'DischargeTime'.
-------------------------------------------------------------------------------
# ID: 44
# Dataset: TJH
```

```
# Task type: Modeling
# Task: Build a binary classification model to predict, for a patient at any given record time
('RecordTime') prior to their discharge ('RecordTime' < 'DischargeTime'), whether they will be
discharged within the next 7 days from that 'RecordTime'. The model should utilize the patient's
static attributes (Sex, Age) and the sequence of all clinical measurements recorded for that
patient up to and including that specific 'RecordTime'.
--------------------------------------------------------------------------------
# ID: 45
# Dataset: TJH
# Task type: Modeling
# Task: Build a regression model to predict the maximum absolute difference between consecutive
'White blood cell count' measurements for a patient during their entire hospitalization
('AdmissionTime' to 'DischargeTime'). Your model should utilize the patient's static attributes
(Sex, Age) and all available clinical measurements recorded within the first 48 hours from their
'AdmissionTime'. For patients with no 'White blood cell count' measurements or only one
measurement across their entire stay, exclude them from the training/testing set. For patients
with 'White blood cell count' measurements but none or only one within the first 48 hours, handle
the missing early data appropriately.
--------------------------------------------------------------------------------
# ID: 46
# Dataset: TJH
# Task type: Modeling
# Task: Build a regression model to predict the value of 'Hypersensitive c-reactive protein' for a
patient on the day of their discharge ('DischargeTime'). Your model should utilize the patient's
static attributes (Sex, Age) and the sequence of all clinical measurements recorded for that
patient up to their last available record time ('RecordTime'). Patients who do not have
'Hypersensitive c-reactive protein' recorded at their 'DischargeTime' will require the model to
infer or predict this value. Patients without any 'Hypersensitive c-reactive protein' measurements
throughout their stay could be handled separately.
--------------------------------------------------------------------------------
# ID: 47
# Dataset: TJH
# Task type: Modeling
# Task: Build a regression model to predict the rate of change of 'Creatinine' for a patient
during the first 48 hours of their hospitalization. The rate of change should be calculated as the
slope of a linear fit to all 'Creatinine' measurements recorded between `AdmissionTime` and
`AdmissionTime` + 48 hours (inclusive). Your model should utilize the patient's static attributes
(Sex, Age) and all available clinical measurements recorded within the first 24 hours from their
`AdmissionTime` (inclusive). Exclude patients who do not have at least two 'Creatinine'
measurements recorded within the first 48 hours.
--------------------------------------------------------------------------------
# ID: 48
# Dataset: TJH
# Task type: Data
# Task: Load the provided patient record data. Structure the data into a single table where each
row represents a patient's record at a specific time point. Ensure that all unique clinical
measurement types (`hemoglobin`, `Serum chloride`, etc.) across all records become columns in the
final table. Handle missing values that arise from patients not having certain measurements at
certain times, as well as potentially missing patient-specific attributes like `Sex` or `Age` in
some records.
--------------------------------------------------------------------------------
# ID: 49
# Dataset: TJH
# Task type: Data
# Task: Transform the longitudinal patient records into a single patient-level representation by
aggregating time-varying clinical measurements. For each unique patient, retain static attributes
like Sex, Age, Outcome, and Admission/Discharge times. For dynamic clinical measurements that
appear across multiple records for a patient, calculate summary statistics. The final output
should be a structured dataset where each row corresponds to a unique patient, containing their
static features and the calculated aggregate values for all relevant clinical parameters. Handle
potential missing values during aggregation by using appropriate strategies.
--------------------------------------------------------------------------------
# ID: 50
# Dataset: TJH
# Task type: Data
# Task: Structure the patient records into longitudinal sequences for time-series analysis. Group
the records by `PatientID` and order them chronologically using the `RecordTime`. Identify the
complete set of unique clinical measurement features that appear across all patient records in the
dataset (excluding static patient attributes like `PatientID`, `AdmissionTime`, `DischargeTime`,
`Outcome`, `LOS`, `Sex`, and `Age`). For each patient, transform their ordered sequence of records
into a sequence of feature vectors. Each vector in the sequence corresponds to a single time point
(record) and contains values for all identified unique clinical measurement features. If a
clinical feature was not measured at a specific `RecordTime` for a patient, represent its value in
the corresponding feature vector using a placeholder for missing data.
--------------------------------------------------------------------------------
# ID: 51
# Dataset: MIMIC-IV
# Task type: Data
```

```
# Task: Count the number of unique PatientIDs that have at least one record where the 'Glascow
coma scale total' is less than 10.
--------------------------------------------------------------------------------
# ID: 52
# Dataset: MIMIC-IV
# Task type: Data
# Task: Calculate the average value of the 'Heart Rate' field for all records where the 'Age'
field is greater than 65 and the 'Oxygen saturation' field is less than 95.
--------------------------------------------------------------------------------
# ID: 53
# Dataset: MIMIC-IV
# Task type: Data
# Task: Find the minimum 'Mean blood pressure' for all records where the 'Age' is less than 40 and
the 'LOS' (Length of Stay) is greater than 50.
--------------------------------------------------------------------------------
# ID: 54
# Dataset: MIMIC-IV
# Task type: Visualization
# Task: Visualize the trend of 'Heart Rate' over 'RecordTime' for a specific 'AdmissionID'. Plot
the 'Heart Rate' values on the y-axis and 'RecordTime' on the x-axis to show how the patient's
heart rate changed throughout that particular hospital admission.
--------------------------------------------------------------------------------
# ID: 55
# Dataset: MIMIC-IV
# Task type: Visualization
# Task: Create a box plot to compare the distribution of 'Heart Rate' for patients based on their
'Outcome' (0 or 1). Plot 'Outcome' on the x-axis and 'Heart Rate' on the y-axis to see if there's
a noticeable difference in heart rate distributions between the two outcome groups.
--------------------------------------------------------------------------------
# ID: 56
# Dataset: MIMIC-IV
# Task type: Visualization
# Task: Create a scatter plot to visualize the relationship between Systolic blood pressure and
Diastolic blood pressure. Plot 'Systolic blood pressure' on the x-axis and 'Diastolic blood
pressure' on the y-axis to explore the correlation between these two vital signs across all
records.
--------------------------------------------------------------------------------
# ID: 57
# Dataset: MIMIC-IV
# Task type: Visualization
# Task: Create a box plot to compare the distribution of 'Length of Stay (LOS)' for patients based
on their 'Outcome'. Plot 'Outcome' (0 or 1) on the x-axis and 'LOS' on the y-axis to visualize how
the distribution of hospital stay duration varies between patients with different outcomes.
--------------------------------------------------------------------------------
# ID: 58
# Dataset: MIMIC-IV
# Task type: Visualization
# Task: Create a scatter plot to visualize the relationship between 'Glucose' and 'Heart Rate'.
Plot 'Glucose' on the x-axis and 'Heart Rate' on the y-axis to explore if there is any correlation
or pattern between these two physiological measurements across all records.
--------------------------------------------------------------------------------
# ID: 59
# Dataset: MIMIC-IV
# Task type: Visualization
# Task: Create a box plot to compare the distribution of 'Weight' for patients based on their
'Sex'. Plot 'Sex' on the x-axis and 'Weight' on the y-axis to visualize how the distribution of
patient weights differs between male (Sex=1) and female (Sex=0) patients.
--------------------------------------------------------------------------------
# ID: 60
# Dataset: MIMIC-IV
# Task type: Data
# Task: Calculate the mean and standard deviation of 'Heart Rate', determine the count of records
where 'Outcome' is 1, and count the number of 'nan' values in the 'Height' column.
--------------------------------------------------------------------------------
# ID: 61
# Dataset: MIMIC-IV
# Task type: Data
# Task: Group the data by 'Sex' and, for each group, calculate the average 'Weight' and the
standard deviation of 'Heart Rate'.
--------------------------------------------------------------------------------
# ID: 62
# Dataset: MIMIC-IV
# Task type: Data
# Task: Calculate the median Length of Stay ('LOS') across all records. Additionally, calculate
the Interquartile Range (IQR) of 'Weight' for all records where 'Sex' is 0.
--------------------------------------------------------------------------------
# ID: 63
# Dataset: MIMIC-IV
```

```
# Task type: Data
# Task: For each unique hospital admission (identified by 'AdmissionID'), calculate the mean
'Heart Rate', 'Temperature', and 'Systolic blood pressure' across all recorded measurements for
that admission. Handle missing values in these vital sign columns by excluding them from the
average calculation.
--------------------------------------------------------------------------------
# ID: 64
# Dataset: MIMIC-IV
# Task type: Data
# Task: For each unique hospital admission (identified by 'AdmissionID'), fill the missing
numerical values in the clinical measurement columns using a forward fill approach within that
admission group. If the first value within an admission for a specific column is still missing
after the forward fill, use a backward fill for those remaining missing values within the same
admission group.
--------------------------------------------------------------------------------
# ID: 65
# Dataset: MIMIC-IV
# Task type: Data
# Task: For each unique hospital admission (identified by 'AdmissionID'), calculate the range
(maximum value minus minimum value) for 'Systolic blood pressure' and 'Heart Rate' across all
recorded measurements for that admission. Handle missing values in these columns by excluding them
from the range calculation.
--------------------------------------------------------------------------------
# ID: 66
# Dataset: MIMIC-IV
# Task type: Visualization
# Task: Build a predictive model to determine the likelihood of a patient being readmitted.

 1. **Data Preparation:** For each unique `AdmissionID`, extract and process relevant features
 from the time-series data. This could involve aggregating the time-series measurements. Combine
 these aggregated features with static patient/admission information ('Age', 'Sex', 'LOS',
 'Outcome') to create a single feature vector per admission.
 2. **Missing Value Handling:** Address the missing values ('nan') in the dataset.
 3. **Model Training:** Train a binary classification model on the prepared admission-level
 dataset to predict the `Readmission` outcome.
 4. **Evaluation and Visualization:** Evaluate your model's performance using appropriate metrics
 for binary classification. Then, create a visualization that helps interpret the model or its
 results.
--------------------------------------------------------------------------------
# ID: 67
# Dataset: MIMIC-IV
# Task type: Visualization
# Task: Build a predictive model to forecast the patient's `Outcome` using only the clinical data
available within the first 24 hours of their admission to the ICU.

 1. **Data Preparation:** For each unique `AdmissionID`, identify the first `RecordTime`. Select
 all records (`RecordID`) associated with that `AdmissionID` whose `RecordTime` is within 24 hours
 of this initial time point. Aggregate the numerical clinical measurements from these early
 records. Calculate summary statistics for each measurement within this 24-hour window. Combine
 these aggregated features with static patient/admission characteristics available at the start of
 the stay ('Age', 'Sex').
 2. **Model Training:** Train a binary classification model on the prepared dataset (where each
 row represents an admission, and features are derived from the first 24 hours) to predict the
 `Outcome` (0 or 1).
 3. **Visualization:** Select two to three clinically relevant aggregated features that you
 calculated from the first 24 hours. Create comparative visualizations that display the
 distribution of these selected features separately for the two `Outcome` classes (patients with
 Outcome 0 and patients with Outcome 1). This visualization should help illustrate how the
 distribution of these key early physiological indicators differs based on the patient's final
 outcome.
--------------------------------------------------------------------------------
# ID: 68
# Dataset: MIMIC-IV
# Task type: Visualization
# Task: Build a time-series predictive model to forecast the patient's `Outcome` using the
sequence of clinical observations recorded during their stay. This differs from aggregating
features by processing the temporal sequence directly.

 1. **Data Preparation:** Group the records by `AdmissionID`. For each admission, create sequences
 of the relevant numerical clinical measurements (`Capillary refill rate` through `pH`). Include
 static features like 'Age' and 'Sex' as appropriate inputs to your sequence model.
 2. **Model Training:** Train a sequence classification model to predict the binary `Outcome` for
 each admission based on the prepared sequences and static features. You might consider using the
 data up to a certain time horizon within each admission for training, or train models that can
 process sequences of varying lengths.
```

3. **Evaluation and Visualization:** Evaluate your sequence model's performance using appropriate binary classification metrics. To fulfill the visualization requirement and analyze the temporal predictive power, create a plot that shows how a key performance metric changes when the model is trained and evaluated using progressively longer segments of the time-series data from the start of the admission. This visualization should illustrate how the predictability of the final `Outcome` evolves over the course of the patient's stay based on the available data.

------------------------------------------------------------------------------

# ID: 69
# Dataset: MIMIC-IV
# Task type: Visualization
# Task: Predict the `Outcome` for each admission by analyzing the longitudinal patterns and summary statistics of the clinical measurements over the patient's entire stay.

1. **Data Preparation:** Group the records by `AdmissionID`. For each admission, engineer features from the time-series data (`Capillary refill rate` through `pH`) that capture not only the *summary statistics* but also *characteristics of the trajectory* for key physiological variables across the *entire* admission. Combine these engineered features with static patient/admission features (`Age`, `Sex`, `LOS`).
2. **Model Training:** Train a binary classification model on the admission-level dataset prepared in step 1 to predict the `Outcome` (0 or 1).
3. **Evaluation and Visualization:** Evaluate your model's performance. To fulfill the visualization requirement, select *at least two* engineered features that represent the *variability* or *trend* of a physiological measurement over time within the admission. Create comparative visualizations showing the distribution of these selected engineered features separately for patients with Outcome 0 and Outcome 1.

------------------------------------------------------------------------------

# ID: 70
# Dataset: MIMIC-IV
# Task type: Visualization
# Task: Build a model to predict whether a patient is likely to have a 'Short Stay' or a 'Long Stay', and to visualize characteristics of patients where the model makes prediction errors.

1. **Data Preparation:**
 * Define 'Short Stay' and 'Long Stay' based on the `LOS`. Calculate the median `LOS` across all admissions.
 * For each unique `AdmissionID`, aggregate the time-series numerical clinical measurements by computing summary statistics. Include static features like 'Age', 'Sex', 'Outcome', and 'Readmission' in the aggregated dataset.

2. **Model Training and Evaluation:**
 * Split the aggregated admission-level dataset into training and testing sets.
 * Train a binary classification model on the training data to predict the 'Short Stay' vs. 'Long Stay' outcome.
 * Evaluate the performance of your trained model on the test set using appropriate binary classification metrics.

3. **Visualization of Prediction Errors vs. Patient Characteristics:**
 * Select *two* features from your aggregated dataset that represent different aspects of patient characteristics or clinical state .
 * Using the test set predictions, identify the instances that were misclassified by your model. Categorize these misclassified instances into False Positives (predicted 'Long Stay', actual 'Short Stay') and False Negatives (predicted 'Short Stay', actual 'Long Stay').
 * Create a scatter plot or a similar visualization showing the distribution or relationship between your two selected features *specifically* for the misclassified instances. Clearly distinguish between False Positives and False Negatives in the visualization. This plot should provide insights into the characteristics of patients that the model struggles to classify correctly, helping to diagnose model weaknesses.

------------------------------------------------------------------------------

# ID: 71
# Dataset: MIMIC-IV
# Task type: Visualization
# Task: Identify distinct groups (clusters) of patients based on their clinical state during the initial period of their admission and to visualize the distribution of the Length of Stay (`LOS`) for each identified group.

1. **Data Preparation:**
 * For each unique `AdmissionID`, aggregate the numerical clinical measurements (`Capillary refill rate` through `pH`) taken within the first 24 hours of the admission. Calculate summary statistics for each measurement across this time window. If a measurement is not recorded within the first 24 hours for a specific admission, handle it appropriately.
 * Combine these aggregated features with the static patient/admission features available early on ('Age', 'Sex').
 * Create a single feature vector for each `AdmissionID`.

2. **Patient Clustering:**
 * Apply a clustering algorithm to the prepared admission-level dataset (using the features derived from the first 24 hours). Choose a suitable number of clusters.
 * Assign each admission to its corresponding cluster.

```
  3. **Visualization of Length of Stay per Cluster:**
    * Create a comparative visualization that displays the distribution of the `LOS` for patients in
    each of the identified clusters.
    * The visualization should clearly show the differences (or similarities) in `LOS` distributions
    across the patient groups defined by their early clinical profiles.
    ----------------------------------------------------------------------------
# ID: 72
# Dataset: MIMIC-IV
# Task type: Reporting
# Task: Analyze the provided patient admission data to identify which clinical measurements (such
as vital signs, GCS scores, glucose, pH), demographic features (age, sex), or admission-specific
details (LOS) are most significantly associated with the patient `Outcome`. Generate a report
summarizing the key findings, including descriptive statistics comparing the distributions of
relevant factors for different outcome groups and an analysis of which variables show the
strongest statistical association or predictive power regarding the outcome.
----------------------------------------------------------------------------
# ID: 73
# Dataset: MIMIC-IV
# Task type: Reporting
# Task: Investigate the temporal dynamics of patient health status during hospitalization using
the provided dataset. The task is to analyze how key clinical measurements (e.g., Heart Rate,
Blood Pressure, Temperature, Oxygen saturation, Glucose, GCS scores) evolve over the course of an
admission. Generate a report that summarizes the typical patterns of these measurements over time
for different patient outcomes (e.g., those with Outcome=0 vs. Outcome=1) or readmission statuses.
The report should include visualizations showing variable trajectories and identify significant
trends or critical points in the time series data that may distinguish patient groups.
----------------------------------------------------------------------------
# ID: 74
# Dataset: MIMIC-IV
# Task type: Reporting
# Task: Analyze the provided patient admission data to identify demographic and initial clinical
characteristics associated with patient readmission. Focus on analyzing patient-level data,
aggregating or selecting features from the start of each admission (identified by `AdmissionID`),
such as Age, Sex, and clinical measurements from the earliest available `RecordTime` within an
admission. Investigate which of these initial features or patient demographic factors are
significantly correlated with or predictive of the `Readmission` flag. Generate a report
summarizing the key findings, including descriptive statistics comparing initial characteristics
of readmitted vs. non-readmitted patients and an analysis identifying the most influential
predictors of readmission risk.
----------------------------------------------------------------------------
# ID: 75
# Dataset: MIMIC-IV
# Task type: Reporting
# Task: Investigate the patterns and potential implications of missing clinical data within the
provided dataset. The task is to identify which clinical measurements (`Capillary refill rate`,
`Diastolic blood pressure`, ..., `pH`) have the highest percentages of missing values (`nan`).
Analyze whether the missingness of specific measurements is correlated with patient demographics
(`Age`, `Sex`), patient outcomes (`Outcome`), or the total length of stay (`LOS`). Furthermore,
explore if there are any observable relationships between the overall level or pattern of
missingness across multiple variables within an admission and the patient's `Readmission` status.
Generate a report summarizing these findings on missing data distribution, including
variable-specific missingness rates and any statistically significant associations found between
missing data patterns and patient characteristics or outcomes.
----------------------------------------------------------------------------
# ID: 76
# Dataset: MIMIC-IV
# Task type: Reporting
# Task: Analyze the impact of maximum physiological severity during a hospital admission on
patient outcomes. For each unique `AdmissionID`, calculate the 'worst' value observed across the
entire stay for key vital signs and clinical scores (e.g., minimum Oxygen saturation, maximum
Heart Rate, minimum Glascow coma scale total, etc.). Investigate the correlation and association
of these worst-case values with patient `Outcome` (hospital mortality) and `LOS` (Length of Stay).
Generate a report summarizing the calculated peak severity metrics, their distribution relative to
patient outcomes, and the statistical relationships found with hospital mortality and length of
stay.
----------------------------------------------------------------------------
# ID: 77
# Dataset: MIMIC-IV
# Task type: Reporting
```

```
# Task: Investigate the relationship between the variability of key physiological measurements
observed during a patient's hospitalization and their Length of Stay (`LOS`). For each unique
`AdmissionID`, calculate measures of variability (such as standard deviation, interquartile range,
or range) for critical vital signs and clinical parameters like `Heart Rate`, `Mean blood
pressure`, `Oxygen saturation`, and `Temperature`, considering all recorded values within that
specific admission. Analyze the correlation between these computed variability metrics and the
`LOS` for each admission. Generate a report summarizing the findings, including descriptive
statistics of the variability measures across the dataset and the statistical associations found
between measurement variability and the duration of hospitalization (`LOS`).
--------------------------------------------------------------------------------
# ID: 78
# Dataset: MIMIC-IV
# Task type: Reporting
# Task: Investigate the association between the early rate of physiological change during a
patient's hospitalization and subsequent clinical outcomes. For each hospital admission
(`AdmissionID`), identify the change in key physiological measurements (such as Heart Rate, Mean
blood pressure, Oxygen saturation, or Glasgow coma scale total) between the first recorded value
and values recorded within the initial 12 to 24 hours of admission. Analyze whether the magnitude
and direction of this early change are statistically associated with the patient's `Outcome`
(hospital mortality) or `LOS` (Length of Stay). Generate a report summarizing the calculated early
change metrics and their relationship to these patient outcomes, highlighting any significant
findings.
--------------------------------------------------------------------------------
# ID: 79
# Dataset: MIMIC-IV
# Task type: Reporting
# Task: Investigate the association between the Length of Stay (LOS) of a hospital admission and
the patient's Readmission status. The task involves analyzing the provided data grouped by unique
hospital admissions (`AdmissionID`). For each admission, the total `LOS` and the `Readmission`
flag are available. Compare the distribution of `LOS` for admissions that resulted in a
readmission (`Readmission` = 1) versus those that did not (`Readmission` = 0). Generate a report
detailing this comparison, including descriptive statistics (mean, median, spread) for `LOS` in
each readmission group and the results of statistical tests assessing whether the observed
difference in `LOS` distributions is statistically significant.
--------------------------------------------------------------------------------
# ID: 80
# Dataset: MIMIC-IV
# Task type: Reporting
# Task: Investigate the association between the average physiological state during a patient's
hospitalization and their clinical outcome (hospital mortality). For each unique `AdmissionID`,
calculate the mean value for a selection of continuous clinical measurements (e.g., `Heart Rate`,
`Mean blood pressure`, `Oxygen saturation`, `Temperature`, `Glucose`, and `pH`) across all
available records for that specific admission. Analyze the relationship between these calculated
average values representing the typical physiological state during the stay and the patient's
`Outcome` (where 1 indicates mortality). Generate a report summarizing the distribution of these
average physiological metrics for patients who survived versus those who did not, and quantify the
statistical association or predictive power of these average states with hospital mortality.
--------------------------------------------------------------------------------
# ID: 81
# Dataset: MIMIC-IV
# Task type: Reporting
# Task: Investigate the internal consistency of the Glasgow Coma Scale (GCS) components and their
individual association with patient outcome. The task requires analyzing the relationship between
the recorded Glasgow coma scale eye opening, motor response, and verbal response scores and the
reported Glasgow coma scale total. Specifically, assess how often the sum of the individual
components matches the reported total GCS score and analyze if specific individual component
scores or their distributions are significantly associated with patient Outcome (0 or 1),
independently or in combination, perhaps revealing insights not captured by the total score alone.
Generate a report summarizing the findings, including the rate of GCS component-total
inconsistency and the distribution and association analysis of individual components with patient
outcome.
--------------------------------------------------------------------------------
# ID: 82
# Dataset: MIMIC-IV
# Task type: Reporting
# Task: Investigate the interplay between patient age and physiological status in predicting
hospital mortality. The task is to analyze how the association between a key physiological
indicator reflecting patient severity during hospitalization (e.g., the lowest recorded Oxygen
saturation or lowest Glasgow coma scale total for each admission) and the patient's outcome
(`Outcome`, hospital mortality) is influenced by the patient's `Age`. Conduct an analysis to
explore if this relationship varies across different age ranges or if there is a statistical
interaction effect between age and the chosen physiological indicator on the likelihood of
mortality. Generate a report summarizing the findings, including descriptive statistics of the
chosen physiological indicator and outcome across different age groups, and quantifying the
evidence for age-dependent prognostic value of the indicator.
--------------------------------------------------------------------------------
# ID: 83
# Dataset: MIMIC-IV
```

```
# Task type: Reporting
# Task: Investigate the frequency and patterns of simultaneous occurrences of clinically
significant abnormal vital sign values during hospital admissions. Define clinically relevant
thresholds for critical ranges for key vital signs (e.g., Heart Rate, Mean blood pressure,
Respiratory rate, Oxygen saturation, Temperature, Glucose, pH). For each record (`RecordID`),
identify which of these vital signs fall outside their defined normal or critical ranges. Analyze
if the number of simultaneously abnormal vital signs or specific combinations of abnormal vital
signs recorded at any single time point within an admission are associated with patient outcomes
(`Outcome` - hospital mortality, `LOS` - length of stay, `Readmission`). Generate a report
summarizing the thresholds used, the prevalence of simultaneous vital sign abnormalities across
the dataset, and the statistical association between these co-occurrence patterns (e.g., number of
abnormal signs, specific combinations) and patient outcomes.
-----------------------------------------------------------------------------
# ID: 84
# Dataset: MIMIC-IV
# Task type: Reporting
# Task: Analyze the provided patient admission data to predict the Length of Stay (LOS) for each
hospital admission (`AdmissionID`). The task is to build a predictive model using patient
demographic information (`Age`, `Sex`) and clinical measurements recorded within the first 24
hours of each admission. For each `AdmissionID`, process the records to extract relevant features
from the initial phase of the hospitalization, such as the first recorded value or the mean value
within the first day for applicable clinical measurements (e.g., `Heart Rate`, `Mean blood
pressure`, `Temperature`, `Glucose`, `Glascow coma scale total`). Handle missing values
appropriately. Train a regression model to predict `LOS` using these early-phase features.
Investigate which features or combinations of features are the most significant predictors of
`LOS`. Generate a report summarizing the predictive performance of the model, including evaluation
metrics (e.g., R-squared, RMSE) and an analysis highlighting the importance of the identified
predictor variables for `LOS`.
-----------------------------------------------------------------------------
# ID: 85
# Dataset: MIMIC-IV
# Task type: Modeling
# Task: Develop a predictive model to classify the hospital outcome for patients using the
provided physiological measurements, demographic information, and admission characteristics.
-----------------------------------------------------------------------------
# ID: 86
# Dataset: MIMIC-IV
# Task type: Modeling
# Task: Develop a predictive model using the provided dataset to classify whether a patient will
be readmitted to the hospital ('Readmission' column) based on their demographic information,
physiological measurements, and admission characteristics.
-----------------------------------------------------------------------------
# ID: 87
# Dataset: MIMIC-IV
# Task type: Modeling
# Task: Develop a regression model to predict the Length of Stay (LOS) for each patient admission.
The model should utilize the provided demographic information (Age, Sex), physiological
measurements (Capillary refill rate, Diastolic blood pressure, Fraction inspired oxygen, Glascow
coma scale scores, Glucose, Heart Rate, Height, Mean blood pressure, Oxygen saturation,
Respiratory rate, Systolic blood pressure, Temperature, Weight, pH), and admission characteristics.
Consider how to aggregate or utilize the multiple records available for a single admission to make
the prediction for that admission.
-----------------------------------------------------------------------------
# ID: 88
# Dataset: MIMIC-IV
# Task type: Modeling
# Task: Develop a time-series forecasting model to predict the value of a specific physiological
measurement, for a patient at the next recorded time point within their hospital admission. The
model should utilize the sequence of available physiological measurements and demographic
information up to the current time point for that patient admission.
-----------------------------------------------------------------------------
# ID: 89
# Dataset: MIMIC-IV
# Task type: Modeling
# Task: Develop a time-series classification model to predict the short-term risk of clinical
deterioration for hospitalized patients. Specifically, train a model using the provided
physiological measurements and patient demographics to predict, at any given recorded time point
during a patient's admission, whether a significant clinical deterioration event will occur within
the subsequent 24 hours. The definition of 'significant clinical deterioration' should be
established based on critical changes or thresholds in the provided physiological parameters.
-----------------------------------------------------------------------------
# ID: 90
# Dataset: MIMIC-IV
# Task type: Modeling
```

```
# Task: Develop a regression model to predict the overall physiological instability observed
during a patient's hospital admission. Physiological instability can be quantified by calculating
the average variability of key vital signs over the entire duration of the admission. The model
should utilize the available physiological measurements and demographic information recorded
within the first 48 hours of the admission to make this prediction.
--------------------------------------------------------------------------------
# ID: 91
# Dataset: MIMIC-IV
# Task type: Modeling
# Task: Develop a predictive model to estimate the patient's Glasgow Coma Scale Total score
approximately 24 hours after their hospital admission begins. The model should utilize patient
demographic information (Age, Sex) and physiological measurements recorded within the first few
hours of the admission. Consider appropriate data processing to handle multiple records per
patient admission and define the target variable as the GCS Total score recorded closest to the
24-hour mark, or a suitable aggregate if multiple records exist around that time.
--------------------------------------------------------------------------------
# ID: 92
# Dataset: MIMIC-IV
# Task type: Modeling
# Task: Develop a predictive model to classify whether a patient's Respiratory Rate ('Respiratory
rate') will exceed a threshold of 30 at any point *after* the first 24 hours of their admission,
using only data recorded within the first 24 hours of that admission. The model should process the
time-series data available in the initial period and predict this future binary outcome.
--------------------------------------------------------------------------------
# ID: 93
# Dataset: MIMIC-IV
# Task type: Modeling
# Task: Develop a regression model to predict the average value of the 'Heart Rate' physiological
measurement over the entire duration of a patient's hospital admission. The model should use
patient demographic information (Age, Sex) and physiological measurements recorded *only within
the first 12 hours* of that specific admission as input features. You will need to calculate the
target variable (average Heart Rate) for each admission by taking the mean of all 'Heart Rate'
records available for that AdmissionID, and extract/aggregate features from the records associated
with that AdmissionID that fall within the initial 12-hour window.
--------------------------------------------------------------------------------
# ID: 94
# Dataset: MIMIC-IV
# Task type: Modeling
# Task: Develop a binary classification model to predict, for a given record representing a time
point within a patient's hospital admission, whether the 'pH' physiological measurement will be
missing ('nan') in the *next* recorded time point for that same admission. The model should
utilize the physiological measurements and demographic information available at the current time
point as input features.
--------------------------------------------------------------------------------
# ID: 95
# Dataset: MIMIC-IV
# Task type: Modeling
# Task: Develop a binary classification model to predict for each hospital admission (identified
by `AdmissionID`), whether the physiological measurements recorded in the *final* record for that
admission indicate a state of "Respiratory Concern". Define "Respiratory Concern" as meeting *at
least one* of the following criteria in the *last* record for that admission: 1. Respiratory Rate
> 25 breaths/min, 2. Oxygen saturation < 90%. The model should be trained using the demographic
information (`Age`, `Sex`) and *all physiological measurements recorded up to the penultimate
(second to last) record* for each admission.
--------------------------------------------------------------------------------
# ID: 96
# Dataset: MIMIC-IV
# Task type: Modeling
# Task: Develop a binary classification model to predict, for each hospital admission (identified
by `AdmissionID`), whether the patient's Systolic Blood Pressure ('Systolic blood pressure') will
fall below 90 mmHg at *any point* during that entire admission. The model should utilize patient
demographic information (`Age`, `Sex`) and physiological measurements recorded *only within the
first 24 hours* of that specific admission as input features.
--------------------------------------------------------------------------------
# ID: 97
# Dataset: MIMIC-IV
# Task type: Modeling
# Task: Develop a binary classification model to predict, for each hospital admission (identified
by `AdmissionID`), whether the patient will experience at least one record exhibiting "Severe
Hypotension and Tachycardia" at any point during their entire hospital stay. Define "Severe
Hypotension and Tachycardia" as any record where the Mean Blood Pressure (`Mean blood pressure`)
is less than 60 mmHg AND the Heart Rate (`Heart Rate`) is greater than 100 beats per minute. The
model should be trained using patient demographic information (`Age`, `Sex`) and all physiological
measurements recorded within the first 24 hours of that specific admission as input features.
--------------------------------------------------------------------------------
# ID: 98
# Dataset: MIMIC-IV
# Task type: Data
```

```
# Task: Preprocess the dataset by first imputing missing values in numerical columns using the
column mean across the entire dataset. Then, aggregate the records for each `AdmissionID` into a
single row. For the imputed numerical columns, calculate the mean value across all records
belonging to that `AdmissionID`. For columns that are constant within an `AdmissionID`, retain
their value from any one record for that `AdmissionID`. The final output should have one row per
unique `AdmissionID`.
--------------------------------------------------------------------------------
# ID: 99
# Dataset: MIMIC-IV
# Task type: Data
# Task: Preprocess the dataset by first sorting records by `AdmissionID` and `RecordTime`. Then,
for each `AdmissionID`, impute missing values in numerical columns using forward fill, followed by
backward fill for any remaining NaNs at the start of a sequence. If a column is entirely NaN for
an `AdmissionID`, impute with 0. After imputation, aggregate the records for each `AdmissionID`
into a single row. For the imputed numerical columns, retain the value from the last record within
that `AdmissionID`. For columns that are constant within an `AdmissionID`, retain their value from
any one record for that `AdmissionID`. The final output should have one row per unique
`AdmissionID`.
--------------------------------------------------------------------------------
# ID: 100
# Dataset: MIMIC-IV
# Task type: Data
# Task: Preprocess the dataset by first handling missing values in numerical columns within each
`AdmissionID`. For each numerical column, impute missing values using the mean of the non-missing
values within that specific `AdmissionID`. If a numerical column is entirely missing for a given
`AdmissionID`, impute with 0. After imputation, aggregate the records for each `AdmissionID` into
a single row. For each imputed numerical column, calculate the mean, standard deviation, minimum,
and maximum values across all records belonging to that `AdmissionID`, creating new features. For
columns that are constant within an `AdmissionID`, retain their value from any one record for that
`AdmissionID`. The final output should have one row per unique `AdmissionID`.
```

## G.2 Details in Executing Tasks

We design different prompts for the frameworks. For Single LLM, the prompt is as follows:

> *Prompt for instructing the single LLM to solve tasks.*
>
> ```
> You have been given a task and a dataset, and you need to write a code to solve the given task.
>
> The columns names of the dataset are as follows:
> {{COLUMN_NAMES}}
>
> The task is as follows:
> {{TASK}}
>
> You need to provide the response strictly according to the following requirements:
>
> 1. Only return the code itself, without any explanation or comments;
> 2. Use Python syntax;
> 3. Ensure that the code can run directly.
> 4. Assume the path of the dataset is "{{DATASET_PATH}}".
> 5. The code needs to save the running results to "{{SAVE_PATH}}".
> ```

SmolAgents, OpenManus and Owl share the same prompt:

> *Prompt for instructing multi-agent frameworks to solve tasks.*
>
> ```
> Analyze the document in '{{DATASET_PATH}}', solve the following task:
>
> {{TASK}}
>
> Save the results in the folder '{{SAVE_PATH}}'
> ```

## G.3 Details in Human Evaluation and Results

We develop a comprehensive evaluation protocol to assess the quality and correctness of solutions generated by different multi-agent frameworks. A panel of six PhD/MD students with diverse expertise (3 computer science PhD students in AI for healthcare, 1 MD, 1 biomedical engineering, 1 biostatistics, ensuring clinical validity) are recruited to review the outputs.

Evaluators are provided with detailed assessment guidelines that specify evaluation criteria tailored to each task category:

- Data extraction and statistical analysis: Correctness of data selection, transformation logic, handling of missing values, and appropriateness of statistical methods.

- Predictive modeling: Appropriateness of model selection, implementation of training procedures, inclusion of necessary evaluation metrics, and adherence to proper validation practices.

- Data visualization: Correctness of visualization techniques, alignment with analytical objectives, aesthetic quality, and overall readability.

- Report generation: Completeness, accuracy, coherence of synthesized findings, and clinical relevance of conclusions.

The evaluators independently review each solution and document their assessments, including specific errors or limitations observed.

After individual assessments, we develop a taxonomy of evaluation outcomes by manually reviewing the evaluators' feedback. Initial categories include various error types specific to each task category. To refine this taxonomy, we employ the DeepSeek-V3 to classify the evaluation comments into coherent groups, followed by manual verification and adjustment to ensure consistency.

In cases where evaluators disagree, we conduct a secondary review to resolve conflicts. The final evaluation for each solution represents a consolidated assessment that accounts for all evaluators' perspectives and judgments.

The comprehensive evaluation results across all models and task categories are presented as follows:

---

*Human evaluation results in task 4.*

```
ID: 1
Single LLM: Correct
SmolAgent: Correct
OpenManus: Correct
Owl: Correct
--------------------------------------------------------------------------
ID: 2
Single LLM: Correct
SmolAgent: Correct
OpenManus: Correct
Owl: Correct w/ Presentation Issues
--------------------------------------------------------------------------
ID: 3
Single LLM: Correct
SmolAgent: Correct
OpenManus: Correct
Owl: Correct
--------------------------------------------------------------------------
ID: 4
Single LLM: Correct
SmolAgent: Incorrect Answer
OpenManus: Correct
Owl: No Result
--------------------------------------------------------------------------
ID: 5
Single LLM: Anomalous Numerical Results
SmolAgent: Correct
OpenManus: No Visualization
Owl: Correct
--------------------------------------------------------------------------
ID: 6
Single LLM: Correct
SmolAgent: Correct
OpenManus: Correct
Owl: Correct
--------------------------------------------------------------------------
ID: 7
Single LLM: Correct
SmolAgent: No Visualization
OpenManus: Correct
Owl: Anomalous Numerical Results
--------------------------------------------------------------------------
ID: 8
Single LLM: Correct
```

```
SmolAgent: No Visualization
OpenManus: Correct
Owl: No Visualization
--------------------------------------------------------------------------
ID: 9
Single LLM: No Visualization
SmolAgent: Correct
OpenManus: Poor Readability
Owl: Anomalous Numerical Results
--------------------------------------------------------------------------
ID: 10
Single LLM: Correct
SmolAgent: Correct
OpenManus: Anomalous Numerical Results
Owl: Correct
--------------------------------------------------------------------------
ID: 11
Single LLM: Incorrect Answer
SmolAgent: Correct
OpenManus: Correct
Owl: Correct
--------------------------------------------------------------------------
ID: 12
Single LLM: Incomplete/Partial
SmolAgent: Correct
OpenManus: Correct
Owl: Incomplete/Partial
--------------------------------------------------------------------------
ID: 13
Single LLM: Incomplete/Partial
SmolAgent: Correct
OpenManus: Incorrect Answer
Owl: Incomplete/Partial
--------------------------------------------------------------------------
ID: 14
Single LLM: Incorrect Answer
SmolAgent: Correct
OpenManus: Correct
Owl: No Result
--------------------------------------------------------------------------
ID: 15
Single LLM: Correct
SmolAgent: Correct
OpenManus: Correct
Owl: Incorrect Answer
--------------------------------------------------------------------------
ID: 16
Single LLM: Correct
SmolAgent: Incorrect Answer
OpenManus: Correct
Owl: No Result
--------------------------------------------------------------------------
ID: 17
Single LLM: Viz. Meaningless (Model Fail)
SmolAgent: Viz. Meaningless (Model Fail)
OpenManus: No Visualization
Owl: Viz. Meaningless (Model Fail)
--------------------------------------------------------------------------
ID: 18
Single LLM: Viz. Meaningless (Model Fail)
SmolAgent: Correct
OpenManus: No Visualization
Owl: No Visualization
--------------------------------------------------------------------------
ID: 19
Single LLM: No Visualization
SmolAgent: Viz. Meaningless (Model Fail)
OpenManus: Viz. Only (No Model)
Owl: No Visualization
--------------------------------------------------------------------------
ID: 20
Single LLM: Viz. Only (No Model)
SmolAgent: No Visualization
OpenManus: Viz. Only (No Model)
Owl: No Visualization
--------------------------------------------------------------------------
ID: 21
Single LLM: No Visualization
```

```
SmolAgent: Viz. Only (No Model)
OpenManus: Viz. Only (No Model)
Owl: No Visualization
-------------------------------------------------------------------------
ID: 22
Single LLM: Viz. Only (No Model)
SmolAgent: Viz. Only (No Model)
OpenManus: Correct
Owl: Viz. Only (No Model)
-------------------------------------------------------------------------
ID: 23
Single LLM: No Visualization
SmolAgent: Viz. Meaningless (Model Fail)
OpenManus: Anomalous Numerical Results
Owl: No Visualization
-------------------------------------------------------------------------
ID: 24
Single LLM: Poor Readability
SmolAgent: Clear Presentation
OpenManus: Clear Presentation
Owl: Clear Presentation
-------------------------------------------------------------------------
ID: 25
Single LLM: No Report
SmolAgent: Clear Presentation
OpenManus: Clear Presentation
Owl: No Report
-------------------------------------------------------------------------
ID: 26
Single LLM: Clear Presentation
SmolAgent: Too Simple (w/ Evidence)
OpenManus: Too Simple (w/ Evidence)
Owl: No Report
-------------------------------------------------------------------------
ID: 27
Single LLM: Lacks Conclusion/Summary
SmolAgent: Clear Presentation
OpenManus: Lacks Conclusion/Summary
Owl: No Report
-------------------------------------------------------------------------
ID: 28
Single LLM: No Report
SmolAgent: Poor Readability
OpenManus: Poor Readability
Owl: Too Simple (w/ Evidence)
-------------------------------------------------------------------------
ID: 29
Single LLM: No Report
SmolAgent: Lacks Conclusion/Summary
OpenManus: No Report
Owl: Clear Presentation
-------------------------------------------------------------------------
ID: 30
Single LLM: No Report
SmolAgent: Clear Presentation
OpenManus: Clear Presentation
Owl: No Report
-------------------------------------------------------------------------
ID: 31
Single LLM: No Report
SmolAgent: Lacks Conclusion/Summary
OpenManus: Clear Presentation
Owl: Poor Readability
-------------------------------------------------------------------------
ID: 32
Single LLM: Lacks Conclusion/Summary
SmolAgent: Lacks Conclusion/Summary
OpenManus: No Report
Owl: Too Simple (w/ Evidence)
-------------------------------------------------------------------------
ID: 33
Single LLM: No Report
SmolAgent: Lacks Conclusion/Summary
OpenManus: No Report
Owl: No Report
-------------------------------------------------------------------------
ID: 34
Single LLM: No Report
```

```
SmolAgent: Clear Presentation
OpenManus: No Report
Owl: Poor Readability
--------------------------------------------------------------------------------
ID: 35
Single LLM: No Report
SmolAgent: No Report
OpenManus: Clear Presentation
Owl: No Report
--------------------------------------------------------------------------------
ID: 36
Single LLM: Correct
SmolAgent: Missing Metrics
OpenManus: Correct
Owl: Correct
--------------------------------------------------------------------------------
ID: 37
Single LLM: No Result
SmolAgent: Missing Metrics
OpenManus: Preprocessing Only
Owl: No Result
--------------------------------------------------------------------------------
ID: 38
Single LLM: No Result
SmolAgent: Anomalous Numerical Results
OpenManus: Missing Metrics
Owl: Preprocessing Only
--------------------------------------------------------------------------------
ID: 39
Single LLM: No Result
SmolAgent: Correct
OpenManus: Correct
Owl: No Result
--------------------------------------------------------------------------------
ID: 40
Single LLM: Model Not Saved
SmolAgent: Fails Requirements
OpenManus: Correct
Owl: Preprocessing Only
--------------------------------------------------------------------------------
ID: 41
Single LLM: Fails Requirements
SmolAgent: Fails Requirements
OpenManus: No Result
Owl: Preprocessing Only
--------------------------------------------------------------------------------
ID: 42
Single LLM: Model Not Saved
SmolAgent: Correct
OpenManus: Anomalous Numerical Results
Owl: No Result
--------------------------------------------------------------------------------
ID: 43
Single LLM: No Result
SmolAgent: Missing Metrics
OpenManus: Missing Metrics
Owl: Preprocessing Only
--------------------------------------------------------------------------------
ID: 44
Single LLM: No Result
SmolAgent: Correct
OpenManus: Correct
Owl: No Result
--------------------------------------------------------------------------------
ID: 45
Single LLM: Fails Requirements
SmolAgent: Fails Requirements
OpenManus: Fails Requirements
Owl: Correct
--------------------------------------------------------------------------------
ID: 46
Single LLM: No Result
SmolAgent: Missing Metrics
OpenManus: Correct
Owl: No Result
--------------------------------------------------------------------------------
ID: 47
Single LLM: Model Not Saved
```

```
SmolAgent: Correct
OpenManus: Missing Metrics
Owl: Missing Metrics
-------------------------------------------------------------------------------
ID: 48
Single LLM: Correct
SmolAgent: Correct
OpenManus: Correct
Owl: No Result
-------------------------------------------------------------------------------
ID: 49
Single LLM: Correct
SmolAgent: Correct
OpenManus: Incomplete/Partial
Owl: Correct
-------------------------------------------------------------------------------
ID: 50
Single LLM: Correct
SmolAgent: No Result
OpenManus: Correct
Owl: No Result
-------------------------------------------------------------------------------
ID: 51
Single LLM: Correct
SmolAgent: Correct
OpenManus: Incorrect Answer
Owl: Correct
-------------------------------------------------------------------------------
ID: 52
Single LLM: Correct
SmolAgent: Incorrect Answer
OpenManus: Correct
Owl: Correct
-------------------------------------------------------------------------------
ID: 53
Single LLM: Correct
SmolAgent: Correct
OpenManus: Correct
Owl: Correct
-------------------------------------------------------------------------------
ID: 54
Single LLM: Anomalous Numerical Results
SmolAgent: Anomalous Numerical Results
OpenManus: Anomalous Numerical Results
Owl: No Visualization
-------------------------------------------------------------------------------
ID: 55
Single LLM: Correct
SmolAgent: Correct
OpenManus: Correct
Owl: Correct
-------------------------------------------------------------------------------
ID: 56
Single LLM: Anomalous Numerical Results
SmolAgent: Anomalous Numerical Results
OpenManus: Anomalous Numerical Results
Owl: Anomalous Numerical Results
-------------------------------------------------------------------------------
ID: 57
Single LLM: Correct
SmolAgent: No Visualization
OpenManus: Correct
Owl: Anomalous Numerical Results
-------------------------------------------------------------------------------
ID: 58
Single LLM: Poor Readability
SmolAgent: Poor Readability
OpenManus: Anomalous Numerical Results
Owl: Poor Readability
-------------------------------------------------------------------------------
ID: 59
Single LLM: Anomalous Numerical Results
SmolAgent: Anomalous Numerical Results
OpenManus: Anomalous Numerical Results
Owl: Anomalous Numerical Results
-------------------------------------------------------------------------------
ID: 60
Single LLM: Correct
```

```
SmolAgent: Correct
OpenManus: Correct
Owl: Correct
--------------------------------------------------------------------------------
ID: 61
Single LLM: Correct
SmolAgent: Correct
OpenManus: Correct
Owl: Correct
--------------------------------------------------------------------------------
ID: 62
Single LLM: Correct
SmolAgent: Correct
OpenManus: Correct
Owl: Correct
--------------------------------------------------------------------------------
ID: 63
Single LLM: Correct
SmolAgent: Correct
OpenManus: Incorrect Answer
Owl: Incorrect Answer
--------------------------------------------------------------------------------
ID: 64
Single LLM: No Result
SmolAgent: No Result
OpenManus: No Result
Owl: No Result
--------------------------------------------------------------------------------
ID: 65
Single LLM: Correct
SmolAgent: Correct
OpenManus: Correct
Owl: Incomplete/Partial
--------------------------------------------------------------------------------
ID: 66
Single LLM: No Visualization
SmolAgent: Correct
OpenManus: Viz. Only (No Model)
Owl: No Visualization
--------------------------------------------------------------------------------
ID: 67
Single LLM: Info Not Extractable
SmolAgent: Info Not Extractable
OpenManus: Info Not Extractable
Owl: No Visualization
--------------------------------------------------------------------------------
ID: 68
Single LLM: No Visualization
SmolAgent: Anomalous Numerical Results
OpenManus: Anomalous Numerical Results
Owl: No Visualization
--------------------------------------------------------------------------------
ID: 69
Single LLM: No Visualization
SmolAgent: Info Not Extractable
OpenManus: No Visualization
Owl: No Visualization
--------------------------------------------------------------------------------
ID: 70
Single LLM: No Visualization
SmolAgent: Poor Readability
OpenManus: Viz. Meaningless (Model Fail)
Owl: No Visualization
--------------------------------------------------------------------------------
ID: 71
Single LLM: No Visualization
SmolAgent: Viz. Meaningless (Model Fail)
OpenManus: Viz. Meaningless (Model Fail)
Owl: No Visualization
--------------------------------------------------------------------------------
ID: 72
Single LLM: Lacks Conclusion/Summary
SmolAgent: Anomalous Numerical Results
OpenManus: Too Simple (w/ Evidence)
Owl: Anomalous Numerical Results
--------------------------------------------------------------------------------
ID: 73
Single LLM: No Report
```

```
SmolAgent: Anomalous Numerical Results
OpenManus: Clear Presentation
Owl: No Report
--------------------------------------------------------------------------------
ID: 74
Single LLM: No Report
SmolAgent: Lacks Conclusion/Summary
OpenManus: Poor Readability
Owl: Lacks Conclusion/Summary
--------------------------------------------------------------------------------
ID: 75
Single LLM: Lacks Conclusion/Summary
SmolAgent: Lacks Conclusion/Summary
OpenManus: Lacks Conclusion/Summary
Owl: Clear Presentation
--------------------------------------------------------------------------------
ID: 76
Single LLM: No Report
SmolAgent: Lacks Conclusion/Summary
OpenManus: Poor Readability
Owl: No Report
--------------------------------------------------------------------------------
ID: 77
Single LLM: No Report
SmolAgent: No Report
OpenManus: No Report
Owl: No Report
--------------------------------------------------------------------------------
ID: 78
Single LLM: No Report
SmolAgent: No Report
OpenManus: No Report
Owl: No Report
--------------------------------------------------------------------------------
ID: 79
Single LLM: Lacks Conclusion/Summary
SmolAgent: Lacks Conclusion/Summary
OpenManus: Lacks Conclusion/Summary
Owl: Clear Presentation
--------------------------------------------------------------------------------
ID: 80
Single LLM: Lacks Conclusion/Summary
SmolAgent: Lacks Conclusion/Summary
OpenManus: Clear Presentation
Owl: No Report
--------------------------------------------------------------------------------
ID: 81
Single LLM: Lacks Conclusion/Summary
SmolAgent: Lacks Conclusion/Summary
OpenManus: Clear Presentation
Owl: No Report
--------------------------------------------------------------------------------
ID: 82
Single LLM: No Report
SmolAgent: Lacks Conclusion/Summary
OpenManus: Clear Presentation
Owl: No Report
--------------------------------------------------------------------------------
ID: 83
Single LLM: No Report
SmolAgent: Lacks Conclusion/Summary
OpenManus: Lacks Conclusion/Summary
Owl: No Report
--------------------------------------------------------------------------------
ID: 84
Single LLM: Lacks Conclusion/Summary
SmolAgent: Clear Presentation
OpenManus: Lacks Conclusion/Summary
Owl: No Report
--------------------------------------------------------------------------------
ID: 85
Single LLM: Model Not Saved
SmolAgent: Correct
OpenManus: Correct
Owl: No Result
--------------------------------------------------------------------------------
ID: 86
Single LLM: Model Not Saved
```

```
SmolAgent: Fails Requirements
OpenManus: Anomalous Numerical Results
Owl: Preprocessing Only
--------------------------------------------------------------------------
ID: 87
Single LLM: Fails Requirements
SmolAgent: Fails Requirements
OpenManus: Preprocessing Only
Owl: No Result
--------------------------------------------------------------------------
ID: 88
Single LLM: No Result
SmolAgent: Missing Metrics
OpenManus: Missing Metrics
Owl: Preprocessing Only
--------------------------------------------------------------------------
ID: 89
Single LLM: Model Not Saved
SmolAgent: Fails Requirements
OpenManus: Missing Metrics
Owl: No Result
--------------------------------------------------------------------------
ID: 90
Single LLM: Anomalous Numerical Results
SmolAgent: Correct
OpenManus: Fails Requirements
Owl: No Result
--------------------------------------------------------------------------
ID: 91
Single LLM: No Result
SmolAgent: Missing Metrics
OpenManus: No Result
Owl: No Result
--------------------------------------------------------------------------
ID: 92
Single LLM: Model Not Saved
SmolAgent: No Result
OpenManus: Preprocessing Only
Owl: No Result
--------------------------------------------------------------------------
ID: 93
Single LLM: Anomalous Numerical Results
SmolAgent: Anomalous Numerical Results
OpenManus: Anomalous Numerical Results
Owl: No Result
--------------------------------------------------------------------------
ID: 94
Single LLM: Model Not Saved
SmolAgent: Correct
OpenManus: Correct
Owl: No Result
--------------------------------------------------------------------------
ID: 95
Single LLM: Model Not Saved
SmolAgent: Missing Metrics
OpenManus: Missing Metrics
Owl: Preprocessing Only
--------------------------------------------------------------------------
ID: 96
Single LLM: Model Not Saved
SmolAgent: No Result
OpenManus: Anomalous Numerical Results
Owl: No Result
--------------------------------------------------------------------------
ID: 97
Single LLM: Anomalous Numerical Results
SmolAgent: Anomalous Numerical Results
OpenManus: No Result
Owl: No Result
--------------------------------------------------------------------------
ID: 98
Single LLM: Correct
SmolAgent: Correct
OpenManus: Correct
Owl: No Result
--------------------------------------------------------------------------
ID: 99
Single LLM: No Result
```

```
    SmolAgent: Correct
    OpenManus: No Result
    Owl: No Result
    -------------------------------------------------------------------
    ID: 100
    Single LLM: Correct
    SmolAgent: Correct
    OpenManus: No Result
    Owl: No Result
```

# H  More In-Depth Analysis of Task 4

## H.1  Illustrative Examples: Success vs. Failure Cases

To provide a concrete understanding of performance differences, we present representative examples from our evaluation. These cases highlight how success often hinges on the agent's ability to correctly parse and execute multi-step logic, whereas failure frequently stems from fundamental misinterpretations of the task requirements.

**Case 1: Data Statistics (Task ID: 11)**

- **Task:** Calculate the average range of 'White blood cell count' for patients with at least two records. The range is the difference between the maximum and minimum values for each patient.

- **Reference Answer:** "23.357969..."

- **Good Output (SmolAgent):** The agent correctly implemented the logic (filtering, grouping, max-min calculation, and averaging) and produced the exact result.

    ```
    23.357969348659005
    ```

- **Failure Case (Single LLM, Judgment: Incorrect Answer):** The agent generated a script with flawed logic (e.g., incorrect patient filtering), leading to an incorrect numerical result.

    ```
    18.991993769470405
    ```

**Case 2: Report Generation (Task ID: 31)**

- **Task:** Analyze the relationship between lab measurement frequency and patient outcome, selecting relevant parameters and generating a structured summary report with findings and interpretations.

- **Reference Answer:** A concise, structured markdown report with clear sections and distilled insights.

> *Reference answer.*
>
> ```
> # Analysis Report: Measurement Density and Patient Outcome
> ## Objective
> Investigate the relationship between lab measurement frequency and patient outcome.
> ## Methodology
> 1. Selected Parameters: Hemoglobin, White Blood Cell Count...
> 2. Analysis: Mann-Whitney U test to compare measurement density...
> ## Key Findings
> - All selected parameters showed significant differences (p < 0.001)...
> ## Interpretations
> - Higher measurement density in deceased patients may indicate more intensive monitoring...
> ```

- **Failure Case (SmolAgent, Judgment: Lacks Conclusion/Summary):** The agent performed statistical tests on all available parameters but failed to synthesize the results. Instead of a structured report, it produced a verbose data dump of all 70+ parameters without selection, structure, or interpretation. This fails the task's core requirement of reasoning and synthesis.

## H.2 Granular Error Analysis for Task 4

Our evaluation reveals that high-level categories often mask distinct root causes. For instance, errors categorized as "Fails Requirements" in the modeling task were not uniform; we found that 73% of these cases stemmed from a fundamental misinterpretation of the analytical goal, whereas the remaining 27% were due to downstream procedural errors during implementation, even when the initial goal was understood. Similarly, visualizations deemed "Meaningless (Model Fail)" could be traced back to two primary sources: 78% were rooted in the agent misunderstanding the initial prompt, which invalidated the entire modeling process, while 22% arose from correct task understanding followed by a specific invalid operation during modeling. We also quantified "Poor Readability" in visualizations, distinguishing between severe failures (60% of cases), such as obscured legends that rendered a chart uninterpretable, and minor cosmetic issues (40%), like missing titles. This detailed analysis highlights that fundamental reasoning and task comprehension, rather than simple code execution, are the primary bottlenecks for current agents.

## H.3 Deeper Analysis of Success/Failure Dynamics

The performance difference between workflow automation and VQA is rooted in "task decomposability" and "information fidelity" across modalities. "Workflow automation" succeeds because it can be broken down into discrete, logical sub-tasks (e.g., load data, run stats, create plot) that align with agent specialization and tool use. "Medical VQA", in contrast, is often a monolithic, perception-bound task where decomposition is artificial. More critically, there is a fundamental information bottleneck: when rich visual information is translated into text for inter-agent communication, critical details are lost. Collaboration cannot recover information that was lost in translation. We illustrate this with two case studies.

**Case Study 1: Success in Workflow Automation (Task ID=66, SmolAgents)**    A multi-agent system successfully executed a request to build a readmission prediction model.

- **Task:**

  *Task.*

  ```
  Build a predictive model to determine the likelihood of a patient being readmitted.
  1. `Data Preparation`: For each unique `AdmissionID`, extract and process relevant features from
  the time-series data. This could involve aggregating the time-series measurements. Combine these
  aggregated features with static patient/admission information ('Age', 'Sex', 'LOS', 'Outcome')
  to create a single feature vector per admission.
  2. `Missing Value Handling`: Address the missing values ('nan') in the dataset.
  3. `Model Training`: Train a binary classification model on the prepared admission-level dataset
  to predict the `Readmission` outcome.
  4. `Evaluation and Visualization`: Evaluate your model's performance using appropriate metrics
  for binary classification. Then, create a visualization that helps interpret the model or its
  results.
  ```

- **Analysis:** The task has a "Natural Decomposition" into a logical pipeline: load data, clean data, train model, evaluate, and plot results. The framework used "Effective Role Assignment & Tool Use": a "Data Engineer" agent used `pandas`, an "ML Scientist" agent used `scikit-learn`, and a "Data Visualization Specialist" agent used `matplotlib`. The task's structure was perfectly suited for this modular, tool-driven approach.

**Case Study 2: Failure in Medical VQA (MedAgent, VQA-RAD dataset)**    A team of AI agents was asked to analyze a brain MRI.

- **The Scene:** A multi-disciplinary team of AI agents, role-playing as a Pediatrician, a Cardiologist, and a Pulmonologist, convenes to analyze a patient's brain MRI.

- **The Question:** "In which brain area is the lesion located?"

- **The Ground Truth:** "Right cerebellopontine angle" (a highly specific anatomical junction).

- **The Collaboration History:** The collaboration devolved into an echo chamber. Generalist agents identified a broad region ("posterior fossa"). A specialist (Cardiologist) offered a more specific but still incorrect answer ("right cerebellar hemisphere"), while admitting "I do not have the specialized knowledge to accurately assess or localize lesions in the brain." The final synthesized answer incorrectly adopted the specialist's confident-sounding but flawed opinion.

- **Analysis:**
  (1) **The Perception Bottleneck is the Root Cause:** The core error lies in the underlying VLM's lack of anatomical precision. It correctly identified the general region but failed to distinguish the fine-grained "Right cerebellopontine angle." No amount of subsequent textual debate can rectify this initial perceptual shortfall.
  (2) **Collaboration Becomes an Echo Chamber:** The multi-agent interaction log reveals that the discussion merely revolved around the initial imprecise terms. The framework acted not as a tool for refinement, but as an echo chamber for the initial inaccurate analysis.
  (3) **Artificial Decomposition Proves Ineffective:** The framework's attempt to decompose the task by assigning roles like "Cardiologist" to interpret a neuroradiology image proved counter-productive. This contrasts sharply with workflow automation, where decomposing a task into "load data" and "run statistics" aligns perfectly with distinct, functional skills.

These cases show that for holistic, perception-heavy tasks, collaboration can amplify initial inaccuracies, whereas for decomposable, tool-based tasks, it can be highly effective.

## H.4 Deeper Investigation of Multi-Agent Performance

Our results highlight a significant inconsistency in the performance of different multi-agent systems across tasks, a phenomenon rooted in their architectural assumptions and practical limitations. For instance, some frameworks' performance can be constrained by their weakest base model; ReConcile's effectiveness in VQA is hindered when a highly capable model like `Qwen-VL-Max` is paired with less advanced models. Furthermore, many frameworks that rely on role-playing, such as MDAgents, ColaCare, and MedAgents, often fail to elicit genuine domain-specific knowledge, as the underlying LLMs may not effectively adopt the nuanced expertise of a specified role (e.g., an internist versus a surgeon). In tool-use scenarios, frameworks like SmolAgent, OpenManus, and Owl frequently exhibit execution failures, including irrational tool calls, inaccurate file path recognition, or getting stuck in loops. More broadly, multi-agent frameworks often suffer from technical fragility, such as frequent `JSON` parsing errors that require manual intervention, and exhibit high sensitivity to prompt design, making their performance unpredictable and difficult to stabilize. These factors collectively indicate why the theoretical benefits of multi-agent collaboration do not always translate into consistent, practical advantages.

