# OpenReview forum: "MedAgentBoard: Benchmarking Multi-Agent Collaboration with Conventional Methods for Diverse Medical Tasks"
_NeurIPS.cc/2025/Datasets_and_Benchmarks_Track — NeurIPS 2025 Datasets and Benchmarks Track poster_

### Official Review · Reviewer_Hr7A · 2025-06-22

**Rating:** 4
**Confidence:** 3

**Summary:**

The paper introduces MedAgentBoard, a benchmark for evaluating multi-agent collaboration, single LLMs, and conventional methods across medical tasks. The core contribution is a systematic comparison showing multi-agent approaches offer niche benefits (e.g., task completeness in workflows) but do not universally outperform single LLMs or specialized conventional models. The benchmark covers diverse data modalities and includes rigorous baselines, with code/datasets open-sourced.

**Dataset Code Accessibility:**

Partly

**Ethical Considerations:**

No, there are no or only very minor ethics concerns

**Final Justification:**

I think current version still need a minor revisions.

**Limitations Weaknesses:**

1. The evaluated multi-agent frameworks show marginal improvements over single LLMs, often failing to justify their complexity. For instance, in EHR prediction, multi-agent approaches trail conventional models by ~10-15% in AUROC (§1-156, Table 5).
2. While diverse, tasks like clinical workflow automation focus on structured data analysis but lack real-time clinical decision support scenarios (e.g., emergency triage). This limits generalizability to dynamic healthcare settings.
3. The paper does not explore why multi-agent collaboration succeeds in workflow automation (e.g., task decomposition) but fails in VQA. A deeper analysis of interaction dynamics (e.g., role assignment efficiency) is missing.
4. In workflow automation, "task completeness" prioritizes output generation over clinical validity. For example, OpenManus achieves 41.67% "Correct" rates, but many solutions have "anomalous numerical results" (§1-169, Table 6).

**Strengths Contributions:**

1. MedAgentBoard addresses prior gaps by integrating four task categories (text, image, structured data) and comparing multi-agent, single LLM, and conventional methods. This breadth reflects real-world clinical diversity (§1-19, §1-54).
2. The study reveals context-dependent performance: multi-agent excels in workflow automation but lags in medical VQA/EHR prediction (§1-124, §1-158). This challenges the "one-size-fits-all" narrative for multi-agent systems.
3. The benchmark uses established datasets (MIMIC-IV, MedQA) and metrics (AUROC, ROUGE), with clear implementation details (Appendix B). Open-sourcing all assets enhances reproducibility (§1-6, §1-313).

---

> ### Author Rebuttal · Authors · 2025-07-30
>
> We thank you for your valuable feedback and for acknowledging the strengths of our work, including its comprehensive scope, nuanced findings, and commitment to reproducibility. We appreciate the opportunity to address the identified weaknesses.
>
> ## 1. On Marginal Improvements and Performance Gaps (W-1)
>
> We thank the reviewer for this accurate observation. MedAgentBoard was designed precisely to provide a rigorous, data-driven counterpoint to the prevailing narrative that more complex multi-agent systems are inherently superior. Our key takeaway is that their utility must be carefully weighed against their complexity and that in many critical medical domains, simpler and more specialized models remain superior. This finding challenges the community to justify the overhead of multi-agent systems with tangible, significant performance gains.
>
> ## 2. On Limited Generalizability to Dynamic Settings (W-2)
>
> Our initial benchmark focuses on foundational analytical workflows using retrospective data, which is a necessary first step. We recognize that dynamic, real-time clinical scenarios are an important direction for future work.
>
> We are already taking steps in this direction. We have curated a much larger dataset of over 1,000 EHR tasks that covers a broader range of medical scenarios (including tasks related to emergency triage) and data types (ICD codes, clinical notes, structured data). We are actively working on expanding our benchmark to include these more dynamic and diverse settings. We will describe the details of this ongoing work in the updated manuscript and plan to open-source the new datasets and benchmark results, further enhancing the generalizability of MedAgentBoard.
>
> ## 3. On Deeper Analysis of Success/Failure Dynamics (W-3)
>
> We thank the reviewer for pushing for a deeper "why." The performance difference between workflow automation and VQA is rooted in `task decomposability` and `information fidelity` across modalities.
>
> - `Workflow automation` succeeds because it can be broken down into discrete, logical sub-tasks (e.g., load data, run stats, create plot) that align perfectly with agent specialization and tool use (Python execution).
> - `Medical VQA`, in contrast, is often a monolithic, perception-bound task. Forcing a decomposition is artificial and counterproductive. More critically, there is a fundamental information bottleneck: when rich visual information is translated into text for inter-agent communication, critical spatial relationships and fine-grained details are lost. Collaboration cannot recover information that was lost in translation.
>
> We will add detailed and more case studies to the revised manuscript to illustrate this.
>
> > Case Study 1 (Task ID=66，smolagents): Success in Workflow Automation (Task: Readmission Prediction): A multi-agent system successfully executed a request to build a readmission prediction model.
>
> ```
> Build a predictive model to determine the likelihood of a patient being readmitted.
> 1. `Data Preparation`: For each unique `AdmissionID`, extract and process relevant features from the time-series data. This could involve aggregating the time-series measurements. Combine these aggregated features with static patient/admission information ('Age', 'Sex', 'LOS', 'Outcome') to create a single feature vector per admission.
> 2. `Missing Value Handling`: Address the missing values ('nan') in the dataset.
> 3. `Model Training`: Train a binary classification model on the prepared admission-level dataset to predict the `Readmission` outcome.
> 4. `Evaluation and Visualization`: Evaluate your model's performance using appropriate metrics for binary classification. Then, create a visualization that helps interpret the model or its results.
> ```
>
> The case shows:
> - `Natural Decomposition`: The goal was broken down into a logical pipeline: load data, clean data, train model, evaluate, and plot results.
> - `Effective Role Assignment & Tool Use`: The framework assigned specific roles. A "Data Engineer" agent used `pandas` to prepare the data. An "ML Scientist" agent then used `scikit-learn` to train a classifier. Finally, a "Data Visualization Specialist" agent used `matplotlib` to plot the ROC curve. The task's structure was perfectly suited for this modular, tool-driven approach.
>
> > Case Study 2: Failure in Medical VQA (Task: Lesion Localization) MedAgent, VQA-RAD dataset, free-form setting
>
> ```
> The Scene: A multi-disciplinary team of AI agents, role-playing as a Pediatrician, a Cardiologist, and a Pulmonologist, convenes to analyze a patient's brain MRI.
> The Question: "In which brain area is the lesion located?"
> The Ground Truth: "Right cerebellopontine angle" (a highly specific anatomical junction).
>
> The Collaboration History:
> Round 1 - Initial Opinions:
> Pediatrician & Pulmonologist (Generalists): Both agents correctly identify the lesion in the "posterior fossa," a broad region, suggesting it could be in the "cerebellum or the brainstem." Their diagnosis is cautious and general.
> Cardiologist (Specialist): More assertively, this agent pinpoints the location as the "right cerebellar hemisphere."
>
> Round 1 - First Review: A crucial moment of self-reflection occurs.
> The Cardiologist agent candidly states: "As a cardiologist, my expertise is in the heart... not in neuroimaging... I do not have the specialized knowledge to accurately assess or localize lesions in the brain." It suggests deferring to a neurologist.
> Despite this, the other agents proceed.
>
> Round 2 - Second Discussion: The conversation circles back to the same general terms. The agents re-state their initial opinions, with no new, more precise anatomical details emerging from the image analysis. The discussion remains stuck at the "cerebellum" and "posterior fossa" level.
>
> The Final Decision: The collaborative framework synthesizes the discussion. Despite the Cardiologist's earlier disclaimer, its more specific (but still incorrect) answer heavily influences the outcome.
> Final Explanation: "This conclusion is based on the consensus... and the cardiologist's suggestion of a more specific location within the right cerebellar hemisphere."
> Final Incorrect Answer: "Right cerebellar hemisphere"
> ```
>
> The case shows a team of AI agents (role-playing as a pediatrician, cardiologist, and pulmonologist) was asked to analyze a brain MRI. The collaboration process exhibits:
>
> 1. The Perception Bottleneck is the Root Cause: The core error lies in the lack of anatomical precision. The underlying VLM correctly identified the general region ("posterior fossa") but failed to distinguish the fine-grained "Right cerebellopontine angle." This is a classic example of a general-purpose model's limitation in a specialized domain. No amount of subsequent textual debate can rectify this initial perceptual shortfall. As the agents' dialogue shows, their entire discussion was anchored to these imprecise initial terms, proving that collaboration cannot create knowledge that isn't present in the source data.
> 2. Collaboration Becomes an Echo Chamber, Not a Refinement Engine: The multi-agent interaction log reveals that the two-round discussion merely "revolved around the posterior fossa, possibly involving the cerebellum or brainstem." The agents debated these general locations, but the dialogue never converged on a more precise anatomical structure. This demonstrates that for atomic, perception-heavy tasks, the collaborative framework acted not as a tool for refinement, but as an echo chamber for the initial imprecise analysis. The final decision simply adopted the most confident-sounding opinion ("right cerebellar hemisphere"), even though it was still incorrect.
> 3. Artificial Decomposition Proves Ineffective and Misleading: The framework’s attempt to decompose the task by assigning roles like "Cardiologist" to interpret a neuroradiology image proved counterproductive. The most insightful comment in the entire exchange came from the Cardiologist agent itself: "As a cardiologist... I do not have the specialized knowledge to accurately assess or localize lesions in the brain." This admission highlights the absurdity of such role-playing for a holistic, non-decomposable task like VQA. The framework, however, ignored this critical piece of self-assessment and ultimately gave weight to the Cardiologist's opinion in its final synthesis. This contrasts sharply with workflow automation, where decomposing a task into "load data" and "run statistics" aligns perfectly with distinct, functional skills.
>
> The two cases show that for holistic, perception-heavy tasks, collaboration can become an echo chamber for initial inaccuracies, whereas for decomposable, tool-based tasks, it can be highly effective.
>
> ## 4. On Clinical Validity vs. "Task Completeness" (W-4)
>
> We wholeheartedly agree. Ensuring clinical validity, not just technical correctness or task completion, is the ultimate and most important standard for any medical AI system.
>
> This principle was the primary motivation for our recent changes to the evaluation protocol. We assembled an `expert panel with clinical, biostatistical, and biomedical engineering backgrounds` to re-evaluate all Task 4 submissions. Their mandate was to assess not just if a task was completed, but if the solution was methodologically sound, the results were clinically plausible, and the conclusions were relevant. This effort is transforming our evaluation from a purely technical assessment to a clinically-grounded analysis that better serves the goal of developing AI for real-world medical applications. We will emphasize this shift in the revised manuscript.

---

> ### Comment · Reviewer_Hr7A · 2025-08-01
> **Official Comment by Reviewer Hr7A**
>
> Thank you for the efforts and responses. I would like to keep my score.

---

> ### Author Response · Authors · 2025-08-07
>
> Dear Reviewer Hr7A,
>
> Thank you for your time and for your thoughtful response to our rebuttal. We are glad to have had the opportunity to address the points you raised.
>
> Your feedback was invaluable in pushing us to think more deeply about our contributions. Specifically, your questions about the underlying reasons for performance differences (the "why") and the importance of clinical validity prompted us to conduct a much deeper qualitative analysis. The new case studies contrasting VQA and workflow automation, along with the re-evaluation by our expanded expert panel, were directly inspired by your review and have, we feel, significantly enriched the paper's insights.
>
> We wanted to politely check if these extensive revisions and new analyses have helped clarify the value of our work, and if there are any remaining concerns or further suggestions you might have. We are eager to ensure the final manuscript is as strong as possible.
>
> Thank you once again for your constructive engagement.
>
> Warmest regards,
>
> All Authors of Submission #436

---

### Official Review · Reviewer_taNB · 2025-07-02

**Rating:** 4
**Confidence:** 4

**Summary:**

The paper introduces MedAgentBoard, a comprehensive benchmark designed to systematically evaluate and compare the performance of multi-agent LLM collaboration, single-LLM approaches, and traditional machine learning methods across a diverse range of medical tasks. Previous evaluations of multi-agent systems in medicine have suffered from a lack of generalizability and incomplete comparisons, often focusing on narrow tasks like multiple-choice question answering while omitting strong conventional baselines.

**Dataset Code Accessibility:**

Yes

**Ethical Considerations:**

No, there are no or only very minor ethics concerns

**Final Justification:**

Thank you for the response. All my previous concerns have been solved point-by-point and I would like to keep my overall positive score.

**Limitations Weaknesses:**

- The novelty of this paper is unclear. The datasets and experiments are obtained from existing papers. Similar observations that multi-agent system will not work in all scenarios on all tasks is also not news. I am wondering if the authors can summarize the key novelty design in the work, making it different from repeating experiments mechanically.
- I think aside from the performance, another thing we are curious about is the cost comparison. It would be better if the authors can do some cost analysis between different multi-agent methods, or even with non multi-agent methods.
- The Clinical Workflow Automation (Task 4) evaluation, while commendably using human assessment, was conducted by "three Computer Science PhD students". Although the paper notes their experience in "health data analysis", this cohort may not fully represent the clinical and statistical expertise of the target end-users (e.g., clinicians, biostatisticians, epidemiologists). The evaluation criteria, such as "appropriateness of model selection" and "clinical relevance of conclusions", have deep clinical and methodological implications that may be best judged by domain experts.
- The paper's conclusions about multi-agent collaboration are intrinsically tied to the performance of the specific, mostly general-purpose frameworks evaluated (e.g., MedAgents, SmolAgents, OpenManus). The finding that these systems are not always superior could be a reflection of the limitations of these current frameworks rather than a fundamental weakness in the concept of multi-agent collaboration for medicine. The paper acknowledges this by noting that "current QA-style collaborative frameworks may not be optimally suited for structured data prediction".
- The evaluation results for Task 4 in Table 6 are presented as aggregated percentages in discrete categories like "Correct," "Missing Metrics," or "Incorrect Answer". This format, while clear, lacks granularity. It does not distinguish between minor, easily correctable errors (e.g., missing a plot title) and critical failures (e.g., using the wrong statistical test or misinterpreting the task entirely). An entry categorized under "Missing Metrics" could be an otherwise perfect analysis or a fundamentally flawed one.

**Strengths Contributions:**

- The paper is well written and easy-to-follow. Good visualizations help readers to comprehend the authors' design and insights provided by experimental results.
- The paper addresses a critical and relevant question: when are complex multi-agent systems genuinely beneficial in medicine compared to simpler LLMs or eastablished conventional methods?. As the medical  field increasingly explores LLMs, this work provides a necessary reality check, guiding researchers and practitioners toward more effective and efficient solutions. Its findings—that multi-agent systems are not universally superior and that conventional models excel in key areas like EHR prediction and VQA—are significant for preventing the misallocation of resources toward unnecessarily complex models.
- The benchmark is not confined to a single task type like multiple-choice QA. Instead, it spans four diverse and clinically relevant categories: (visual) question answering, lay summary generation, EHR predictive modeling, and clinical workflow automation, covering various data modalities. This comprehensive scope directly addresses the lack of generalizability in prior evaluations.

---

> ### Author Rebuttal · Authors · 2025-07-30
>
> We thank you for the insightful comments and for recognizing the clarity of our paper and the importance of the research question we address. We appreciate the chance to elaborate on novelty, cost, and evaluation rigor.
>
> ## 1. On the Novelty of MedAgentBoard (W-1)
>
> The novelty of MedAgentBoard is not that multi-agent systems have limitations, but that it is the **first benchmark to systematically adjudicate the deeply conflicting claims in the current literature**. The medical AI community faces a paradox, with some top-tier publications reporting significant gains from multi-agent systems while others find their benefits minimal.
>
> |Paper|Venue|Key Claim / Finding|
> |:--|:--|:--|
> |`Pro-Collaboration Viewpoint`|||
> |MedAgents (Tang et al.) [1]|ACL 2024|"...excels at mining and harnessing medical expertise within LLMs."|
> |ReConcile (Chen et al.) [2]|ACL 2024|"Surpassing prior single-agent and multi-agent baselines by up to 11.4%."|
> |ECON (Saha et al.) [3]|ICML 2025|"Outperforms existing multi-LLM approaches by 11.2% on average..."|
> |DPSDP (Li et al.) [4]|ICML 2025|"An ablation study further confirms the benefits of multi-agent collaboration."|
> |`Countervailing / Nuanced Viewpoint`|||
> |Cemri et al. [5]|arXiv 2025|"Performance gains often remain minimal... high failure rates."|
> |Zhang et al. [6]|arXiv 2025|"...fail to reliably outperform simple single-agent baselines."|
> |`MedAgentBoard`|Ours|Provides the first, comprehensive, unified benchmark to systematically adjudicate these conflicting claims across diverse medical tasks and methods.|
>
> [1] Tang, Xiangru, et al. "MedAgents: Large Language Models as Collaborators for Zero-shot Medical Reasoning." Findings of the Association for Computational Linguistics ACL 2024. 2024.
>
> [2] Chen, Justin, Swarnadeep Saha, and Mohit Bansal. "ReConcile: Round-Table Conference Improves Reasoning via Consensus among Diverse LLMs." Proceedings of the 62nd Annual Meeting of the Association for Computational Linguistics (Volume 1: Long Papers). 2024.
>
> [3] Yi, Xie, et al. From Debate to Equilibrium: Belief‑Driven Multi‑Agent LLM Reasoning via Bayesian Nash Equilibrium. Proceedings of the 42nd International Conference on Machine Learning. 2025.
>
> [4] Yuan, Yurun, et al. Reinforce LLM Reasoning through Multi-Agent Reflection. Proceedings of the 42nd International Conference on Machine Learning. 2025.
>
> [5] Cemri, Mert, et al. "Why do multi-agent llm systems fail?." arXiv preprint arXiv:2503.13657 (2025).
>
> [6] Zhang, Hangfan, et al. "Stop Overvaluing Multi-Agent Debate--We Must Rethink Evaluation and Embrace Model Heterogeneity." arXiv preprint arXiv:2502.08788 (2025).
>
> This contradiction arises because prior work is often limited to narrow task sets and omits comparisons to strong, conventional non-LLM baselines. MedAgentBoard resolves this conflict by:
> 1. `Unifying Evaluation`: Providing a unified platform to test all three paradigms (conventional, single-LLM, multi-agent) across diverse tasks (QA, generation, prediction, workflow) and modalities (text, image, structured data).
> 2. `Including Critical Baselines`: Demonstrating that for key medical tasks like EHR prediction and VQA, specialized conventional methods still dominate—a crucial finding absent from most agent-focused evaluations.
>
> Our work provides a timely, evidence-based guide for researchers to understand where multi-agent systems offer tangible benefits in medicine.
>
> ## 2. Provide Additional Cost Analysis (W-2)
>
> As requested, we provide a cost analysis based on collaboration rounds and estimated API costs.
>
> > Table: Average Number of Discussion Rounds Per Question
>
> |Framework|MedQA|PubMedQA|PubMedQA|PathVQA|VQA-RAD|VQA-RAD|
> |:-:|:-:|:-:|:-:|:-:|:-:|:-:|
> ||Multiple Choice|Multiple Choice|Free-Form|Multiple Choice|Multiple Choice|Free-Form|
> |ColaCare|1.20|1.06|1.03|1.06|1.16|1.23|
> |MDAgents|0.81|0.88|0.83|0.88|0.035|0.39|
> |MedAgents|1.23|1.23|1.07|1.23|1.42|1.89|
> |ReConcile|1.14|1.20|2.30|1.20|1.15|1.98|
>
> ReConcile has the most discussion rounds, as its use of diverse base models for each agent often leads to divergent opinions that require more rounds to resolve. In contrast, MDAgents' difficulty-gating mechanism reduces them by using a single agent for simpler questions (discussion round = 0).
>
> > Table: Estimated Cost (USD, DeepSeek pricing strategy) on Selected Test Set
>
> |Method|MedQA|PubMedQA|PubMedQA|PathVQA|VQA-RAD|VQA-RAD|
> |:--|:-:|:-:|:-:|:-:|:-:|:-:|
> ||Multiple Choice|Multiple Choice|Free-Form|Multiple Choice|Multiple Choice|Free-Form|
> |ColaCare|2.64|4.05|5.99|1.47|1.19|1.51|
> |MDAgents|2.13|4.15|4.85|0.79|0.27|0.72|
> |MedAgents|2.71|4.32|5.52|2.34|1.54|2.23|
> |ReConcile|3.05|5.29|9.32|1.89|1.85|2.78|
> |SingleLLM (Zero-shot)|0.39|0.91|2.15|0.14|0.14|0.36|
> |SingleLLM (SC)|0.75|1.02|8.56|0.21|0.20|1.41|
> |SingleLLM (CoT)|1.84|2.81|3.28|0.55|0.51|0.52|
> |SingleLLM (CoT-SC)|7.99|10.52|13.5|2.32|2.27|2.15|
>
> The cost table is consistent with findings on discussion rounds, showing multi-agent frameworks are generally more expensive with costs varying by framework design and task complexity. Notably, for textual QA tasks, CoT-SC is the most expensive method, surpassing even the most communication-heavy multi-agent frameworks.
>
> ## 3. On Evaluator Expertise for Task 4 (W-3)
>
> We agree clinical relevance is crucial. We have expanded our evaluation team to include a more diverse panel of `six PhD/MD students` (3 CS PhD students with expertise in AI for healthcare, 1 MD of clinical background, 1 PhD in biomedical engineering, and 1 PhD in biostatistics).
>
> All 100 workflow automation tasks have been re-evaluated by this expert panel. As detailed in our response to Reviewer SW5z, we have also calculated Fleiss' Kappa to ensure inter-rater reliability, which showed substantial agreement. We have updated the benchmarking results in Table 6 to reflect this expert clinical judgment, using a more fine-grained error taxonomy.
>
> > Updated Table 6 Results (Percentage %)
>
> |Task Type|Evaluation Category|MIMIC-IV||||TJH||||
> |---|---|:-:|:-:|:-:|:-:|:-:|:-:|:-:|:-:|
> |||Single LLM|SmolAgents|OpenManus|Owl|Single LLM|SmolAgents|OpenManus|Owl|
> |Data|Correct|80.58|90.25|65.33|50.00|67.92|70.54|78.23|37.23|
> ||No Result|16.67|4.17|20.84|34.66|1.23|3.85|0.00|32.08|
> ||Incorrect Answer|2.75|4.17|13.84|6.91|15.46|20.38|11.38|15.31|
> ||Incomplete/Partial|0.00|0.00|0.00|8.42|15.38|5.23|7.77|11.54|
> ||Correct w/ Presentation Issues|0.00|1.42|0.00|0.00|0.00|0.00|2.61|3.85|
> |Modeling|Correct|9.08|47.62|39.84|0.00|8.42|48.91|64.0|15.34|
> ||No Result|14.15|12.77|15.38|76.92|50.00|0.00|4.17|41.67|
> ||Preprocessing Only|0.00|0.00|11.61|18.08|0.00|0.00|4.17|30.83|
> ||Missing Metrics|0.00|12.85|11.54|1.31|1.42|20.91|13.92|4.17|
> ||Model Not Saved|57.46|6.23|6.23|3.69|31.83|13.50|5.42|8.00|
> ||Anomalous Numerical Results|15.46|9.00|11.54|0.00|0.00|4.17|4.17|0.00|
> ||Fails Requirements|3.85|11.54|3.85|0.00|8.34|12.50|4.17|0.00|
> |Visualization|Correct|18.09|29.25|22.34|12.5|48.69|44.92|46.23|32.08|
> ||No Visualization|41.67|8.33|8.33|58.33|25.85|23.00|23.08|43.54|
> ||Anomalous Numerical Results|23.58|30.5|33.42|20.83|5.08|3.85|7.69|10.3|
> ||Poor Readability|9.75|15.33|11.17|8.33|0.00|2.61|5.08|1.31|
> ||Info Not Extractable|4.17|9.67|6.83|0.00|0.00|0.00|2.61|0.00|
> ||Viz. Only (No Model)|1.33|1.33|8.16|0.00|11.38|8.92|15.31|6.31|
> ||Viz. Meaningless (Model Fail)|1.42|5.58|9.75|0.00|9.00|16.69|0.00|6.46|
> |Reporting|Clear Presentation|5.15|17.92|34.46|20.54|9.83|37.59|39.0|20.92|
> ||No Report|55.23|17.92|15.38|66.69|65.25|8.33|37.5|48.67|
> ||Lacks Conclusion/Summary|32.00|51.31|28.31|8.92|14.00|30.67|7.00|2.75|
> ||Anomalous Numerical Results|0.00|7.69|1.31|3.85|0.00|0.00|0.00|0.00|
> ||Too Simple (w/ Evidence)|5.08|3.92|11.61|0.00|5.33|17.92|12.33|18.0|
> ||Poor Readability|2.54|1.23|8.92|0.00|5.58|5.50|4.17|9.67|
>
> ## 4. On the Generalization of Conclusions (W-4)
>
> Our primary audience includes medical AI researchers and practitioners who need actionable insights on representative, real-world systems that they might realistically deploy or build upon. Therefore, MedAgentBoard provides a timely assessment of the SOTA multi-agent frameworks, helping the community understand where today's tools add value versus where they introduce unnecessary complexity. We will clarify in the manuscript that our findings reflect the capabilities and limitations of existing implementations in this rapidly evolving field.
>
> ## 5. On Evaluation Granularity for Task 4 (W-5)
>
> In response, we have performed a more granular error analysis (See Table above) and will be detailed in the revised manuscript. This new analysis reveals deeper insights:
>
> - `Deconstructing "Fails Requirements" in Modeling`: Our re-analysis of these errors shows two distinct root causes:
>   - Fundamental Task Misinterpretation (73% of cases): The agent aimed for the wrong analytical goal entirely.
>   - Downstream Procedural Errors (27% of cases): The agent understood the goal but failed during implementation.
> - `Analyzing "Viz. Meaningless (Model Fail)" in Visualization`: This error, where a visualization is useless due to an upstream modeling failure, also has two primary causes:
>   - Rooted in Task Misinterpretation (78% of cases): The model was flawed because the agent misunderstood the prompt, rendering the visualization invalid.
>   - Rooted in Invalid Modeling Operations (22% of cases): The agent understood the task but made an error during modeling that invalidated the subsequent plot.
> - `Quantifying "Poor Readability" in Visualization`: We now categorize these issues by severity:
>   - Severe Readability Failures (60% of cases): Issues like obscured legends that render the chart uninterpretable.
>   - Minor Cosmetic Issues (40% of cases): Problems like missing titles that affect presentation but not core understanding.
>
> This level of detail reveals that fundamental reasoning and task understanding, not just coding execution, are the primary bottlenecks.

---

> > ### Comment · Reviewer_taNB · 2025-08-01
> >
> > Thank you for the response. All my previous concerns have been solved point-by-point and I would like to keep my overall positive score.

---

> ### Author Response · Authors · 2025-08-07
>
> Dear Reviewer taNB,
>
> Thank you very much for your detailed review and for acknowledging that our rebuttal has addressed all your concerns point-by-point. We are grateful for your positive assessment of our work.
>
> Your suggestions were particularly impactful. The cost analysis you requested has added a crucial practical dimension to our benchmark, and your feedback on evaluation granularity for Task 4 pushed us to develop a much more nuanced error taxonomy and recruit a more diverse expert panel. We believe these changes would have substantially strengthened our paper.
>
> We truly appreciate your engagement. As we prepare the final version, we would be very grateful to know if you have any further suggestions or thoughts on how we might continue to improve the manuscript.
>
> Thank you again for your constructive guidance.
>
> Warmest regards,
>
> All Authors of Submission #436

---

### Official Review · Reviewer_Eef2 · 2025-07-02

**Rating:** 4
**Confidence:** 3

**Summary:**

This paper proposes the MedAgentBoard, a framework designed to evaluate the healthcare capacity between conventional method, single-agent Large Language Models (LLMs) and multi-agent collaborations. It categorizes publicly available datasets into four types of tasks: medical question answering, lay summary generation, structured Electronic Health Record (EHR) data predictive modeling, and clinical workflow automation. Finally, the paper develops a unified platform to reconcile conflicting claims about the efficacy of both single-agent and multi-agent systems in healthcare applications.

**Dataset Code Accessibility:**

Yes

**Dataset Code Comments:**

The code public on GitHub with clear user guidence.

**Ethical Comments:**

All the data this paper used are public avaliable.

**Ethical Considerations:**

No, there are no or only very minor ethics concerns

**Final Justification:**

The author have addressed my concerns, so I keep my positive score.

**Limitations Weaknesses:**

The benchmark evaluation includes conventional methods, single-agent systems, and multi-agent collaborations. However, this paper does not evaluate Large Language Models (LLMs) that are finetuned on medical data, such as MedLlama2.

**Strengths Contributions:**

This paper provides a benchmark for evaluating multi-agent collaborations, single Large Language Models (LLMs), and conventional methods across diverse medical tasks and data modalities. The benchmark includes various types of real-world healthcare tasks, advancing the application of multi-agent systems in practical healthcare settings.

---

> ### Author Rebuttal · Authors · 2025-07-30
>
> We thank the reviewer for your positive assessment and for recognizing that our benchmark advances the application of AI in practical healthcare settings by covering diverse tasks and data modalities. We appreciate the opportunity to address your concerns.
>
> ## On Benchmarking Domain-Specific LLMs like MedLlama-2 (W-1)
>
> We thank the reviewer for this highly relevant suggestion. Following your advice, we conducted new experiments with domain-specific medical LLMs.
>
> First, we tested MedLlama-2 on Task 3 (EHR predictive modeling). We discovered that the model suffered from `Critical Instruction-Following Failures`, succeeding in generating a valid, structured JSON response in fewer than 10% of trials. Even when the task was simplified to require only a single numeric output, the success rate was merely ~20%, rendering the model practically unusable for this task.
>
> We then extended our experiments to MedLlama-3, the successor to MedLlama-2. While it performed better, it still failed to follow formatting constraints in approximately 50% of cases.
>
> More importantly, our analysis of correctly formatted responses from these medical LLMs (including MedLlama-2, MedLlama-3, and OpenBioLLM-8B) revealed a `lack of data-driven reasoning`. Even when the output format was correct, the content lacked semantic validity:
> - MedLlama-2 often repeated parts of the prompt or generated non-sensical text, indicating a failure to ground its response in the provided patient data.
> - MedLlama-3 returned the exact same probability (0.85) in 100% of its correctly formatted responses. This value matches the example provided in our few-shot prompt, strongly suggesting the model was merely echoing the demonstration instead of performing genuine, data-driven inference on the new patient data.
>
> This highlights a critical gap: while these models may be specialized in medical terminology, their core instruction-following and reasoning capabilities lag significantly behind the general-purpose LLMs (e.g., GPT-4o, DeepSeek-V3, DeepSeek-R1) we benchmarked.
>
> In fact, our original benchmark for Task 3 already included several medically-aligned LLMs, such as OpenBioLLM, Gemma-3 (Google's latest model with medical applications), and HuatuoGPT-o1 (a medical reasoning model). Furthermore, DeepSeek has demonstrated state-of-the-art performance on multiple medical benchmarks, ranking first on the Stanford MedHELM leaderboard [1] and outperforming many proprietary models in a *Nature Medicine* study [2] on clinical decision-making. Its open-source nature also addresses privacy concerns, making it more practical for real-world clinical deployment compared to closed-source models that risk exposing sensitive patient data. This is why we chose DeepSeek as the primary base model for our single-LLM and multi-agent experiments.
>
> [1] Bedi, S., Cui, H., Fuentes, M., Unell, A., Wornow, M., Banda, J. M., ... & Shah, N. H. (2025). MedHELM: Holistic Evaluation of Large Language Models for Medical Tasks. arXiv preprint arXiv:2505.23802.
>
> [2] Benchmark evaluation of DeepSeek large language models in clinical decision-making
>
> **Conclusion and Benchmarking Implications:** The tasks in MedAgentBoard are designed not just to test knowledge recall but also to evaluate core capabilities critical for real-world healthcare applications, such as clinical reasoning and strict instruction adherence. Our new experiments show that while domain-specific LLMs like MedLlama-2/3 may possess valuable medical knowledge, they currently lack the robust instruction-following and reasoning abilities required for complex, structured tasks like EHR prediction. Their inclusion would not change our current conclusions.
>
> We thank you again for raising this important point. We will add a detailed discussion of these findings to the manuscript.

---

> ### Author Response · Authors · 2025-08-07
>
> Dear Reviewer Eef2,
>
> We hope this message finds you well. We sincerely appreciate the time and effort you have dedicated to reviewing our manuscript and for your valuable suggestion regarding domain-specific LLMs.
>
> In response to your feedback, we conducted new experiments with models like MedLlama-2 and MedLlama-3. Our findings, which we detailed in our rebuttal, revealed critical challenges in their instruction-following and reasoning capabilities for the complex tasks within MedAgentBoard. We believe this analysis adds an important dimension to our paper's conclusions.
>
> We are very keen to know if our rebuttal and new experiments have adequately addressed your initial concerns. As the discussion period is nearing its end, we would be deeply grateful if you could share any further thoughts or questions you might have. We are on standby and eager to engage in further discussion.
>
> Thank you once again for your thoughtful feedback and guidance.
>
> Warmest regards,
>
> All Authors of Submission #436

---

### Official Review · Reviewer_SW5z · 2025-07-03

**Rating:** 5
**Confidence:** 4

**Summary:**

This manuscript proposed the MedAgentBoard to benchmark "conventional" methods, single LLM, and multi-agent framework on four different tasks: text question answering (QA) / Visual QA, medical texts summarization (lay summary), event prediction (readmission and mortality) using Electronic Health Record (EHR), and agentic workflow (i.e., code execution for data analysis/visualization, predictive modeling, and report generation).

The major difference compared to previous benchmarking works is that the current work also included some "conventional" methods (except for agentic workflow task) and design a novel "clinical" workflow task to examine the potential of multi-agent framework in medical domain, with a primary focus on automating some workflows such as data analysis/visualization, predictive modeling, and report generation/summarization from results.

The debating points include 1. The methods selected for benchmarking are not representative enough; many strong/promising methods are left out. 2. Experiment setup is inconsistent, to some extent.

**Additional Feedback:**

1. On pages 32-34, it seems the visualization task also includes the modeling task (e.g., ID20). Why does the author have this kind of hybrid design? While some visualizations do not have a modeling task, such as ID 71

2. Table 6 does not follow previous formatting practice, i.e., bold the best performance and underline the second best.

3. It is obvious that a multi-agent system suits complex problems better than a single LLM. There is an interesting observation from the results that different multi-agent systems perform quite differently/inconsistently across different tasks. It would have a greater impact if the authors could conduct a more holistic, in-depth investigation into that (i.e., why and what caused the difference).

Given the questions and concerns the reviewer has, especially for task 4 and experiment design, which is fundamental to this manuscript's contribution, the reviewer would like to tentatively provide a borderline reject and looks forward to discussing with the authors in the rebuttal.

**Dataset Code Accessibility:**

Yes

**Dataset Code Comments:**

Data for task 4, clinical workflow automation, seems not readily accessible, available in its final form in a usable format.

**Update after revision:**

The authors uploaded the ground truth of task 4 in GitHub Releases, which fixes the previously inaccessible ground truth issue.

**Ethical Considerations:**

No, there are no or only very minor ethics concerns

**Final Justification:**

The authors' rebuttal successfully addressed all my concerns. I reviewed everyone's reviews and corresponding replies and did not identify new significant issues. Therefore, I raised my rating from 3 to 5.

**Limitations Weaknesses:**

1. Inconsistent experiment setup: In different tasks, it seems the Single LLM refers to different models. This is an issue, especially when comparing the EHR prediction task to others. In Table 5, there are 9 LLMs being tested; however, in other tasks (Tables 3,4,6), there is only one LLM candidate. What caused this inconsistency, and why would the authors adopt such an inconsistent setup?

2. Definition of "Conventional methods": The reviewer understands the authors want to include results from "conventional" methods, but the definition seems a bit problematic and inconsistent.
- There is no conventional method for VQA, or even for text QA; they are all (multimodal) language models. How do those (Gatortron, BiomedGPT, or others) fundamentally differ from LLM? They should be treated as LLM or multimodal LLM.

3. Selected benchmarking methods are not representative enough, missing some strong candidates, for example, llava-med [1], med-flamingo [2] for VQA, or Med-BERT [3] for prediction using EHR data.

4. Task 4 only contains 100 samples, which seems too limited. In addition, there is no example of such data points in the supplementary materials. From pages 31 to 47, it only has the prompt but no desired output or ground truth. This part of the data was also missing in the GitHub repo/project page. And in addition, how many people, with what kind of expertise, were involved in the evaluation was not disclosed. And what is the inter-rater reproducibility for human evaluation? This should be at least listed as a limitation in Section 5. If this dataset/task is curated in a rigorous way and a larger scale, the reviewer believes its sole contribution may warrant a higher score.

References:

1. Li, Chunyuan, et al. "Llava-med: Training a large language-and-vision assistant for biomedicine in one day." Advances in Neural Information Processing Systems 36 (2023): 28541-28564.

2. Moor, Michael, et al. "Med-flamingo: a multimodal medical few-shot learner." Machine Learning for Health (ML4H). PMLR, 2023.

3. Rasmy, Laila, et al. "Med-BERT: pretrained contextualized embeddings on large-scale structured electronic health records for disease prediction." NPJ digital medicine 4.1 (2021): 86.

**Strengths Contributions:**

1. This work not only examines LLM and multi-agent frameworks, but also benchmarks some conventional methods, providing some insights from a larger, more comprehensive scope (i.e., in what situations a multi-agent framework could be considered, given its higher computational cost).

2. Task 4, defined as Clinical Workflow Automation in the manuscript, seems to provide novel insights beyond previous benchmarking works. Previous benchmarks merely focused on VQA or visual reasoning tasks, while a task like this will provide insights into the ability to execute and act given a complex task, in an autonomous way for a multi-agent system.

These two strengths set this work apart from previous benchmarking works and could be considered significant strengths. However, there are still some debating points/weaknesses in the current format, and it is worth discussing; please find them below.

---

> ### Author Rebuttal · Authors · 2025-07-30
>
> We sincerely thank you for your feedback! We thank you for recognizing our "comprehensive benchmarking" across LLMs, multi-agent frameworks, and conventional methods, which offers valuable insights into their respective trade-offs. We thank you for highlighting the novelty of our designed `Clinical Workflow Automation` as a meaningful extension beyond prior QA-centric benchmarks. We appreciate the opportunity to clarify the points you raised!
>
> ## 1. On Inconsistent Experiment Setup of LLM selection (W-1)
>
> We thank the reviewer for this astute observation. We acknowledge that the rationale for our LLM selection strategy was not sufficiently explained. This was a methodological choice driven by the distinct research questions posed in different tasks, which we will clarify in the revised manuscript. The design of our LLM settings was guided by two primary considerations:
>
> - `Controlled Comparison of Agent Paradigms (Tasks 1, 2, 4)`: For Medical QA/VQA (Table 3), Lay Summary Generation (Table 4), and Clinical Workflow Automation (Table 6), our primary objective was to perform a controlled comparison between single-agent and multi-agent paradigms. To isolate the impact of the collaborative framework, it was crucial to use a single, powerful base model (DeepSeek-V3) for both the single-LLM baselines and as the engine for each agent. This ensures that observed performance differences are attributable to the agentic structure, not the underlying model's capability.
>
> - `Benchmarking Diverse LLMs for a Comprehensive and Cautious Understanding of LLM Performance on EHR Data (Task 3)`: In Table 5, our inclusion of a more diverse set of LLMs was motivated by two factors. (1) Lack of consensus on LLM's ability to interpret structured EHR data [1]. (2) Preliminary experiments revealed large performance gaps among different LLMs. E.g., DeepSeek-R1-7B's ~50% AUROC, indicating almost no meaningful reasoning capability on EHR, whereas the full-sized DeepSeek-R1 >80% AUROC. Therefore, we deliberately benchmarked diverse established and recent LLMs (covering general, medical, and reasoning LLMs) to assess their capabilities. This expanded set of LLMs serves as a practical guide for our target audience (e.g., medical informatics researchers), advising caution when using smaller, distilled LLMs, which may perform poorly on specialized, out-of-distribution tasks. Also, general-purpose SOTA LLMs (like DeepSeek and ChatGPT) can outperform specialized medical LLMs, as they've also been trained on extensive medical data and possess superior instruction-following and reasoning abilities, making them more suitable for a broader range of medical scenarios.
>
> [1] Sui, Yuan, et al. "Table meets llm: Can large language models understand structured table data? a benchmark and empirical study." Proceedings of the 17th ACM International Conference on Web Search and Data Mining. 2024.
>
> In response, we have also conducted additional experiments on different prompting strategies for understanding EHRs and will incorporate the updated results into the revised manuscript.
>
> |||MIMIC-IV Mortality|MIMIC-IV Mortality|MIMIC-IV Readmission|MIMIC-IV Readmission|TJH Mortality|TJH Mortality|
> |:--|:--|:-:|:-:|:-:|:-:|:-:|:-:|
> |||AUROC|AUPRC|AUROC|AUPRC|AUROC|AUPRC|
> |`DeepSeek-V3`|Basic|78.07±6.13|76.86±4.71|66.70±4.76|34.02±5.89|89.59±1.93|85.06±3.01|
> ||Optimized|79.78±4.60|43.20±9.95|65.05±4.39|31.83±5.21|88.56±2.00|81.13±3.74|
> ||Opt.+ICL|76.86±4.71|33.47±9.58|62.68±4.49|30.91±5.30|89.67±1.90|82.93±3.58|
> |`DeepSeek-R1`|Basic|73.68±7.52|33.27±9.51|65.31±4.98|38.68±6.88|90.63±1.99|83.59±3.85|
> ||Optimized|73.36±8.12|43.49±11.08|71.76±4.74|45.30±7.76|91.06±1.88|86.06±3.37|
> ||Opt.+ICL|83.95±4.60|42.10±9.95|73.92±3.78|43.59±6.42|85.59±1.97|76.87±3.56|
>
> - `Basic`: Directly feeding EHR data values.
> - `Optimized`: Additionally incorporating the unit and reference range of each feature for better LLM understanding.
> - `Opt.+ICL`: Upon the Optimized setting, additionally adding one in-context learning example.
>
> ## 2. On the Definition of "Conventional Methods" (W-2)
>
> We agree that models like GatorTron and BiomedGPT share Transformer architectures with modern LLMs. To clarify, our use of "Conventional Methods" refers not to their architecture but more to their methodological paradigm:
>
> - `Conventional Methods`: Models finetuned on task-specific datasets for a single, specialized downstream task. This reflects the long-standing practice of adapting a pretrained model for a specific application. E.g., while BiomedGPT is based on a vision-language model architecture, we train it specifically on each VQA task with distinct model parameters.
> - `LLM-based Methods`: General-purpose models applied directly via prompting (zero-shot or few-shot) without task-specific training. They demonstrate transferability and generalization across tasks and datasets, avoiding the need for task-specific components like classification heads.
>
> This distinction, we believe it is crucial for understanding how task performance varies between specialized, finetuned models and generalist LLMs applied out-of-the-box. We provide two examples to illustrate this evolution:
>
> - `Medical QA/VQA`: The mainstream approach has been supervised finetuning, often requiring a classification head for multiple-choice tasks, sometimes preceded by unsupervised pretraining. The paradigm is shifting toward general-purpose (V)LMs, reframing tasks as open-ended generation with greater generalizability. Thus, we identify M3AE, MUMC, and BiomedGPT as representative of the conventional finetuning paradigm—all are pretrained on image–text pairs and subsequently finetuned on task-specific MedVQA data.
> - `Medical Lay Summarization`: The standard approach for generating layperson-friendly summaries involves finetuning encoder-decoder Transformer models like BART, PEGASUS, and T5 on biomedical text. E.g., participants in the BioLaySumm 2023 shared task primarily used task-specific finetuned versions of BART-base/large or Flan-T5 [2]. These models, though based on Transformer principles, are tailored through supervised training on domain-specific summarization datasets.
>
> [2] Xiao, Chenghao, et al. "Overview of the biolaysumm 2025 shared task on lay summarization of biomedical research articles and radiology reports." Proceedings of the 24th Workshop on Biomedical Language Processing. 2025.
>
> ## 3. More representative baselines (W-3)
>
> - `For VQA`: We have run new experiments with LLaVA-Med and Med-Flamingo:
>
> |Model|PathVQA (MC Acc%)|VQA-RAD (MC Acc%)|VQA-RAD (FF Score)|
> |:--|:-:|:-:|:-:|
> |LLaVA-Med|59.25 ± 2.12|48.70 ± 3.04|19.94 ± 2.22|
> |Med-Flamingo|66.15 ± 1.95|45.10 ± 2.01|18.38 ± 3.05|
>
> - `For EHR Prediction`: Regarding Med-BERT, though vital, its design is for sequences of ICD codes. Our Task 3 uses raw numerical lab values and demographic data to test a model's ability to interpret non-coded, structured information, as it targets a more prevailing and important aspect of EHR data interpretation. We will clarify this distinction in the paper.
>
> ## 4. On Task 4's clarification (W-4)
>
> We thank the reviewer for highlighting these critical aspects.
>
> - `Data and Ground Truth Availability`: The full set of 100 tasks, evaluation criteria, and human-labeled ground-truth solutions are in our GitHub Repository / GitHub Releases. The specific data points for each task can be constructed from the components provided in the Appendix (task descriptions starting on Line 923, the in-context learning patient example on Line 919) and the final instructions (Line 942, Line 944), which can be assembled using the scripts released on GitHub. Moreover, we have already curated and will release a much larger dataset with over 1,000 EHR tasks covering diverse data types (ICD codes, clinical notes, structured data, etc.) grounded in established research.
> - `Evaluator Expertise and Inter-Rater Reliability`: To strengthen clinical validity, we have expanded our evaluation team to a total of `six PhD/MD students` (3 CS PhD students with expertise in AI for healthcare, 1 MD of clinical background, 1 PhD in biomedical engineering, and 1 PhD in biostatistics). All 100 tasks were re-evaluated by this panel. We now report Fleiss' Kappa (indicating inter-rater reliability) for each evaluation category, which shows moderate to substantial agreement between evaluators.
>
> |Task Category|Data|Modeling|Visualization|Reporting|
> |:--|:-:|:-:|:-:|:-:|
> |Fleiss' Kappa|0.6053|0.5566|0.5441|0.4007|
>
> ## Response to Additional Feedback
>
> - **[Feedback 1] Hybrid Design**: The integration of modeling & visualization (e.g., task ID20) is intentional, reflecting real-world analytical workflows where visualization is a key part of model interpretation. By including both integrated tasks and standalone visualization tasks (like ID71), our benchmark demonstrates this versatility and provides a more realistic testbed for agent capabilities.
> - **[Feedback 2] Table 6 Formatting**: We have corrected the formatting of Table 6.
> - **[Feedback 3] Deeper Investigation of Multi-Agent Performance (Feedback 3)**: The inconsistent effectiveness of multi-agent systems is rooted in their architectural assumptions. E.g.,
>   - ReConcile's performance can be constrained by its weakest base model, especially in VQA where Qwen-VL-Max's capabilities might have been hindered by the less capable Qwen2.5-VL.
>   - MDAgents, ColaCare and MedAgents revealed that role-playing prompts (e.g., acting as an internist, surgeon, or radiologist) often failed to elicit domain-specific knowledge effectively.
>   - SmolAgent, OpenManus, and Owl revealed issues such as irrational tool calls, inaccurate file path recognition, and getting stuck in loops.
>   - Multi-agent frameworks frequently encounter technical issues like `JSON parsing errors`, requiring multiple runs or manual intervention to correct. They also exhibit high sensitivity to prompt design.

---

> > ### Comment · Reviewer_SW5z · 2025-08-03
> > **Thank you for the clarification and new results**
> >
> > I would like to thank the authors for supplying additional results and clarifying some of my confusions. The revision addressed my concern about the inconsistent setup, and together with replies to other reviewers (e.g., about Evaluation Granularity for Task 4), makes the setup and evaluation of Task 4 clearer. I am now inclined to change my rating to positive in the final justification.
> >
> > But before that, I have two more follow-ups, all focused on Task 4:
> >
> >  - Is it possible to supply some examples, possibly in the appendix, to articulate the difference between the good output and the failure case (i.e., expand tables on pages 48-56 to include some examples).
> >  - Is it the correct understanding that for Task 4, it is open-ended and has no ground-truth (i.e., no physicians curate "gold-standard" output for different workflows? And therefore, the datapoint for Task 4 is just the 100 prompts from pages 31-47? If one wants to reproduce the workflow automation, one should connect those prompts with the corresponding dataset.
> >
> > Some other comments:
> >  - While authors were committed to open-source 1000 clinical workflow automation datapoints, Table 6 is still based on 100 samples, which is somewhat small.
> >  - Adding inter-rater repeatability is a big plus, and thank you for supplying that. The Fleiss' Kappa makes a lot of sense --- as some perspectives (modeling, visualization, and data) are somehow objective and decisive, hence a better agreement and some evaluation metrics of reporting might be more subjective.
> >
> > Great work and thank you!

---

> ### Author Response · Authors · 2025-08-03
> **Thank You! Addressing Follow-up Questions on Task 4 (Examples and Ground Truth)**
>
> We sincerely thank you for your detailed and constructive feedback. Your follow-up questions are very helpful, and we are pleased to provide the requested clarifications.
>
> ## Follow-up Question 1: Examples of Good vs. Failure Cases for Task 4
>
> To clearly illustrate the distinction between successful and failed outputs in Task 4, we will add representative examples to the appendix. E.g.,
>
> > Case 1: Data Statistics (Task ID: 11)
>
> - Task: Calculate the average range of 'White blood cell count' for patients with at least two records. The range is the difference between the maximum and minimum values for each patient.
> - Reference Answer: `23.357969...`
> - Good Output (SmolAgent): The agent correctly implemented the logic (filtering, grouping, max-min calculation, and averaging) and produced the exact result.
> ```
> 23.357969348659005
> ```
> - Failure Case (Single LLM, Judgment: Incorrect Answer): The agent generated a script with flawed logic (e.g., incorrect patient filtering), leading to an incorrect numerical result.
> ```
> 18.991993769470405
> ```
>
> > Case 2: Report Generation (Task ID: 31)
>
> - Task: Analyze the relationship between lab measurement frequency and patient outcome, selecting relevant parameters and generating a structured summary report with findings and interpretations.
> - Reference Answer: A concise, structured markdown report with clear sections and distilled insights.
> ```markdown
> # Analysis Report: Measurement Density and Patient Outcome
> ## Objective
> Investigate the relationship between lab measurement frequency and patient outcome.
> ## Methodology
> 1. Selected Parameters: Hemoglobin, White Blood Cell Count...
> 2. Analysis: Mann-Whitney U test to compare measurement density...
> ## Key Findings
> - All selected parameters showed significant differences (p < 0.001)...
> ## Interpretations
> - Higher measurement density in deceased patients may indicate more intensive monitoring...
> ```
> - Failure Case (SmolAgent, Judgment: Lacks Conclusion/Summary): The agent performed statistical tests on *all* available parameters but failed to synthesize the results. Instead of a structured report, it produced a verbose data dump of all 70+ parameters without selection, structure, or interpretation. This fails the task's core requirement of reasoning and synthesis.
> ```markdown
> Analysis Report: Relationship between Lab Measurement Frequency and Patient Outcome
> Parameter: hemoglobin
> - Median measurements (survived): 2.00, (died): 2.00
> - Mann-Whitney U test p-value: 0.4481
> - Not significant difference between groups
> ... (continues for 70+ more parameters) ...
> ```
>
> ## Follow-up Question 2: Ground Truth and Reproducibility for Task 4
>
> While the tasks are open-ended, they are not without ground truth.
>
> - `Ground Truth`: For each of the 100 tasks, we manually curated a comprehensive `reference answer`. This includes expected numerical results, data files, model checkpoints, visualizations, and/or structured reports.
>   - For `objective tasks` (e.g., data statistics), the reference is a specific, verifiable output (e.g., a single number).
>   - For `complex tasks` (e.g., report generation), the reference serves as a high-quality guideline. Human evaluators use it, along with a detailed rubric, to assess the agent's output quality, structure, and insights.
>
> - `Open-Sourcing and Reproducibility`: We have made every effort to ensure full reproducibility. You could first visit our main `MedAgentBoard` GitHub repository. The `README.md` file there provides a direct link to a separate repository dedicated to task 4: `MedAgentBoard-WorkflowAutomation`. This repository contains the complete code, the 100 task prompts, the full outputs from each agent framework we tested, and our curated reference answers for the publicly accessible TJH dataset (See GitHub Releases zone). This setup allows anyone to directly compare the agent outputs against our ground truth and replicate our evaluation.
>
> (Note: MIMIC-IV dataset, its use is governed by a data use agreement with PhysioNet. To comply with this agreement, the reference answers for MIMIC-IV tasks are available upon request to credentialed users. The reference materials were shared among expert evaluators during the assessment process.)
>
> ## Response to Other Comments
>
> - `On the sample size of Table 6`: We agree a larger sample size for Table 6 would be ideal. This was constrained by the highly intensive nature of expert evaluation; our panel spent over two workdays each assessing the outputs. To address this, we are developing a semi-automated pipeline using an `LLM-as-a-judge` with human verification to scale our benchmark to 1,000+ tasks.
> - `On Inter-Rater Repeatability`: Thank you. We are glad you found the Fleiss' Kappa analysis valuable. Your interpretation is correct—the agreement level corresponds to task objectivity. We believe this nuanced analysis is a key step toward building reliable evaluation standards for complex agentic systems.
>
> Thank you once again for your constructive engagement!

---

> > ### Comment · Reviewer_SW5z · 2025-08-04
> >
> > Thank you very much for your prompt reply and effort!
> >
> > Supplying examples of good and bad execution is important. And I was able to find the reference answer and model output in the GitHub Releases section. Given that all my concerns are addressed successfully, I will revise my rating. Please include changes made in the rebuttal in the final paper.

---

> > > ### Author Response · Authors · 2025-08-07
> > >
> > > Dear Reviewer SW5z,
> > >
> > > We would like to express our sincerest gratitude for your incredibly thorough and constructive engagement throughout the review process. Your insightful feedback and follow-up questions were instrumental in helping us significantly improve the quality and clarity of our work.
> > >
> > > We were delighted to see that our responses and revisions have successfully addressed your concerns. Your guidance has made our benchmark more rigorous and our contributions more explicit. Thank you once again for your time, expertise, and invaluable support!
> > >
> > > Warmest regards,
> > >
> > > All Authors of Submission #436

---

### Author Response · Authors · 2025-08-09
**General Rebuttal**

Dear Program Chairs, Senior Area Chairs, Area Chairs, and Reviewers,

We would like to express our sincere gratitude for your thorough and insightful review of our submission, *MedAgentBoard*. We deeply appreciate your constructive feedback and for recognizing the strengths of our work.

In particular, we appreciate that all reviewers acknowledged the importance of *MedAgentBoard* in providing a comprehensive benchmark across conventional methods, single-LLMs, and multi-agent collaborations. We thank Reviewer SW5z for highlighting the **novelty of our Clinical Workflow Automation task** and its insights beyond previous benchmarks, Reviewer Eef2 for acknowledging the benchmark's relevance to **real-world healthcare settings**, Reviewer taNB for noting our **clear writing** and that the work addresses a **critical and relevant question**, and Reviewer Hr7A for recognizing the **nuanced findings** and commitment to **reproducibility**.

We also valued the constructive suggestions that have been instrumental in significantly improving the rigor and clarity of our work. During the rebuttal period, we provided the following responses and substantial additions:

1.  **Enhanced Evaluation Rigor for Task 4 (Clinical Workflow Automation):** Following the suggestions of Reviewers SW5z, taNB, and Hr7A, we significantly strengthened the evaluation. We expanded our evaluation panel to include diverse clinical and biostatistical expertise, re-evaluated all 100 tasks, reported Fleiss' Kappa to confirm inter-rater reliability, and adopted a more fine-grained error taxonomy to distinguish between different types of failures.
2.  **Addressing Baseline Representativeness:** In response to Reviewers SW5z and Eef2, we conducted new experiments with representative baselines. We included LLaVA-Med and Med-Flamingo for VQA, and tested domain-specific LLMs like MedLlama-2/3, providing a detailed analysis on their instruction-following and reasoning limitations in our benchmark.
3.  **Cost and Efficiency Analysis:** As requested by Reviewer taNB, we provided a comprehensive cost analysis comparing multi-agent frameworks with single-LLM approaches based on discussion rounds and estimated API costs, which provides crucial practical insights.
4.  **Deeper Analysis of Success/Failure Dynamics:** Addressing Reviewer Hr7A's request for a deeper "why," we analyzed the underlying reasons for the inconsistent performance of multi-agent systems. We detailed why multi-agent frameworks succeed in workflow automation (due to task decomposition and tool use) but often fail in VQA (due to perception bottlenecks and the loss of visual information in textual communication).
5.  **Clarifying Novelty and "Conventional Methods":** We clarified the novelty of MedAgentBoard, emphasizing its role in systematically adjudicating conflicting claims in the literature and highlighting the crucial comparison with specialized conventional methods (e.g., XGBoost) that remain superior in key areas. We also clarified our definition of "conventional methods" based on their paradigm (task-specific finetuning) rather than their architecture.

We have provided comprehensive details and additional results in our individual responses, confirming that the new data and analyses continue to support the primary conclusions of our original manuscript. We commit to incorporating all the above additions into the revised manuscript.

Once again, we sincerely thank the Program Chairs, Senior Area Chairs, Area Chairs, and Reviewers for your time and dedication in reviewing our submission.

Best,

All Authors of Submission #436

---

### Decision · Program_Chairs · 2025-09-18

**Decision:**

Accept (poster)

**Comment:**

This paper introduces MedAgentBoard, a benchmark for systematically evaluating multi-agent collaboration, single LLMs, and conventional methods across a range of medical tasks. The benchmark spans diverse data modalities (text, image, structured data) and integrates established datasets and metrics.

The study is carefully executed, with clear baselines, rigorous comparisons, and open-sourced resources. Results highlight that while multi-agent frameworks provide benefits in workflow automation, they do not universally outperform single LLMs or conventional methods, and in certain tasks (e.g., medical VQA, EHR prediction) they underperform.

Strengths
- The work aims to address a timely and important question: whether complex multi-agent systems offer genuine benefits over simpler LLM setups or well-established conventional approaches in medicine.
- Proposes a comprehensive benchmark spanning four task categories and multiple modalities.
- Includes systematic comparisons across multi-agent, single LLM, and conventional baselines.
- Provides detailed implementation, uses established datasets (MIMIC-IV, MedQA) and metrics (AUROC, ROUGE).

Weaknesses
- Some tasks omit dynamic, real-time clinical scenarios (e.g., emergency triage).
- Workflow automation metric ("task completeness") emphasizes output generation over clinical accuracy/validity.
- Initial evaluation lacked finetuned LLMs and cost comparisons (added in rebuttal).
- Evaluation relied initially on a small group of CS students (before expansion in rebuttal).

Overall, despite limitations, the reviewers agree this work makes a valuable contribution by providing a well-designed benchmark to rigorously assess multi-agent vs. single-agent vs. conventional methods in medicine.
Rebuttal phase addressed reviewer concerns with additional experiments and expanded expert evaluation.
No ethics concerns were identified.